# An Empirical Process Approach to the Union Bound: Practical Algorithms for Combinatorial and Linear Bandits

**Julian Katz-Samuels**
Allen School of Computer Science & Engineering
University of Washington
jkatzsam@cs.washington.edu

**Lalit Jain**
Foster School of Business
University of Washington
lalitj@uw.edu

**Zohar Karnin**
Amazon Web Services
zkarnin@gmail.com

**Kevin Jamieson**
Allen School of Computer Science & Engineering
University of Washington
jamieson@cs.washington.edu

## Abstract

This paper proposes near-optimal algorithms for the pure-exploration linear bandit problem in the fixed confidence and fixed budget settings. Leveraging ideas from the theory of suprema of empirical processes, we provide an algorithm whose sample complexity scales with the geometry of the instance and avoids an explicit union bound over the number of arms. Unlike previous approaches which sample based on minimizing a worst-case variance (e.g. G-optimal design), we define an experimental design objective based on the Gaussian-width of the underlying arm set. We provide a novel lower bound in terms of this objective that highlights its fundamental role in the sample complexity. The sample complexity of our fixed confidence algorithm matches this lower bound, and in addition is computationally efficient for combinatorial classes, e.g. shortest-path, matchings and matroids, where the arm sets can be exponentially large in the dimension. Finally, we propose the first algorithm for linear bandits in the the fixed budget setting. Its guarantee matches our lower bound up to logarithmic factors.

## 1 Introduction

The pure exploration stochastic multi-armed bandit (MAB) problem has received attention in recent years because it offers a useful framework for designing algorithms for sequential experiments. In this paper, we consider a very general formulation of the pure exploration MAB problem, namely, *pure exploration (transductive) linear bandits* [12] : given a set of measurement vectors $\mathcal{X} \subset \mathbb{R}^d$, a set of candidate items $\mathcal{Z} \subset \mathbb{R}^d$, and an *unknown* parameter vector $\theta \in \mathbb{R}^d$, an agent plays a sequential game where at each round she chooses a measurement vector $x \in \mathcal{X}$ and observes a stochastic random variable whose expected value is $x^\top \theta$. The goal is to identify $z_* \in \arg\max_{z \in \mathcal{Z}} z^\top \theta$. This problem generalizes many well-studied problems in the literature including best arm identification [11, 21, 23, 25, 6], Top-K arm identification [22, 28, 9], the thresholding bandit problem [27], combinatorial bandits [10, 13, 8, 5, 20], and linear bandits where $\mathcal{X} = \mathcal{Z}$ [30, 33, 31].

The recent work of [12] proposed an algorithm that is within a $\log(|\mathcal{Z}|)$ multiplicative factor of previously known lower bounds [30] on the sample complexity. This term reflects a naive union bound over all informative directions $\{z_* - z : z \in \mathcal{Z} \setminus \{z_*\}\}$. Although one might be inclined to dismiss $\log(|\mathcal{Z}|)$ as a small factor, in many practical problems it can be extremely large. For example,

in Top-K $\log(|\mathcal{Z}|) = \Theta(k \log(d))$ which would introduce an additional factor of $k$ that does not appear in the upper bounds of specialized algorithms for this class [22, 8, 25]. As another example, if $\mathcal{Z}$ consists of many vectors pointing in nearly the same direction, $\log(|\mathcal{Z}|)$ can be arbitrarily large, while we show that the true sample complexity does not depend on $\log(|\mathcal{Z}|)$. Finally, in many applications of linear bandits such as content recommendation $|\mathcal{Z}|$ can be enormous and thus the factor $\log(|\mathcal{Z}|)$ can have a dramatic effect on the sample complexity.

The high-level goal of this paper is to study how the geometry of the measurement vectors $\mathcal{X}$ and the candidate items $\mathcal{Z}$ influences the sample complexity of the pure exploration transductive linear bandit problem in the moderate confidence regime. We appeal to the fundamental TIS-inequality [19] which describes the deviation of the suprema of a Gaussian process from its expectation, leading us to propose an experimental design based on minimizing the expected suprema. We make the following contributions. First, we show a novel lower bound for the non-interactive oracle MLE algorithm, which devises a fixed sampling scheme using knowledge of $\theta$. While this non-interacting lower bound is not a lower bound for adaptive algorithms, it is suggestive of what union bounds are necessary and can be a multiplicative dimension factor larger than known adaptive lower bounds. Second, we develop a new algorithm for the fixed confidence setting (defined below) that nearly matches the performance of this oracle algorithm. Moreover, this algorithm recovers many of the state-of-the-art sample complexity results for combinatorial bandits as special cases. Third, applied specifically to the combinatorial bandit setting, we develop a practical and computationally efficient algorithm. We include experiments that show that our algorithm outperforms existing algorithms, often by an order of magnitude. Finally, we show that our techniques extend to the fixed budget setting where we provide the first fixed budget algorithm for transductive linear bandits. This algorithm matches the lower bound up to a factor that in most standard settings is bounded by $\log(d)$.

## 2  Preliminaries

In the (transductive) linear bandit problem, the agent is given a set $\mathcal{X} \subset \mathbb{R}^d$ and a set of items $\mathcal{Z} \subset \mathbb{R}^d$. At each round $t$, an algorithm $\mathcal{A}$ selects a measurement $X_t \in \mathcal{X}$ which is measurable with respect to the history $\mathcal{F}_{t-1} = (X_s, Y_s)_{s<t}$ and observes a noisy observation $Y_t = X_t^\top \theta + \eta$ where $\theta \in \mathbb{R}^d$ is the unknown model parameter and $\eta$ is independent mean-0 Gaussian noise[1]. We assume that $\operatorname{argmax}_{z \in \mathcal{Z}} z^\top \theta = \{z_*\}$, and the goal is to identify $z_*$. We consider two distinct settings.

**Definition 1.** *Fixed-Confidence: Fix $\mathcal{X}, \mathcal{Z}, \Theta \subset \mathbb{R}^d$. An algorithm $\mathcal{A}$ is $\delta$-PAC for $(\mathcal{X}, \mathcal{Z}, \Theta)$ if 1) the algorithm has a stopping time $\tau$ wrt $(\mathcal{F}_t)_{t \in \mathbb{N}}$ and 2) at time $\tau$ it makes a recommendation $\widehat{z} \in \mathcal{Z}$ and for all $\theta \in \Theta$ it satisfies $\mathbb{P}_\theta(\widehat{z} = z_*) \geq 1 - \delta$.*

**Definition 2.** *Fixed-Budget: Fix $\mathcal{X}, \mathcal{Z}, \Theta \subset \mathbb{R}^d$ and a budget $T$. An algorithm $\mathcal{A}$ for fixed-budget returns a recommendation $\widehat{z} \in \mathcal{Z}$ after $T$ rounds.*

Linear bandits is popular for applications such as content recommendation, digital advertisements, and A/B testing. For instance, in content recommendation $\mathcal{X} = \mathcal{Z} \subset \mathbb{R}^d$ may be sets of feature vectors describing songs (e.g., beats per minute, genre, etc.) and $\theta \in \mathbb{R}^d$ may represent an individual user's preferences over the song library. An important sub-class of linear bandits is known as combinatorial bandits which is a focus of this work.

**Combinatorial Bandits:** In the *combinatorial bandit* setting, $\mathcal{X} = \{\mathbf{e}_1, \ldots, \mathbf{e}_d\}$ (where $\mathbf{e}_i$ is the $i$-th canonical basis vector) and $\mathcal{Z} \subset \{0, 1\}^d$. We will sometimes overload notation by treating $\mathcal{Z}$ as a collection of sets, e.g., for $z \in \mathcal{Z}$ writing $i \in z$ iff $\mathbf{e}_i^\top z = 1$. We next give some examples of the combinatorial bandit setting.

**Example 1** (MATROID). *$\mathcal{M} = (S, \mathcal{I})$ is a matroid where $S$ is a set of ground elements and $\mathcal{I} \subset 2^S$ is a collection of independent sets. This setting includes best arm identification, Top-K arm identification, identifying the minimum spanning tree with largest expected reward in a graph, and other important applications (see [7] for a list of applications).*

**Example 2** (MATCHING). *For a balanced bipartite graph with $d$ edges and $2\sqrt{d}$ vertices let $\mathcal{Z}$ denote the set of $\sqrt{d}!$ perfect bipartite matchings. The goal is to identify the matching $z \in \mathcal{Z}$ that maximizes $\theta^\top z$.*

In some of these settings, $|\mathcal{Z}|$ is exponential in the dimension $d$. For example, in the problem of finding a best matching in a bipartite graph, $|\mathcal{Z}| = (\sqrt{d})!$. In this setting a naive evaluation of $\text{argmax}_{z \in \mathcal{Z}} z^\top \theta$ by enumerating $\mathcal{Z}$ becomes impossible even if $\theta$ were known. For such problems, we assume access to a linear maximization oracle

$$\text{ORACLE}(w) = \arg\max_{z \in \mathcal{Z}} z^\top w, \tag{1}$$

which is available in many cases, including matroids, MATCHING, and identifying a shortest path in a directed acyclic graph (DAG). We will characterize the computational complexity of an algorithm in terms of the number of calls to the maximization oracle.

## 3 Review of Gaussian Processes

We now discuss how our work departs from previous approaches to the pure exploration linear bandit problem. Consider for a moment a fixed design where $n \geq d$ measurements $x_1, \ldots, x_n$ were decided before observing any data, and subsequently for each $1 \leq i \leq n$ we observe $y_i = x_i^\top \theta + \eta_i$ with $\eta_i \sim \mathcal{N}(0, 1)$. In this setting the maximum likelihood estimator (MLE) is given by ordinary least squares as $\widehat{\theta} = (\sum_{i=1}^n x_i x_i^\top)^{-1} \sum_{i=1}^n y_i x_i$. Substituting the value of $y_i$ into this expression, we obtain $\widehat{\theta} = \theta + (\sum_{i=1}^n x_i x_i^\top)^{-1/2} \eta$ in distribution where $\eta \sim \mathcal{N}(0, I_d)$. After collecting $\{(x_i, y_i)\}_{i=1}^n$ and computing $\widehat{\theta}$, the most reasonable estimate for $z_* = \arg\max_{z \in \mathcal{Z}} z^\top \theta$ is just $\widehat{z} = \arg\max_{z \in \mathcal{Z}} z^\top \widehat{\theta}$. The good event that $\widehat{z} = z_*$ occurs if and only if $(z_* - z)^\top \widehat{\theta} > 0$ for all $z \in \mathcal{Z} \setminus \{z_*\}$. Since $\widehat{\theta}$ is a Gaussian random vector, for each $z \in \mathcal{Z}$, $(z_* - z)^\top (\widehat{\theta} - \theta) \sim \mathcal{N}(0, (z_* - z)^\top (\sum_{i=1}^n x_i x_i^\top)^{-1} (z_* - z))$. If we apply a standard sub-Gaussian tail-bound with a union bound over all $z \in \mathcal{Z} \setminus \{z_*\}$, then we have with probability greater than $1 - \delta$ that

$$(z_* - z)^\top \widehat{\theta} \geq (z_* - z)^\top \theta - \sqrt{2\|z_* - z\|_{A^{-1}}^2 \log(|\mathcal{Z}|/\delta)} \tag{2}$$

for all $z \in \mathcal{Z} \setminus \{z_*\}$ *simultaneously*, where we have taken $A = \sum_{i=1}^n x_i x_i^\top$ and used the notation $\|x\|_W^2 = x^\top W x$ for any square $W$. Thus, we conclude that if $n$ and $\{x_1, \ldots, x_n\}$ are chosen such that $\max_{z \in \mathcal{Z}} \frac{2\|z_* - z\|_{(\sum_{i=1}^n x_i x_i^\top)^{-1}}^2 \log(|\mathcal{Z}|/\delta)}{((z_* - z)^\top \theta)^2} > 1$ then with probability at least $1 - \delta$ we will have that $(z_* - z)^\top \widehat{\theta} > 0$ for all $z \in \mathcal{Z} \setminus \{z_*\}$ and consequently, $\widehat{z} = z_*$. This simple argument is the core of all approaches to pure exploration linear bandits until this paper [30, 23, 33, 12]. However, applying a naive union bound over all $z \in \mathcal{Z}$ can be extremely weak and does not exploit the geometry of $\mathcal{Z}$ that induces many correlations among the random variables $(z_* - z)^\top (\widehat{\theta} - \theta)$.

At the heart of our approach is the following concentration inequality for the suprema of a Gaussian process (Theorem 5.8 in [2]).

**Theorem 1** (Tsirelson-Ibragimov-Sudakov Inequality [19]). *Let $\mathbb{S} \subset \mathbb{R}^d$ be bounded. Let $(V_s)_{s \in \mathbb{S}}$ be a Gaussian process such that $\mathbb{E}[V_s] = 0$ for all $s \in \mathbb{S}$. Define $\sigma^2 = \sup_{s \in \mathbb{S}} \mathbb{E}[V_s^2]$. Then, for all $u > 0$,*

$$\mathbb{P}(|\sup_{s \in \mathbb{S}} V_s - \mathbb{E} \sup_{s \in \mathbb{S}} V_s| \geq u) \leq 2 \exp\left(\frac{-u^2}{2\sigma^2}\right).$$

Setting $\mathbb{S} = \mathcal{Z}$, we can apply this to the Gaussian process $V_z := (z_* - z)^\top (\widehat{\theta} - \theta) = (z_* - z)^\top (\sum_{i=1}^n x_i x_i^\top)^{-1/2} \eta$ where, again, $\eta \sim \mathcal{N}(0, I_d)$. We then have with probability at least $1 - \delta$

$$(z_* - z)^\top \widehat{\theta} \geq (z_* - z)^\top \theta - \mathbb{E}_\eta \left[ \sup_{z \in \mathcal{Z} \setminus \{z_*\}} (z_* - z)^\top A^{-1/2} \eta \right] - \sqrt{2 \sup_{z \in \mathcal{Z}} \|z_* - z\|_{A^{-1}}^2 \log(\frac{1}{\delta})}$$

for all $z \in \mathcal{Z} \setminus \{z_*\}$ simultaneously. This bound naturally breaks into two components. The second-term is the *high-probability* term, and as the discussion above implies, naturally motivates the experimental design objective $\min_{x_1, \cdots x_n} \max_{z \in \mathcal{Z} \setminus \{z_*\}} \|z_* - z\|_{(\sum_{i=1}^n x_i x_i^\top)^{-1}}^2$ from past works on linear-bandit pure exploration. The first term, $\mathbb{E}_{\eta \sim N(0, I_d)} \left[ \sup_{z \in \mathcal{Z} \setminus \{z_*\}} (z_* - z)^\top (\sum_{i=1}^n x_i x_i^\top)^{-1/2} \eta \right]$ is

the *Gaussian-width* of the set $\{\left(\sum_{i=1}^n x_i x_i^\top\right)^{-1/2}(z_* - z)\}_{z\in\mathcal{Z}\setminus\{z_*\}}$ [32]. This term represents the penalty we pay for the union bound over the possible values of $\mathcal{Z}$ and reflects the underlying geometry of our arm set. For moderately sized values of $\delta \in (0, 1)$ such as the science-stalwart $\delta = 0.05$, the Gaussian width term can be substantially larger than the high probability term. Analogous to above, this motivates choosing $x_1, \cdots, x_n$ to minimize the Gaussian width term.

**Relaxation to Continuous Experimental Designs.** In practice, optimizing over all finite sets of $\mathcal{X}$ of size $n$ to minimize an experimental design objective is NP-hard. Define $\boldsymbol{\Delta} := \{\lambda \in \mathbb{R}^{|\mathcal{X}|} : \sum_i \lambda_i = 1, \lambda_i \geq 0\}$ to be the simplex over elements $\mathcal{X}$ and define $A(\lambda) = \sum_{x\in\mathcal{X}} \lambda_x x x^\top$ where $\lambda \in \boldsymbol{\Delta}$ denotes a convex combination of the measurement vectors. Defining the design that minimizes the high probability term motivates the definition

$$\rho^* := \inf_{\lambda\in\boldsymbol{\Delta}} \rho^*(\lambda) \qquad \text{where} \qquad \rho^*(\lambda) := \sup_{z\in\mathcal{Z}\setminus\{z_*\}} \frac{\|z_* - z\|_{A(\lambda)^{-1}}^2}{(\theta^\top(z_* - z))^2}.$$

On the other hand, minimizing the Gaussian width term motivates the definition

$$\gamma^* := \inf_{\lambda\in\boldsymbol{\Delta}} \gamma^*(\lambda) \qquad \text{where} \qquad \gamma^*(\lambda) := \mathbb{E}_{\eta\sim N(0,I)}[\sup_{z\in\mathcal{Z}\setminus\{z_*\}} \frac{(z_* - z)^\top A(\lambda)^{-1/2}\eta}{\theta^\top(z_* - z)}]^2.$$

While the above suggests the importance of the quantities $\rho^*$ and $\gamma^*$, we will show later how they are intrinsic to the problem hardness. For now, we point out that these quantities are easily relatable.

**Proposition 1.** *There exists universal constants* $c, c' > 0$ *such that for any* $\mathcal{X}$ *and* $\mathcal{Z}$ *we have*
$c\rho^* - \inf_{z\neq z_*} \inf_{\lambda\in\boldsymbol{\Delta}} \frac{\|z_* - z\|_{A(\lambda)^{-1}}^2}{(\theta^\top(z_* - z))^2} \leq \gamma^* \leq \min(c' \log(|\mathcal{Z}|)\rho^*, d\rho^*).$

Typically, $\inf_{z\neq z_*} \inf_{\lambda\in\boldsymbol{\Delta}} \frac{\|z_* - z\|_{A(\lambda)^{-1}}^2}{(\theta^\top(z_* - z))^2} \ll \rho^*$, in which case $\rho^* \lesssim \gamma^*$. While there are instances where $\gamma^* = \Theta(d\rho^*)$, the upper bound is not necessarily tight.

**Proposition 2.** *There exists an instance of transductive linear bandits where* $\gamma^* \geq cd\rho^*$, *and a separate instance for which* $\gamma^* \leq c' \log(d)\rho^*$ *where* $c, c' > 0$ *are universal constants.*

## 4 Towards the true sample complexity

This section formally justifies the quantities $\rho^*$ and $\gamma^*$ defined above. The following result holds for any $\mathcal{X}$ and $\mathcal{Z}$ and was first proven in this generality in [12], extending [30, 29, 8].

**Theorem 2** (Lower bound for any adaptive algorithm [12]). *For any* $\delta \in (0, 1)$, *any* $\delta$-*PAC algorithm wrt* $(\mathcal{X}, \mathcal{Z}, \mathbb{R}^d)$ *with stopping time* $\tau$ *satisfies* $\mathbb{E}_\theta[\tau] \geq \log(\frac{1}{2.4\delta})\rho^*.$

Mirroring the approaches developed in [24, 8, 14], it is possible to develop an algorithm that satisfies $\lim_{\delta\to 0} \frac{\mathbb{E}_\theta[\tau]}{\log(\frac{1}{\delta})} = \rho^*$, demonstrating the tightness of Theorem 2 in the regime of $\delta$ tending towards 0. However, for fixed $\delta \in (0, 1)$, algorithms for linear bandits to date have only been able to match this lower bound up to additive factors of $d\rho^*$ or $\log(|\mathcal{Z}|)\rho^*$ [24, 12] (note, this does not rule out optimality as $\delta \to 0$). In particular, the lower and the upper bounds of linear bandits do not reflect the underlying geometry of general sets $\mathcal{X}$ and $\mathcal{Z}$ in union bounds and are loose in general. For example, in the well-studied case of Top-K, these bounds do not capture some additive factors that are necessary and achievable in addition to $\rho^*$ alone [28, 9].

As a step towards characterizing the true sample complexity, we next demonstrate a lower bound that incorporates the geometry of $\mathcal{X}$ and $\mathcal{Z}$ for, presumably, the best possible *non-interactive* algorithm. Precisely, the procedure chooses an allocation $\{x_{I_1}, x_{I_2}, \dots\} \in \mathcal{X}$, then observes $\{y_{I_1}, y_{I_2}, \dots\} \in \mathbb{R}$ where $y_{I_t} \sim \mathcal{N}(x_{I_t}^\top\theta, 1)$, and finally forms the MLE $\widehat{\theta} = \arg\min_{\tilde{\theta}} \sum_t (y_{I_t} - x_{I_t}^\top\tilde{\theta})^2$ and outputs $\widehat{z} = \arg\max_{z\in\mathcal{Z}} z^\top\widehat{\theta}$. We emphasize that this procedure can pick any allocation; in particular, it can use the allocation that achieves $\rho^*$.

**Theorem 3** (Lower bound for non-interactive MLE). *Fix* $\mathcal{X}, \mathcal{Z} \subset \mathbb{R}^d$ *and a problem* $\theta \in \mathbb{R}^d$. *Let* $\delta \in (0, 0.015]$ *and* $c > 0$ *be a universal constant. If the non-interactive MLE uses less than* $c(\gamma^* + \log(1/\delta)\rho^*)$ *samples on the problem instance* $(\mathcal{X}, \mathcal{Z}, \theta)$, *it makes a mistake with probability at least* $\delta$.

**Input:** Confidence level $\delta \in (0,1)$, rounding parameter $\epsilon \in (0,1)$ with default value of $\frac{1}{10}$;
$\mathcal{Z}_1 \longleftarrow \mathcal{Z}, k \longleftarrow 1, \delta_k \longleftarrow \delta/2k^2$ ;
$B := \inf_{\lambda \in \Delta} \mathbb{E}_{\eta \sim N(0,I)}[\max_{z,z' \in \mathcal{Z}}(z-z')^{\top} A(\lambda)^{-1/2}\eta]^2 + 2\log(\frac{1}{\delta_1})\max_{z,z' \in \mathcal{Z}} \|z-z'\|^2_{A(\lambda)^{-1}} \vee 1$;
**while** $|\mathcal{Z}_k| > 1$ **do**
    Let $\lambda_k$ and $\tau_k$ be the solution and value of the following optimization problem

$$\inf_{\lambda \in \Delta} \tau(\lambda; \mathcal{Z}_k) := \mathbb{E}_{\eta \sim N(0,I)}[\max_{z,z' \in \mathcal{Z}_k}(z-z')^{\top} A(\lambda)^{-1/2}\eta]^2 + 2\log(\frac{1}{\delta_k})\max_{z,z' \in \mathcal{Z}_k} \|z-z'\|^2_{A(\lambda)^{-1}}$$

    Set $N_k \longleftarrow \left\lceil 2(1+\epsilon)\tau_k(\frac{2^{k+1}}{B})^2 \right\rceil \vee q(\epsilon)$ and find $\{x_1, \ldots, x_{N_k}\} \longleftarrow \text{ROUND}(\lambda_k, N_k, \epsilon)$;
    Pull arms $x_1, \ldots, x_{N_k}$ and receive rewards $y_1, \ldots, y_{N_k}$;
    Let $\widehat{\theta}_k \longleftarrow (\sum_{s=1}^{N_k} x_s x_s^{\top})^{-1} \sum_{s=1}^{N_k} x_s y_s$ ;
    $\mathcal{Z}_{k+1} \longleftarrow \mathcal{Z}_k \setminus \{z \in \mathcal{Z}_k : \exists z' \text{ such that } (z'-z)^{\top}\widehat{\theta}_k - \frac{B}{2^{k+1}} \geq 0\}$;
    $k \longleftarrow k+1$
**return** $\mathcal{Z}_k = \{\widehat{z}\}$.

**Algorithm 1:** Fixed Confidence Peace. See text for explanation of ROUND sub-routine.

By Proposition 2, $\gamma^*$ can be larger than $\rho^*$ by a multiplicative factor of the dimension $d$, demonstrating that the lower bound of Theorem 3 can be much larger than the lower bound of Theorem 2. While there exists problem instances in which the best known adaptive algorithm can achieve a sample complexity strictly smaller than the lower bound of Theorem 3 (e.g., best-arm identification), we are unaware of any settings in which the sample complexity of the best adaptive algorithm improves over Theorem 3 by more than a factor of $\log(d)$, which is typically considered insignificant.

## 5 Fixed Confidence Setting Algorithms

In this section, we present Algorithm 1, Peace, that achieves the state-of-the-art sample complexity for (transductive) linear bandits in the fixed confidence setting. In each round $k$ we eliminate from the set of candidates $\mathcal{Z}$ all the elements that are roughly $2^{-k}$ suboptimal. In each round the query allocation is fixed according to the best non-adaptive strategy.

Our algorithm must round a design to an integral solution. It uses an efficient rounding procedure ROUND$(\lambda, N, \epsilon)$ that for $\lambda \in \Delta$ and $N \geq q(\epsilon)$ returns $\kappa \in \mathbb{N}^{|\mathcal{X}|}$ such that $\sum_{x \in \mathcal{X}} \kappa_x = N$ and $\tau(\kappa; Z') \leq (1+\epsilon)\tau(N\lambda; Z')$ [1]. It suffices to take $q(\epsilon) = O(d/\epsilon^2)$ (see the Supplementary Material). Define $S_k := \{z \in \mathcal{Z} : \theta^{\top}(z_* - z) \leq B2^{-k}\}$, $\Delta_z := \theta^{\top}(z_* - z)$, and $\Delta_{min} := \min_{z \in \mathcal{Z} \setminus \{z_*\}} \Delta_z$.

**Theorem 4.** *With probability at least $1 - \delta$, Algorithm 1 terminates and returns $z_*$ after a number of samples no more than*

$$[\gamma^* + \rho^* \log(\log(\tfrac{B}{\Delta_{min}})/\delta)]c\min(\log(\tfrac{B}{\Delta_{min}}), \log(\tfrac{B}{\min_{k:|S_k|>1}\min_{\lambda \in \Delta}\tau(\lambda;S_k)})) + cd\log(\tfrac{B}{\Delta_{min}}).$$

$\log(B) = O(\log(d))$ when $\mathcal{X} = \mathcal{Z}$ and in combinatorial bandits, and $B$ can be replaced by an upper bound on $\max_{z \in \mathcal{Z}} \Delta_z$ when one is known. $\tau(\lambda; \mathcal{Z}_k)$ can be optimized using stochastic mirror descent; we show that after a suitable transformation, it is convex in the combinatorial bandit setting. We conjecture that it is convex in the general case, as well.

While our upper bound has an extra additive factor of $d$ compared to the lower bound of Theorem 3, this factor is necessary in many cases. The following theorem shows that in the combinatorial setting, an additive factor of $d$ is necessary if the agent has no apriori knowledge about $\theta$.

**Theorem 5.** *Consider the combinatorial setting where $\mathcal{X} = \{\mathbf{e}_1, \ldots, \mathbf{e}_d\}$ and $\mathcal{Z} \subset \{0,1\}^d$. Let $\delta \in (0, 1/4)$. Fix $\theta \in \mathbb{R}^d$ such that $\text{argmax}_{z \in \mathcal{Z}} z^{\top}\theta$ is unique. If an algorithm $\mathcal{A}$ is $\delta$-PAC wrt $(\mathcal{X}, \mathcal{Z}, \mathbb{R}^d)$, then $\mathbb{E}_{\theta}[\sum_{i=1}^d T_i] \geq \frac{d}{2}$ where $T_i$ denotes the number of times that $\mathcal{A}$ pulls $\mathbf{e}_i$.*

The intuition behind the argument in Theorem 9 is that if $\Omega(d)$ directions are not explored with constant probability, then there is some $\theta_i$ that the algorithm has no information about with constant probability. Thus, an adversary can perturb $\theta_i$ to alter the best $z$, making the agent incorrect with a constant probability, which contradicts the $\delta$-PAC assumption.

## 5.1 Computationally Efficient Algorithm for Combinatorial Bandits

A drawback of Algorithm 1 is that it is computationally inefficient when $|\mathcal{Z}|$ is exponentially large in the dimension. In this section, we develop an algorithm for combinatorial bandits that is computationally efficient when the *linear maximization oracle* defined in (1) is available. We introduce the following notation for a set $Z' \subset \mathcal{Z}$:

$$\gamma(Z') := \min_{\lambda \in \mathbf{\Delta}} \mathbb{E}[\sup_{z,z' \in Z'} (z - z')^\top A^{-1/2}(\lambda)\eta]^2. \tag{3}$$

We also introduce the subroutine $\text{UNIQUE}(\mathcal{Z}, \widehat{\theta}_k, 2^{-k}\Gamma)$, which uses calls to the linear maximization oracle to determine whether the gaps are sufficiently well-estimated to terminate (see the Supplementary Material).

---

**Input:** Confidence level $\delta > 0$, rounding parameter $\epsilon \in (0,1)$ with default value of $\frac{1}{10}$, $\alpha > 0$ ($\alpha = 42941$ suffices though this is wildly pessimistic; we recommend using $\alpha = 4$) ;

$\widehat{\theta}_0 = \mathbf{0} \in \mathbb{R}^d$, $\Gamma \longleftarrow \gamma(\mathcal{Z}) \vee 1$, $\delta_k \longleftarrow \frac{\delta}{2k^3}$ ;

**for** $k = 0, 1, 2, \ldots$ **do**

    $\tilde{z}_k \longleftarrow \arg\max_{z \in \mathcal{Z}} \widehat{\theta}_k^\top z$;

    Let $\lambda_k, \tau_k$ be the solution and value of the following optimization problem

$$\inf_{\lambda \in \mathbf{\Delta}} \mathbb{E}_{\eta \sim N(0,I)}[\max_{z \in \mathcal{Z}} \frac{(\tilde{z}_k - z)^\top A(\lambda)^{-1/2}\eta}{2^{-k}\Gamma + \widehat{\theta}_k^\top(\tilde{z}_k - z)}]^2 \tag{4}$$

    Set $N_k \longleftarrow \alpha \lceil \tau_k \log(1/\delta_k)(1+\epsilon) \rceil \vee q(\epsilon)$ and find $\{x_1, \ldots, x_{N_k}\} \longleftarrow \text{ROUND}(\lambda_k, N_k)$;

    Pull arms $x_1, \ldots, x_{N_k}$ and receive rewards $y_1, \ldots, y_{N_k}$;

    Let $\widehat{\theta}_{k+1} \longleftarrow (\sum_{s=1}^{N_k} x_s x_s^\top)^{-1} \sum_{s=1}^{N_k} x_s y_s$ ;

    **if** $\text{UNIQUE}(\mathcal{Z}, \widehat{\theta}_k, 2^{-k}\Gamma)$ **then return** $\tilde{z}_k$

**Algorithm 2:** Fixed Confidence Peace with a linear maximization oracle.

---

The objective (4) in Algorithm 2 acts a surrogate for $\gamma^*$ that becomes increasingly accurate over the course of the game. Enough samples are taken at round $k$ to ensure with high probability $\widehat{\theta}_k^\top(\tilde{z}_k - z) \approx \Delta_z$ for all $z \in \mathcal{Z}$ such that $\Delta_z \geq 2^{-k}\Gamma$. Thus, at round $k$, (4) behaves approximately as $\mathbb{E}_{\eta \sim N(0,I)}[\max_{z \in \mathcal{Z}} \frac{(z_* - z)^\top A(\lambda)^{-1/2}\eta}{2^{-k}\Gamma + \Delta_z}]^2$. As such, (4) ensures that *(i)* Algorithm 2 does not take too many sample at any round and *(ii)* enough samples are taken to estimate $\Delta_z$ for each $z \in \mathcal{Z}$ at a progressively finer level of granularity.

In the Supplementary Material, we provide procedures for computing $\gamma(\mathcal{Z})$ and (4) only using calls to the linear maximization oracle. The main challenge is to compute an unbiased estimate of the gradient of the objective in (4) (for an appropriate first-order optimization procedure such as stochastic mirror descent), which we now sketch. Since the expectation in (4) is non-negative, it suffices to optimize the square root of the objective function in (4). Writing $g(\lambda; \eta; z) = \frac{(\tilde{z}_k - z)^\top A(\lambda)^{-1/2}\eta}{2^{-k}\Gamma + \widehat{\theta}_k^\top(\tilde{z}_k - z)}$, since we may exchange the gradient with respect to $\lambda$ and the expectation over $\eta$, to obtain an unbiased estimate, it suffices to draw $\eta \sim N(0,I)$, and compute $\nabla_\lambda \max_{z \in \mathcal{Z}} g(\lambda; \eta; z)$. Since for a collection of differentiable functions $\{h_1, \ldots, h_l\}$, a sub-gradient $\nabla_y \max_i h_i(y)$ is simply $\nabla_y h_0(y)$ where $h_0(y) = \arg\max_i h_i(y)$, it suffices to find $\arg\max_{z \in \mathcal{Z}} g(\lambda; \eta; z)$. We reformulate this optimization problem as the following equivalent linear program:

$$\min_s s \qquad \text{subject to } \max_{z \in \mathcal{Z}} (\tilde{z}_k - z)^\top A(\lambda)^{-1/2}\eta - s[2^{-k}\Gamma + \widehat{\theta}_k^\top(\tilde{z}_k - z)] \leq 0 \tag{5}$$

A call to the linear maximization oracle can check whether the constraint in (5) is satisfied so the above linear program can be solved using binary search and multiple calls to the maximization oracle.

It would be ideal to also design a surrogate for $\rho^*$ that can be optimized using linear maximization oracle calls in a similar way to (4). Unfortunately, the above technique appears to fail since $\max_{z \in \mathcal{Z}} \left\| \frac{(\tilde{z}_k - z)}{2^{-k}\Gamma + \widehat{\theta}_k^\top(\tilde{z}_k - z)} \right\|^2_{A(\lambda)^{-1}}$ contains quadratic terms that cannot be optimized using linear maximization oracle calls. Fortunately, leveraging properties of Gaussian width, we show that optimizing (4) leads to only a small loss in sample complexity.

**Algorithm 3:** Fixed Budget Peace

**Theorem 6.** *Consider the combinatorial bandit setting. With probability at least $1 - 4\delta$ Algorithm 2 terminates and returns $z_*$ after at most*

$$[(\gamma^* + \rho^*) \log(\log(\gamma(\mathcal{Z})/\Delta_{min})/\delta) + d]c \log(\gamma(\mathcal{Z})/\Delta_{min})$$

*samples and if $\delta \in (\frac{1}{2^d}, 1)$, then with probability at least $1 - 4\delta$, the number of oracle calls is upper bounded by*

$$\tilde{O}([d + \log(\frac{d[max_{z \in \mathcal{Z}}\Delta_z + \Gamma]}{\Delta_{min}\delta})] \log(d)^2 \frac{d^3}{\Delta_{min}^2} \frac{\log(\gamma(\mathcal{Z})/\Delta_{min})^5}{\delta^2})$$

Theorem 6 nearly matches the sample complexity of Theorem 4. The latter scales like $\gamma^* + \rho^* \log(1/\delta)$ whereas the former scales like $(\gamma^* + \rho^*) \log(1/\delta)$, reflecting a tradeoff of statistical efficiency for computational efficiency. It is unknown if this tradeoff is necessary.

## 6 Fixed Budget Setting

Next, we turn to the fixed budget setting, where the goal is to minimize the probability of returning a suboptimal item $z \in \mathcal{Z} \setminus \{z_*\}$ given a budget of $T$ total measurements. Algorithm 3 is a generalization of the successive halving algorithm [23] and the first algorithm for fixed-budget linear bandits. It divides the budget into equally sized epochs and progressively shrinks the set of candidates $\mathcal{Z}_k$. In each epoch, it computes a design that minimizes $\gamma(\mathcal{Z}_k)$ and samples according to a rounded solution. At the end of an epoch, it sorts the remaining items in $\mathcal{Z}_k$ by their estimated rewards and eliminates enough of the items with the smallest estimated rewards to ensure that $\gamma(\mathcal{Z}_{k+1}) \leq \frac{\gamma(\mathcal{Z}_k)}{2}$.

**Theorem 7.** *Suppose that $\gamma(\{z, z_*\}) \geq 1$ for all $z \in \mathcal{Z} \setminus \{z_*\}$. Then, if $T \geq cmax([\rho^* + \gamma^*], d) \log(\gamma(\mathcal{Z}))$, Algorithm 3 returns $\widehat{z} \in \mathcal{Z}$ such that*

$$\mathbb{P}(\widehat{z} \neq z_*) \leq 2 \lceil \log(\gamma(\mathcal{Z})) \rceil \exp(-\frac{T}{c'[\rho^* + \gamma^*]\log(\gamma(\mathcal{Z}))}).$$

We note that the combinatorial bandit setting satisfies the assumption that $\gamma(\{z, z_*\}) \geq 1$ for all $z \in \mathcal{Z} \setminus \{z_*\}$, but this lower bound is unessential and the algorithm can be modified to accommodate another lower bound. Theorem 7 implies that if $T \geq O(\log(1/\delta)[\rho^* + \gamma^*] \log(\gamma(\mathcal{Z})) \log(\log(\gamma(\mathcal{Z}))))$, then Algorithm 3 returns $z_*$ with probability at least $1 - \delta$. Finally, $\log(\gamma(\mathcal{Z}))$ is $O(\log(d))$ in many cases, e.g., combinatorial bandits and in linear bandits when $\mathcal{X} = \mathcal{Z}$.

## 7 Discussion and Prior Art

**Transductive Linear Bandits:** There is a long line of work in pure-exploration linear bandits [30, 33, 31] culminating in the formulation of the transductive linear bandit problem in [12] where the authors developed the first algorithm to provably achieve $\rho^* \log(|\mathcal{Z}|/\delta)$. The sample complexity of Theorem 4, $\gamma^* + \rho^* \log(1/\delta)$, is never worse than [12] since $\gamma^* \leq \rho^* \log(|\mathcal{Z}|)$ by Proposition 1.

On the other hand, it is possible to come up with examples where $\gamma^*$ does not scale with $|\mathcal{Z}|$, but just $\rho^*$ (see experiments). While our algorithms work for arbitrary $\mathcal{X}, \mathcal{Z} \subset \mathbb{R}^d$, problem instances of combinatorial bandits most clearly illustrate the advances of our new results over prior art.

**Combinatorial Bandits:** The pure exploration combinatorial bandit was introduced in [10], and followed by [13]. These papers are within a $\log(d)$ factor of the lower bound for the setting where $\mathcal{Z}$ is a matroid. If $\tilde{\Delta}_i = \theta^\top z_* - \max_{z \in \mathcal{Z}: i \in z} \theta^\top z$ when $i \notin z_*$ and $\theta^\top z_* - \max_{z \in \mathcal{Z}: i \notin z} \theta^\top z$ otherwise, then a lower bound is known to scale as $\sum_{i=1}^d \tilde{\Delta}_i^{-2} \log(1/\delta)$. The following result shows that $\gamma^*$ is within $\log(d)$ of the lower bound, implying that our sample complexity scales as $\sum_{i=1}^d \tilde{\Delta}_i^{-2} \log(d/\delta)$.

**Proposition 3.** *Consider the combinatorial bandit setting and suppose that $\mathcal{Z}$ is a matroid. Then, $\gamma^* \le c \log(d) \sum_{i=1}^d \tilde{\Delta}_i^{-2}$ for some absolute constant c.*

However, in the general setting where $\mathcal{Z}$ is not necessarily a matroid, [8] points out a class with $|\mathcal{Z}| = 2$ where the sample complexity of [10, 13] is loose by a multiplicative factor of $d$. Chen et al. [8] was the first to provide a lower bound equivalent to $\rho^* \log(1/\delta)$ for the general combinatorial bandit problem, as well as an upper bound of $\rho^* \log(|\mathcal{Z}|/\delta)$. However, as stressed in the current work, the $\log(|\mathcal{Z}|)$ term is not necessary in many scenarios; for example, in Top-K, $\rho^* \log(|\mathcal{Z}|)$ is larger than the best achievable sample complexity by a multiplicative factor of $k$ [9, 28]. This is not in contradiction with the lower bound provided in Theorem 1.9 of [8] which provides a specific worst-case class of instances where the $\log(|\mathcal{Z}|)$ is needed.

The next technological leap in combinatorial bandits is the algorithm of [5] (and the follow-up [20]). They provided an algorithm with a novel sample complexity that replaces $\log(|\mathcal{Z}|)$ with a more geometrically inspired term. Define the sphere $B(z, r) = \{z' \in \mathcal{Z} : \|z - z'\|_2 = r\}$, and the complexity parameter $\varphi_i := \max_{z \in \mathcal{Z} \setminus \{z_*\}: i \in z_* \Delta z} \frac{\|z-z'\|_2^2 \log(d|B(z_*, \|z-z'\|_2)|)}{\Delta_z^2}$. Then [5] provide a sample complexity scaling like $\varphi^* := \sum_{i=1}^n \varphi_i$. The following shows that $\gamma^*$ is never more than $\log\log(d)$ larger than this complexity.

**Proposition 4.** *Consider the combinatorial bandit setting. Then, $\gamma^* \le O(\varphi^* \log(\log(d)))$.*

However, for even these sample complexity results that take the geometry into account, there exist clear examples of looseness that our approach avoids.

**Proposition 5.** *There exists an instance of Top-K where $\varphi^* = \Omega(k \log(d) \rho^*)$ but $\gamma^* = O(\log(d) \rho^*)$.*

In summary, we have the first algorithm with a sample complexity that simultaneously is nearly optimal for matroids, essentially matches our novel lower bound $\gamma^* + \log(1/\delta) \rho^* \le \log(|\mathcal{Z}|/\delta) \rho^*$, and is never worse than the sample complexity $\varphi^*$ from [5, 20].

**Computational Results in Combinatorial Bandits:** The algorithm CLUCB from [10] is computationally efficient and user-friendly. [5] and [8] provide computationally efficient algorithms, but their running times scale very poorly with problem-dependent parameters, making these algorithms impractical and we are unaware of any implementations.

# 8 Experiments

**Combinatorial Bandits:** We use $\delta = 0.05$ on all the experiments and the empirical probability of failure never exceeded $\delta$ in all of our experiments. We consider three combinatorial structures. *(i) Matching:* we use a balanced complete bipartite graph $G = (U \cup V, E)$ where $|U| = |V| = 14$. Note that $|\mathcal{Z}| = 14! \ge 8 \cdot 10^{10}$. We took two disjoint matchings $M_1$ and $M_2$ and set $\theta_e = 1$ if $e \in M_1$ and $\theta_e = 1 - h$ if $e \in M_2$ for $h \in \{.15, .1, .05, .025\}$. Otherwise, $\theta_e = 0$. *(ii) Shortest Path:* we consider a DAG where a source leads into two disjoint feed-forward networks with 26 width-2 layers that then lead into a sink (see Figure 2 for an illustration). Note that $|\mathcal{Z}| \ge 10^8$. We consider two paths $P_1$ and $P_2$ such that they are in the disjoint feed-forward networks. We set $\theta_e = 1$ if $e \in P_1$ and $\theta_e = 1 - h$ if $e \in P_2$ for $h \in \{.2, .15, .1, .05\}$. Otherwise, $\theta_e = -1$.

*(iii) Biclique:* In the biclique problem, we are given a complete balanced bipartite graph with $\sqrt{d}$ nodes in each group. $\mathcal{Z}$ is the set of bicliques with $\sqrt{s}$ nodes from each group in the bipartite graph. This problem is NP-hard, so there is no linear maximization oracle, and therefore, we consider a small instance where $\sqrt{d} = 8$ and $\sqrt{s} = 2$. We pick two random non-overlapping

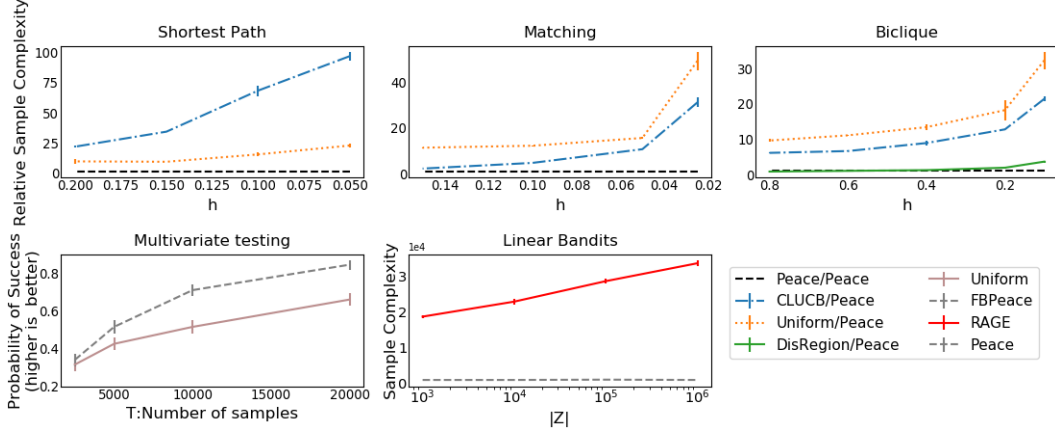

Figure 1: In row 1, panels (i) and (ii) depict the relative performance of CLUCB and UA to PEACE, and panel (iii) depicts the relative performance of CLUCB, DisRegion, and UA to PEACE. In row 2, Panel (i) compares uniform sampling and FBPeace in the fixed budget setting, and panel (ii) compares the performance of RAGE to Peace on the linear bandits experiment.

bicliques and let $\mathcal{B}_1$ and $\mathcal{B}_2$ denote the set of their respective edges. If $e \in \mathcal{B}_1$, we set $\theta_e = 1$, and if $e \in \mathcal{B}_2$, we set $\theta_e = 1 - h$ for $h \in \{.1, .8, .6, .4, .2\}$. Otherwise, we set $\theta_e = 0$.

As discussed in the related work, all of the algorithms in the literature are either inefficient or have burdensome running times, with the sole exception being CLUCB from [10]. Therefore, for the shortest path and matching experiments, we compare Algorithm 2 against a uniform allocation strategy (UA) and CLUCB. The biclique instance is small enough that $\mathcal{Z}$ can be enumerated, so we also compare against Algorithm 4 from [5] (denoted DisRegion), which achieves the best sample complexity result from that paper.

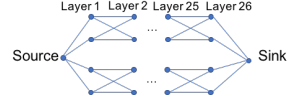

Figure 2: Shortest Path Problem

The first row of panels in Figure 1 depicts the ratio of the average performance of the competing algorithms to the average performance of our algorithm. In the matching experiment, as the gap between the best matching $M_1$ and the second best matching $M_2$ get smaller, CLUCB pays a cost of roughly $|U|/h^2$ to distinguish $M_1$ from $M_2$ whereas our algorithm pays a cost of roughly $1/h^2$. A similar phenomenon occurs in the shortest path problem. In the biclique experiment, as the gap between the best biclique and the second best biclique decreases, the performance of the competing algorithms degrades relative to Peace. For example, for large $h$, Peace and DisRegion have similar performace but for $h = .2$, DisRegion requires more than 3 times as many samples as Peace.

**Multivariate Testing** We consider multivariate testing [16, 15] in which there are $d$ options, each having $k$ possible levels. For example, consider determining the optimal content for a display-ad with slots such as headline, body, etc. and each slot has several variations. A layout is specified by a $d$-tuple $f = (f_1, \cdots, f_d) \in \{1, \cdots, k\}^d$ indicating the level chosen for each option. For each option $I$, $1 \leq I \leq d$ and level $f$, $1 \leq f \leq k$, there is a weight $W_f^I \in \mathbb{R}$, and for each pair of options $I, J$ and factors $f_I, f_J$, there is a weight $W_{f_I, f_J}^{I,J} \in \mathbb{R}$ capturing linear and quadratic interaction terms respectively. The total reward of a layout $f = (f_1, \cdots, f_d)$ is given by $W_0 + \sum_{I=1}^d W_{f_I}^I + \sum_{I=1}^d \sum_{J=1}^d W_{f_I, f_J}^{I,J}$. The fixed budget experiment in Figure 1 considers a scenario when $k = 6$ and $d = 3$ and compares Algorithm 3 (FBPeace) to uniform sampling. We set $W_{1,1}^{1,2} = .8$ and $W_{1,1}^{2,3} = .1$ and all other weights to zero, capturing a setting where the three options must be synchronized. At 10000 samples, FBPeace is 30% more likely to return the true optimal layout.

**Linear Bandits.** We considered a setting in $\mathbb{R}^2$, where $\mathcal{X} = \{\mathbf{e}_1, \cos(3\pi/4)\mathbf{e}_1 + \sin(3\pi/4)\mathbf{e}_2\}$ and $\mathcal{Z} = \{\cos(\pi/4 + \phi_i)\mathbf{e}_1 + \sin(\pi/4 + \phi_i)\mathbf{e}_2\}_{i=1}^n$ where $\phi_i \sim \text{Uniform}([0, .05])$. The parameter vector is fixed at $\theta = \mathbf{e}_1$. In Figure 1 we see that as the number of arms increases (from $10^3$ to $10^6$), the number of samples by our algorithms is constant, yet grows linearly in $\log(|\mathcal{Z}|)$ for RAGE [12]. This reflects the main goal of the paper - optimal union bounding for large classes.

## Broader Impact

In this paper, we developed adaptive learning algorithms for linear and combinatorial settings. These algorithms hold the promise of decreasing the amount of data that is required to make discoveries. Given the generic nature of these algorithms, it is possible that practitioners will apply these algorithms towards goals that are ultimately harmful for society. However, we believe that our algorithms also hold significant promise to benefit society. By making the learning process more data-efficient, we are optimistic that our algorithms can be applied to accelerate drug discovery, as well as the rate of scientific discovery in a wide range of fields ranging from biology to the social sciences. Our belief is that the potential benefits outweigh the potential negative consequences.

## Acknowledgement and Disclosure of Funding

This work was supported in part by NSF IIS-1907907 and an Amazon Research Award.

## Footnotes

[1]Our results still apply in the case where the noise is sub-Gaussian, but for simplicity here we assume that the noise is Gaussian (see the Supplementary Material).

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
