[Supplementary Material]

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

# A   Outline and Notation

Section B gives the proof of Theorem 3. Section C presents proofs of the two results for the fixed confidence setting. Section D proves provides the main results on the computational efficiency of Algorithm 2. Section E provides the proof of our upper bound for the fixed budget setting. Section F proves various results related to $\gamma^*$. Section G gives additional lower bounds for the transductive linear bandit problem. Section H provides a discussion of rounding. Section I presents technical lemmas. Section J discusses the convexity of $\gamma^*$. Section K discusses the sample complexity results of other papers. Section M gives further details on the experiments.

For the combinatorial bandit setting, we assume wlog that for all $i \in [d]$ there exist $z, z' \in \mathcal{Z}$ such that $i \in z$ and $i \notin z'$. We will sometimes write $z \cap z'$ to denote $(z_1 \cdot z_1', \ldots, z_d \cdot z_d')^\top$. In a similar way, we will use $z \Delta z'$ to denote the symmetric difference of $z$ and $z'$, viewed as sets. We use $c, c', \cdots$ to denote positive universal constants whose values may change from line to line.

# B   Proof of Theorem 3

*Proof of Theorem 3.* For simplicity, we suppose that $\mathcal{Z}$ is finite; the extension is straightforward by taking an $\epsilon$ of room. Define $\Delta_z = \theta^\top(z_* - z)$. Let $\mathcal{X} = \{x_1, \ldots, x_m\}$. Fix $\{x_{I_1}, \ldots, x_{I_T}\} \subset \mathcal{X}$ to be the measurement vectors pulled by the algorithm. Define the matrix

$$
X = \begin{pmatrix} x_{I_1}^\top \\ \vdots \\ x_{I_T}^\top \end{pmatrix}
$$

Define $\widehat{\theta} = (X^\top X)^{-1} X^\top Y$.

Let $\lambda \in \boldsymbol{\Delta}$ be the associated allocation: $\lambda_i = \frac{1}{T} \sum_{s=1}^T \mathbb{1}\{I_s = i\}$. Note that

$$
\mathbb{E}_{\eta \sim N(0,I)} \Big[ \sup_{z \in \mathcal{Z} \setminus \{z_*\}} \frac{(z_* - z)^\top (X^\top X)^{-1/2} \eta}{\Delta_z} \Big] = \frac{1}{\sqrt{T}} \mathbb{E}_{\eta \sim N(0,I)} \Big[ \sup_{z \in \mathcal{Z} \setminus \{z_*\}} \frac{(z_* - z)^\top A(\lambda)^{-1/2} \eta}{\Delta_z} \Big]
$$

and

$$
\sup_{z \in \mathcal{Z} \setminus \{z_*\}} \frac{\|z_* - z\|_{(X^\top X)^{-1}}}{\Delta_z} = \frac{1}{\sqrt{T}} \sup_{z \in \mathcal{Z} \setminus \{z_*\}} \frac{\|z_* - z\|_{A(\lambda)^{-1}}}{\Delta_z}.
$$

Recall

$$
\rho^*(\lambda) := \sup_{z \in \mathcal{Z} \setminus \{z_*\}} \frac{\|z_* - z\|_{A(\lambda)^{-1}}^2}{\Delta_z^2}.
$$

$$
\gamma^*(\lambda) := \mathbb{E}_{\eta \sim N(0,I)} \Big[ \sup_{z \in \mathcal{Z} \setminus \{z_*\}} \frac{(z_* - z)^\top A(\lambda)^{-1/2} \eta}{\Delta_z} \Big]^2.
$$

**Case 1:** $T \leq \frac{1}{2} \rho^*(\lambda) \log(1/\delta)$. First, suppose that $T \leq \frac{1}{2} \log(1/\delta) \rho^*(\lambda)$. By definition of $\rho^*(\lambda)$, there exists $\bar{z} \in \mathcal{Z} \setminus \{z_*\}$ such that

$$
\frac{\|z_* - \bar{z}\|_{A(\lambda)^{-1}}^2}{\Delta_{\bar{z}}^2} = \rho^*(\lambda).
$$

Note that

$$
\frac{(\bar{z} - z_*)^\top (\widehat{\theta} - \theta)}{\Delta_{\bar{z}}} = \frac{(\bar{z} - z_*)^\top A(T\lambda)^{-1/2} \eta}{\Delta_{\bar{z}}} \sim N\Big(0, \frac{\|z_* - \bar{z}\|_{A(T\lambda)^{-1}}^2}{\Delta_{\bar{z}}^2}\Big) \tag{6}
$$

and by assumption

$$
\mathbb{V}\Big(\frac{(\bar{z} - z_*)^\top (\widehat{\theta} - \theta)}{\Delta_{\bar{z}}}\Big) \geq \frac{\rho^*(\lambda)}{T} \geq \frac{2}{\log(1/\delta)}. \tag{7}
$$

By Proposition 2.1.2 of [32], we have that if $g \sim N(0, \sigma^2)$, then

$$\mathbb{P}(g/\sigma \geq t) \geq (\frac{1}{t} - \frac{1}{t^3})\frac{1}{\sqrt{2\pi}}e^{-t^2/2}.$$

Let $\bar{g} \sim N(0, \frac{2}{\log(1/\delta)})$. Therefore, using (7),

$$\mathbb{P}(\frac{(\bar{z} - z_*)^\top(\widehat{\theta} - \theta)}{\Delta_{\bar{z}}} > \sqrt{2}) \geq \mathbb{P}(\bar{g} > \sqrt{2})$$

$$\geq (\frac{1}{\sqrt{\log(1/\delta)}} - \frac{1}{\sqrt{\log(1/\delta)}^3})\frac{1}{\sqrt{2\pi}}\delta^{1/2}$$

$$\geq \delta.$$

where the last inequality follows since $\delta \in (0, 0.015]$ and inspecting the graph of the functions. Thus, with probability at least $\delta$, we have

$$\frac{(\bar{z} - z_*)^\top(\widehat{\theta} - \theta)}{\Delta_{\bar{z}}} > \sqrt{2},$$

which implies that

$$(\bar{z} - z_*)^\top\widehat{\theta} > 0,$$

in which case the algorithm makes a mistake. Thus, we may suppose for the remainder of the proof that $T > \frac{1}{2}\log(1/\delta)\rho^*(\lambda)$.

**Case 2:** $T > \frac{1}{2}\log(1/\delta)\rho^*(\lambda)$. Next, suppose

$$T \leq \frac{1}{4}\mathbb{E}_{\eta \sim N(0,I)}[\sup_{z \in \mathcal{Z}\setminus\{z_*\}} \frac{(z_* - z)^\top A(\lambda)^{-1/2}\eta}{\theta^\top(z_* - z)}]^2. \tag{8}$$

Note that $\widehat{\theta} \sim N(\theta, (X^\top X)^{-1})$ so that

$$\mathbb{E} \sup_{z \in \mathcal{Z}\setminus\{z_*\}} \frac{(z - z_*)^\top(\widehat{\theta} - \theta)}{\Delta_z} = \mathbb{E}_{\eta \sim N(0,I)}[\sup_{z \in \mathcal{Z}\setminus\{z_*\}} \frac{(z - z_*)^\top(X^\top X)^{-1/2}\eta}{\Delta_z}]$$

$$= \mathbb{E}_{\eta \sim N(0,I)}[\sup_{z \in \mathcal{Z}\setminus\{z_*\}} \frac{(z_* - z)^\top(X^\top X)^{-1/2}\eta}{\Delta_z}]$$

where we used the fact that $(z_* - z)^\top(X^\top X)^{-1/2}\eta$ and $(z - z_*)^\top(X^\top X)^{-1/2}\eta$ are equal in distribution.

By Theorem 5.8 in [2], with probability at least $1 - e^{-1/2}$,

$$\mathbb{E}_{\eta \sim N(0,I)}[\sup_{z \in \mathcal{Z}\setminus\{z_*\}} \frac{(z_* - z)^\top(X^\top X)^{-1/2}\eta}{\Delta_z}] - \sup_{z \in \mathcal{Z}\setminus\{z_*\}} \frac{(z - z_*)^\top(\widehat{\theta} - \theta)}{\Delta_z}$$

$$\leq \sup_{z \in \mathcal{Z}\setminus\{z_*\}} \frac{\|z_* - z\|_{(X^\top X)^{-1}}}{\Delta_z}$$

Towards a contradiction, suppose that inequality (8) does not hold. Then, with probability at least $1 - e^{-1/2}$ we have

$$\sup_{z \in \mathcal{Z}} \frac{(z - z_*)^\top(\widehat{\theta} - \theta)}{\Delta_z}$$

$$\geq \mathbb{E}_{\eta \sim N(0,I)}[\sup_{z \in \mathcal{Z}\setminus\{z_*\}} \frac{(z_* - z)^\top(X^\top X)^{-1/2}\eta}{\Delta_z}] - \sup_{z \in \mathcal{Z}\setminus\{z_*\}} \frac{\|z_* - z\|_{(X^\top X)^{-1}}}{\Delta_z}$$

$$= \frac{1}{\sqrt{T}}\mathbb{E}_{\eta \sim N(0,I)}[\sup_{z \in \mathcal{Z}\setminus\{z_*\}} \frac{(z_* - z)^\top A(\lambda)^{-1/2}\eta}{\Delta_z}] - \frac{1}{\sqrt{T}}\sup_{z \in \mathcal{Z}\setminus\{z_*\}} \frac{\|z_* - z\|_{A(\lambda)^{-1}}}{\Delta_z}$$

$$\geq \frac{1}{\sqrt{T}}\mathbb{E}_{\eta \sim N(0,I)}[\sup_{z \in \mathcal{Z}\setminus\{z_*\}} \frac{(z_* - z)^\top A(\lambda)^{-1/2}\eta}{\Delta_z}] - 1 \tag{9}$$

$$> 1 \tag{10}$$

where inequality (9) follows from $T > \frac{1}{2}\log(1/\delta)\rho^*(\lambda) \geq \rho^*(\lambda)$ since $\delta \in (0, 0.015]$ and inequality (10) follows from the inequality (8). Rearranging the above inequality, if (8) holds, then there exists a $z \in \mathcal{Z} \setminus \{z_*\}$ such that

$$(z - z_*)^\top \widehat{\theta} > 0.$$

Combining the two cases imply that

$$T \geq c[\rho^*(\lambda) + \gamma^*(\lambda)] \geq c[\rho^* + \gamma^*].$$

$\square$

## C  Fixed Confidence Upper Bound Proofs

### C.1  Peace Algorithm Proofs

*Proof of Theorem 4.* **Step 1: Define a good event.** Define $\delta_k = \frac{\delta}{k^2}$. Let $x_1, \ldots, x_{N_k}$ denote the pulled measurement vectors in round $k$. By Theorem 5.8 in [2], with probability at least $1 - \frac{\delta}{k^2}$

$$\sup_{z,z' \in \mathcal{Z}_k} |(z - z')^\top (\widehat{\theta}_k - \theta)|$$

$$\leq \mathbb{E}[\sup_{z,z' \in \mathcal{Z}_k} (z - z')^\top (\widehat{\theta}_k - \theta)] + \sqrt{2\log(2k^2/\delta)\max_{z,z' \in \mathcal{Z}_k} \|z - z'\|^2_{(\sum_{i=1}^{N_k} x_i x_i^\top)^{-1}}}$$

$$= \mathbb{E}_{\eta \sim N(0,I)}[\sup_{z,z' \in \mathcal{Z}_k} (z - z')^\top (\sum_{i=1}^{N_k} x_i x_i^\top)^{-1/2}\eta] + \sqrt{2\log(2k^2/\delta)\max_{z,z' \in \mathcal{Z}_k} \|z - z'\|^2_{(\sum_{i=1}^{N_k} x_i x_i^\top)^{-1}}}$$

$$\leq \sqrt{\frac{(1+\epsilon)}{N_k}}\left(\mathbb{E}_{\eta \sim N(0,I)}[\sup_{z,z' \in \mathcal{Z}_k} (z - z')^\top A(\lambda_k)^{-1/2}\eta]\right.$$

$$\left. + \sqrt{2\log(2k^2/\delta)\max_{z,z' \in \mathcal{Z}_k} \|z - z'\|^2_{A(\lambda_k)^{-1}}}\right) \tag{11}$$

$$\leq \sqrt{\frac{2(1+\epsilon)\tau_k}{N_k}} \tag{12}$$

where inequality (11) follows by the guarantee on the the rounding subroutine ROUND and Lemma 11, and the line (12) uses $\sqrt{a} + \sqrt{b} \leq \sqrt{2a + 2b}$ and the definition of $\tau_k$. Define the events

$$\mathcal{E}_k = \{\sup_{z,z' \in \mathcal{Z}_k} |(z - z')^\top (\widehat{\theta}_k - \theta)| \leq \sqrt{\frac{2(1+\epsilon)\tau_k}{N_k}}\}$$

$$\mathcal{E} = \cap_{k=1}^\infty \mathcal{E}_k.$$

Note that line (12) implies that $\mathbb{P}(\mathcal{E}_k) \geq 1 - \frac{\delta}{k^2}$. Thus, we have

$$\mathbb{P}(\mathcal{E}) = \prod_{k=1}^\infty \mathbb{P}(\mathcal{E}_k | \cap_{l=1}^{k-1} \mathcal{E}_l) \geq \prod_{k=1}^\infty (1 - \frac{\delta}{k^2}) = \frac{\sin(\pi\delta)}{\pi\delta} \geq 1 - \delta$$

where the last line used $\delta \in (0, 1)$. We suppose $\mathcal{E}$ holds for the remainder of the proof.

**Step 2: Correctness.** Define $S_k := \{z \in \mathcal{Z} : \theta^\top (z_* - z) \leq B2^{-k}\}$. We show that $z_* \in \mathcal{Z}_k$ and $\mathcal{Z}_k \subset S_{k-1}$ for $k = 2, 3 \ldots$. Using the event $\mathcal{E}$, we have that

$$\sup_{z,z' \in \mathcal{Z}_1} |(z - z')^\top (\widehat{\theta}_1 - \theta)| \leq \sqrt{\frac{2(1+\epsilon)\tau_1}{N_1}} \leq \frac{B}{4}$$

where we used $N_k \geq 2\tau_k(\frac{2^{k+1}}{B})^2(1+\epsilon)$. First, fix any $z \notin S_1$. We will then show that $z \notin \mathcal{Z}_2$. By definition, $\theta^\top(z_* - z) \geq \frac{B}{2}$. Note that

$$
\begin{aligned}
(z_* - z)^\top \widehat{\theta}_1 - B2^{-2} &= (z_* - z)^\top(\widehat{\theta}_1 - \theta) + \theta^\top(z_* - z) - B2^{-2} \\
&\geq (z_* - z)^\top(\widehat{\theta}_1 - \theta) + \frac{B}{4} \\
&\geq -\frac{B}{4} + \frac{B}{4} \\
&\geq 0
\end{aligned}
$$

where we applied the assumption that $z \notin S_1$ and the event. Thus, by the elimination rule, $z \notin \mathcal{Z}_2$.

Now, we show that $z_* \in \mathcal{Z}_1$. Let $z \in \mathcal{Z}_1$. Then, using the event we have that

$$
\begin{aligned}
(z - z_*)^\top \widehat{\theta}_1 - B2^{-2} &= (z - z_*)^\top(\widehat{\theta}_1 - \theta) + \theta^\top(z_* - z) - B2^{-2} \\
&< (z - z_*)^\top(\widehat{\theta}_1 - \theta) - B2^{-2} \\
&\leq \frac{B}{4} - \frac{B}{4} \\
&= 0.
\end{aligned}
$$

This proves the base case.

Next, we prove the inductive step. Suppose that $\mathcal{Z}_{k-1} \subset S_{k-2}$; we show that $\mathcal{Z}_k \subset S_{k-1}$. For any $z, z' \in \mathcal{Z}_{k-1}$,

$$
\begin{aligned}
|(z - z')^\top(\widehat{\theta}_{k-1} - \theta)| &\leq \sqrt{\frac{2(1+\epsilon)\tau_k}{N_k}} \\
&\leq B2^{-(k+1)}.
\end{aligned}
$$

Let $z \in S_{k-1}^c$ so that $\theta^\top(z_* - z) > B2^{-k+1}$. Then,

$$
\begin{aligned}
(z_* - z)^\top \widehat{\theta}_{k-1} - B2^{-(k+1)} &= (z_* - z)^\top(\widehat{\theta}_{k-1} - \theta) + (z_* - z)^\top\theta - B2^{-(k+1)} \\
&\geq (z_* - z)^\top(\widehat{\theta}_{k-1} - \theta) + B2^{-(k+1)} \\
&\geq -B2^{-(k+1)} + B2^{-(k+1)} \\
&= 0
\end{aligned}
$$

Thus, $z \notin \mathcal{Z}_k$, proving one part of the inductive step.

Next, we show $z_* \in \mathcal{Z}_k$. By the inductive hypothesis, $z_* \in \mathcal{Z}_{k-1}$. Let $z \in \mathcal{Z}_{k-1}$. Then,

$$
\begin{aligned}
(z - z_*)^\top \widehat{\theta}_{k-1} - B2^{-(k+1)} &= (z - z_*)^\top(\widehat{\theta}_{k-1} - \theta) + (z - z_*)^\top\theta - B2^{-(k+1)} \\
&< (z - z_*)^\top(\widehat{\theta}_{k-1} - \theta) - B2^{-(k+1)} \\
&\leq B2^{-(k+1)} - B2^{-(k+1)} \\
&= 0
\end{aligned}
$$

**Step 3: Upper bounding the sample complexity.** Now, we bound the number of samples taken until the algorithm terminates. Since $\mathcal{Z}_k \subset S_{k-1}$ for $k = 2, 3 \ldots$ as we showed in the previous step, once $k \geq c\log(B/\Delta_{min})$, we have that $\mathcal{Z}_k = \{z_*\}$ and thus there are at most $c\log(B/\Delta_{min})$ rounds. In round $k$, the algorithm takes $N_k = \left\lceil 2\tau_k(\frac{2^{k+1}}{B})^2(1+\epsilon)\right\rceil \vee q(\epsilon)$ samples and, thus, the sample complexity is bounded by the following sum

$$
\sum_{k=1}^{c\log(B/\Delta_{min})} N_k \leq c'[\log(B/\Delta_{min})d + \sum_{k=1}^{c\log(B/\Delta_{min})} \tau_k(\frac{2^k}{B})^2] \tag{13}
$$

where we used $q(\epsilon) = O(d)$ by the guarantees on the rounding procedure and $\epsilon = 1/10$. Now, we focus on upper bounding the second term in the above expression. For $k = 1$, then

$$
\tau_1(\frac{2^1}{B})^2 \leq \frac{c}{B} \leq c' \tag{14}
$$

where we used the relation $B = \tau_1 \vee 1$.

Next, we bound the terms $k > 1$. Note that

$$\tau_k(\frac{2^k}{B})^2 = \mathbb{E}_{\eta \sim N(0,I)}[\max_{z,z' \in \mathcal{Z}_k}(z - z')^\top A(\lambda)^{-1/2}\eta]^2(\frac{2^k}{B})^2$$

$$+ 2\log(\frac{1}{\delta_k})\max_{z,z' \in \mathcal{Z}_k} \|z - z'\|^2_{A(\lambda)^{-1}}(\frac{2^k}{B})^2$$

We begin by bounding the second term. Fix $\lambda$. Then,

$$\max_{z,z' \in \mathcal{Z}_k} \|z - z'\|^2_{A(\lambda)^{-1}}(\frac{2^k}{B})^2 \le \max_{z,z' \in S_k} \|z - z'\|^2_{A(\lambda)^{-1}}(\frac{2^k}{B})^2 \tag{15}$$

$$\le c\max_{z \in S_k \setminus \{z_*\}} \|z_* - z\|^2_{A(\lambda)^{-1}}(\frac{2^k}{B})^2 \tag{16}$$

$$\le c\max_{z \in \mathcal{Z} \setminus \{z_*\}} \frac{\|z_* - z\|^2_{A(\lambda)^{-1}}}{\theta^\top(z_* - z)^2} \tag{17}$$

where line (15) follows since $\mathcal{Z}_k \subset S_{k-1}$ for $k = 2, 3, \ldots$, line (16) follows since the triangle inequality implies $\max_{z,z' \in S_k} \|z - z'\|^2_{A(\lambda)^{-1}} \le c\max_{z \in S_k \setminus \{z_*\}} \|z_* - z\|^2_{A(\lambda)^{-1}}$, and line (17) follows since for all $z \in S_k \setminus \{z_*\}$, $\Delta_z \le 2^{-k}B$ by definition. Next, we bound the first term:

$$\mathbb{E}_{\eta \sim N(0,I)}[\max_{z,z' \in \mathcal{Z}_k} \frac{(z - z')^\top A(\lambda)^{-1/2}\eta}{2^{-k}B}]^2$$

$$= 4\mathbb{E}_{\eta \sim N(0,I)}[\max_{z \in \mathcal{Z}_k} \frac{(z_* - z)^\top A(\lambda)^{-1/2}\eta}{2^{-k}B}]^2$$

$$\le 4\mathbb{E}_{\eta \sim N(0,I)}[\max_{z \in S_k} \frac{(z_* - z)^\top A(\lambda)^{-1/2}\eta}{2^{-k}B}]^2 \tag{18}$$

$$\le 4\mathbb{E}_{\eta \sim N(0,I)}[\max(\max_{z \in S_k \setminus \{z_*\}} \frac{(z_* - z)^\top A(\lambda)^{-1/2}\eta}{\theta^\top(z_* - z)}, 0)]^2 \tag{19}$$

$$\le 8\big[\mathbb{E}_{\eta \sim N(0,I)}[\max_{z \in S_k \setminus \{z_*\}} \frac{(z_* - z)^\top A(\lambda)^{-1/2}\eta}{\theta^\top(z_* - z)}]^2$$

$$+ \max_{z \in S_k \setminus \{z_*\}} \frac{\|z_* - z\|^2_{A(\lambda)^{-1}}}{\theta^\top(z_* - z)^2}\big] \tag{20}$$

where line (18) follows by $\mathcal{Z}_k \subset S_{k-1}$, line (19) follows by Lemma 14, for all $z \in S_k \setminus \{z_*\}$, $\Delta_z \le 2^{-k}B$, and $z_* \in S_k$, and line (20) follows by Lemma 16. Thus, combining (17) and (20), and taking the infimum over $\lambda$, we obtain

$$\tau_k(\frac{2^k}{B})^2 \le c[\inf_{\lambda} \mathbb{E}_{\eta \sim N(0,I)}[\max_{z \in \mathcal{Z} \setminus \{z_*\}} \frac{(z_* - z)^\top A(\lambda)^{-1/2}\eta}{\theta^\top(z_* - z)}]^2$$

$$+ \max_{z \in \mathcal{Z} \setminus \{z_*\}} \frac{\|z_* - z\|^2_{A(\lambda)^{-1}}}{\theta^\top(z_* - z)^2}\log(k^2/\delta)]$$

$$\le c'[\gamma^* + \rho^*\log(k^2/\delta)] \tag{21}$$

where line (21) follows by Lemma 13. Thus, combining (13), (14), and (21), we obtain

$$\sum_{k=1}^{c\log(B/\Delta_{min})} N_k \le c\log(B/\Delta_{min})[d + \gamma^* + \rho^*\log(\log(B/\Delta_{min})/\delta)]. \tag{22}$$

Next, we will prove

$$\sum_{k=1}^{c\log(B/\Delta_{min})} N_k \le c\log(B/\Delta_{min})d + \log(\frac{B}{\min_{k:|S_k|>1} F_k})[\gamma^* + \rho^*\log(\log(B/\Delta_{min})/\delta)].$$

$$\tag{23}$$

where

$$F_k := \begin{cases} \inf_\lambda \max_{z,z' \in S_k} \|z - z'\|^2_{A(\lambda)^{-1}} \log(\frac{2k^2}{\delta}) + \mathbb{E}_\eta[\max_{z \in S_k}(z - z')^\top A(\lambda)^{-1/2}\eta]^2 & k \geq 1 \\ B & k = 0 \end{cases}$$

(22) and (23) together would imply the result. By a similar argument used to establish (22), it suffices to prove

$$\sum_{k=2}^{c\log(B/\Delta_{min})} \tau_k(\frac{2^k}{B})^2 \leq \log(\frac{B}{\min_{k:|S_k|>1} F_k})[\gamma^* + \rho^* \log(\log(B/\Delta_{min})/\delta)]$$

Let $L$ be the largest integer such that $|S_L| > 1$. Define

$$H_i = \{k \in [L] : F_k \in (\frac{F_0}{2^{-(i+1)}}, \frac{F_0}{2^{-i}}]\}.$$

and define

$$k_i = \max(k : k \in H_i)$$

for $i \in \lceil \log_2(F_0/F_L) \rceil$. Then, the sample complexity is upper bounded by

$$\sum_{k=2}^{c\log(B/\Delta_{min})} \tau_k(\frac{2^k}{B})^2 \leq \sum_{k=2}^{c\log_2(B/\Delta_{min})} F_k(\frac{2^k}{B})^2 \tag{24}$$

$$= c \sum_{i=1}^{\lceil \log_2(F_0/F_L) \rceil} \sum_{k \in H_i} F_k(\frac{2^k}{B})^2$$

$$\leq c' \sum_{i=1}^{\lceil \log_2(F_0/F_L) \rceil} \max_{k \in H_i} F_k \sum_{k \in H_i}(\frac{2^k}{B})^2$$

$$\leq c'' \sum_{i=1}^{\lceil \log_2(F_0/F_L) \rceil} \max_{k \in H_i} F_k(\frac{2^{k_i}}{B})^2 \tag{25}$$

$$\leq c''' \sum_{i=1}^{\lceil \log_2(F_0/F_L) \rceil} F_{k_i}(\frac{2^{k_i}}{B})^2$$

$$\leq c'''' \lceil \log_2(F_0/F_L) \rceil \Big[ \inf_\lambda \mathbb{E}_{\eta \sim N(0,I)}[\max_{z \in \mathcal{Z}\backslash} \frac{(z_* - z)^\top A(\lambda)^{-1/2}\eta}{\theta^\top(z_* - z)}]^2$$

$$+ \max_{z \in \mathcal{Z}\backslash} \frac{\|z_* - z\|^2_{A(\lambda)^{-1}}}{\theta^\top(z_* - z)^2} \Big] \log(\log(B/\Delta_{min})/\delta) \Big] \tag{26}$$

where line (24) follows since $\mathcal{Z}_k \subset S_{k-1}$, line (25) follows since $\sum_{l=1}^m (2^l)^2 \leq c2^{2m}$, and line (26) follows by (21).

$\square$

## C.2  Computationally Efficient Algorithm for Combinatorial Bandits Proofs

Before giving the proof of Theorem 6, we restate the algorithm with subroutines for solving the optimization problems approximately. Define $\mathbf{0} = (0, \ldots, 0)^\top$.

We briefly note that the optimization problem in (27) includes $\gamma(\mathcal{Z})$ as a special case by the following identity:

$$\mathbb{E}_{\eta \sim N(0,I)}[\max_{z,z' \in \mathcal{Z}}(z - z')^\top A(\lambda)^{-1/2}\eta]^2 = 4E_{\eta \sim N(0,I)}[\max_{z \in \mathcal{Z}} z^\top A(\lambda)^{-1/2}\eta]^2.$$

We also define the UNIQUE subroutine (Algorithm 5), originally provided in [8]. It finds the empirical best $\tilde{z}$ and the empirical second best $z'$ and determines whether enough samples have been collected to conclude that $\tilde{z}$ is the best. It uses at most $d$ calls to the linear maximization oracle.

**Algorithm 4:** Fixed Confidence Peace with a linear maximization oracle.

**Input:** $\mathcal{Z}$, estimate $\tilde{\theta}$, shift $b > 0$ ;

$\tilde{z} \longleftarrow \arg\max_{z \in \mathcal{Z}} \tilde{\theta}^\top z$;

**for** $i = 1, 2, \ldots, d$ s.t. $i \in \tilde{z}$ **do**

$$\tilde{\theta}^{(i)} = \begin{cases} \tilde{\theta}_j & j \neq i \\ -\infty & j = i \end{cases}$$

$\quad \tilde{z}^{(i)} \longleftarrow \arg\max_{z \in \mathcal{Z}} (\tilde{\theta}^{(i)})^\top z$;

$\quad$ **if** $\tilde{\theta}^\top(\tilde{z} - \tilde{z}^{(i)}) - b \leq 0$ **then return** False

**return** True

**Algorithm 5:** UNIQUE.

*Proof of Theorem 6.* We will first show that if we can solve the optimization problem

$$\mathbb{E}\max_z \frac{(z_0 - z)^\top A(\lambda)^{-1/2}\eta}{b + \theta_0^\top(z_0 - z)}.$$

for arbitrary $\theta_0 \in \mathbb{R}^d$, $z_0 \in \mathcal{Z}$, and $b > 0$, then the sample complexity claim follows. In particular, this implies solving the optimization problems $\gamma(\mathcal{Z})$ and (27). Then, we will show that solving it approximately using the subroutine ComputeAlloc only affects up to a constant factor and bound the number of oracle calls.

**Step 1: Good event holds with high probability.** Define the sets

$$S_k = \begin{cases} \{z \in \mathcal{Z} : \Delta_z \leq \Gamma 2^{-k}\} & k \geq 1 \\ \mathcal{Z} & k = 0 \end{cases}$$

and define $\delta_k = \frac{\delta}{2k^3}$. Define the events for all $j \in [k]$

$$\Sigma_{k,j} = \{ \sup_{z,z' \in S_j} |(z - z')^\top(\widehat{\theta}_k - \theta)| \leq$$

$$\sqrt{2(1+\epsilon)(1 + \pi\log(1/\delta_k))\frac{\mathbb{E}[\sup_{z,z' \in S_j}(z - z')^\top A(\lambda_k)^{-1/2}\eta]^2}{N_k}}\}$$

$$\Sigma_k = \cap_{j=0}^k \Sigma_{k,j}$$

$$\Sigma = \cap_{k=1}^{\log(\Gamma/\Delta_{min})} \cap_{j=0}^k \Sigma_{k,j}$$

Let $x_1, \ldots, x_{N_k}$ denote the measurement vectors selected in round $k$. Theorem 5.8 from [2] implies that with probability at least $1 - \frac{\delta}{k^3}$

$$
\sup_{z,z' \in S_j} |(z - z')^\top (\widehat{\theta}_{k+1} - \theta)|
$$

$$
\leq \mathbb{E} \sup_{z,z' \in S_j} (z - z')^\top (\widehat{\theta}_k - \theta) + \sqrt{2 \log(1/\delta_k) \max_{z,z' \in S_j} \|z - z'\|^2_{(\sum_{i=1}^{N_k} x_i x_i^\top)^{-1}}}
$$

$$
= \mathbb{E} \sup_{z,z' \in S_j} (z - z')^\top (\sum_{i=1}^{N_k} x_i x_i^\top)^{-1/2} \eta + \sqrt{2 \log(1/\delta_k) \max_{z,z' \in S_j} \|z - z'\|^2_{(\sum_{i=1}^{N_k} x_i x_i^\top)^{-1}}}
$$

$$
\leq \mathbb{E} \sup_{z,z' \in S_j} (z - z')^\top (\sum_{i=1}^{N_k} x_i x_i^\top)^{-1/2} \eta
$$

$$
+ \sqrt{\pi \log(1/\delta_k) \mathbb{E}[\sup_{z,z' \in S_j} (z - z')^\top (\sum_{i=1}^{N_k} x_i x_i^\top)^{-1/2} \eta]^2} \tag{28}
$$

$$
\leq \sqrt{2(1 + \pi \log(1/\delta_k)) \mathbb{E}[\sup_{z,z' \in S_j} (z - z')^\top (\sum_{i=1}^{N_k} x_i x_i^\top)^{-1/2} \eta]^2} \tag{29}
$$

$$
\leq \sqrt{2(1 + \epsilon)(1 + \pi \log(1/\delta_k)) \frac{\mathbb{E}[\sup_{z,z' \in S_j} (z - z')^\top A(\lambda_k)^{-1/2} \eta]^2}{N_k}} \tag{30}
$$

where line (28) follows by Lemma 12, line (29) follows by $\sqrt{a} + \sqrt{b} \leq \sqrt{2(a + b)}$, and line (30) follows by Lemma 11. Therefore, $\mathbb{P}(\Sigma^c_{k,j}) \leq \frac{\delta}{k^3}$. By law of total probability,

$$
\mathbb{P}(\Sigma^c) \leq \sum_{k=1}^{\infty} \sum_{j=0}^{k} \mathbb{P}(\Sigma^c_{k,j} | \cap_{l=1}^{k-1} \Sigma_l) \leq \sum_{k=1}^{\infty} (k+1) \frac{\delta}{k^3} \leq 3\delta.
$$

We suppose the event $\Sigma$ holds for the rest of the proof.

**Step 2: gaps are well estimated every round** $k$ Now, we show that the following hold: at every round $k \geq 1$,

1. if $z \in S^c_k$,

$$
|(z_* - z)^\top (\widehat{\theta}_k - \theta)| \leq \frac{\Delta_z}{8}
$$

2. if $z \in S_k$,

$$
|(z_* - z)^\top (\widehat{\theta}_k - \theta)| \leq \frac{2^{-k}\Gamma}{8}.
$$

We proceed inductively. First, we prove the base case $k = 1$. On the event $\Sigma_{1,1}$, we have using the definition of $N_1$, for all $z \in \mathcal{Z}$,

$$
|(z_* - z)^\top (\widehat{\theta}_1 - \theta)| \leq \sup_{z,z' \in \mathcal{Z}} |(z - z')^\top (\widehat{\theta}_1 - \theta)|
$$

$$
\leq \sqrt{\frac{2(1 + \epsilon)(1 + \pi \log(1/\delta_k)) \mathbb{E}[\sup_{z,z' \in \mathcal{Z}} (z - z')^\top A(\lambda)^{-1/2} \eta]^2}{N_0}}
$$

$$
\leq \sqrt{\frac{8(1 + \epsilon)(1 + \pi \log(1/\delta_k)) \mathbb{E}[\sup_{z \in \mathcal{Z}} (\tilde{z}_0 - z)^\top A(\lambda)^{-1/2} \eta]^2}{N_0}}
$$

$$
\leq \frac{2^{-1}\Gamma}{8} \tag{31}
$$

where in the last line we used $N_k = \alpha \lceil \tau_k \log(1/\delta_k)(1 + \epsilon) \rceil \vee q(\epsilon)$. Observe that whether $z \in S_1$ or $z \in S_1^c$ the base case follows. Next, we show the inductive step. Suppose that at round $k \geq 1$, if $z \in S_k^c$,

$$|(z_* - z)^\top (\widehat{\theta}_k - \theta)| \leq \frac{\Delta_z}{8}$$

and if $z \in S_k$,

$$|(z_* - z)^\top (\widehat{\theta}_k - \theta)| \leq \frac{2^{-k}\Gamma}{8}.$$

Now, consider round $k + 1$. Fix $z_0 \in S_{k+1}^c$. If $\Delta_z \geq \frac{\Gamma}{2}$, there is nothing to show by (31). Thus, suppose $\Delta_z \leq \frac{\Gamma}{2}$. Then, there exists $j \leq k$ such that $\Gamma 2^{-(j+1)} \leq \Delta_{z_0} \leq \Gamma 2^{-j}$. Then,

$$\frac{|(z_* - z_0)^\top (\widehat{\theta}_{k+1} - \theta)|}{\Delta_{z_0}} \leq \sup_{z, z' \in S_j} |\frac{(z - z')^\top (\widehat{\theta}_{k+1} - \theta)}{\Delta_{z_0}}|$$

$$\leq \sqrt{2(1 + \epsilon)(1 + \pi \log(1/\delta_k)) \frac{\mathbb{E}[\sup_{z, z' \in S_j} \frac{(z-z')^\top A(\lambda)^{-1/2}\eta}{\Delta_{z_0}}]^2}{N_k}} \quad (32)$$

$$\leq \sqrt{8(1 + \epsilon)(1 + \pi \log(1/\delta_k)) \frac{\mathbb{E}[\sup_{z \in S_j} \frac{(\tilde{z}_k-z)^\top A(\lambda)^{-1/2}\eta}{\Delta_{z_0}}]^2}{N_k}}$$

$$\leq \sqrt{36(1 + \epsilon)(1 + \pi \log(1/\delta_k)) \frac{\mathbb{E}[\sup_{z \in S_j} \frac{(\tilde{z}_k-z)^\top A(\lambda)^{-1/2}\eta}{\Delta_z + 2^{-k}\Gamma}]^2}{N_k}} \quad (33)$$

$$\leq \sqrt{36(1 + \epsilon)(1 + \pi \log(1/\delta_k)) \frac{\mathbb{E}[\sup_{z \in \mathcal{Z}} \frac{(\tilde{z}_k-z)^\top A(\lambda)^{-1/2}\eta}{\Delta_z + 2^{-k}\Gamma}]^2}{N_k}} \quad (34)$$

$$\leq \sqrt{162(1 + \epsilon)(1 + \pi \log(1/\delta_k)) \frac{\mathbb{E}[\sup_{z \in \mathcal{Z}} \frac{(\tilde{z}_k-z)^\top A(\lambda)^{-1/2}\eta}{(\tilde{z}_k-z)^\top \widehat{\theta}_k + 2^{-k}\Gamma}]^2}{N_k}} \quad (35)$$

$$\leq \frac{1}{8} \quad (36)$$

where line (32) follows by the event $\Sigma$, line (33) follows from Lemma 14 since $\tilde{z}_k \in S_j$ and for all $z \in S_j$, $3\Delta_{z_0} \geq \Delta_z + 2^{-k}\Gamma$, (35) follows by the inductive hypothesis and Lemma 1, and (36) follows by the definition of $N_k$. Next, fix $z_0 \in S_{k+1}$; a similar series of inequalities shows that

$$|(z_* - z_0)^\top (\widehat{\theta}_{k+1} - \theta)| \leq \frac{2^{-(k+1)}\Gamma}{8},$$

yielding the claim.

**Step 3: Correctness.** To show correctness, it suffices to show that at round $k$, if $\tilde{z}_k \neq z_*$, then the UNIQUE$(\mathcal{Z}, \widehat{\theta}_k, 2^{-k}\Gamma)$ returns false. Inspection of the subroutine reveals that it suffices to show that $(\tilde{z}_k - z_*)^\top \widehat{\theta}_k - 2^{-k}\Gamma \leq 0$. By the claim in Step 2, we have that

$$(\tilde{z}_k - z_*)^\top \widehat{\theta}_k - 2^{-k}\Gamma = (\tilde{z}_k - z_*)^\top (\widehat{\theta}_k - \theta) - \Delta_{\tilde{z}_k} - 2^{-k}\Gamma$$

$$\leq \max(\frac{\Delta_{\tilde{z}_k}}{8}, \frac{2^{-k}\Gamma}{8}) - \Delta_{\tilde{z}_k} - 2^{-k}\Gamma$$

$$\leq 0$$

proving correctness.

**Step 4: Upper bound the sample complexity.** Note that at round $k$, UNIQUE$(\mathcal{Z}, \widehat{\theta}_k, 2^{-k}\Gamma)$ checks whether the gap between $\tilde{z}_k$ and $\arg\max_{z \neq \tilde{z}_k} \widehat{\theta}_k^\top z$ is at least $2^{-k}\Gamma$, and terminates if it is. Thus, by the claim in Step 2, the algorithm terminates and outputs $z_*$ once $k \geq c \log(\Gamma/\Delta_{\min})$. Thus, the

sample complexity is upper bounded by

$$\sum_{k=1}^{c\log(\Gamma/\Delta_{min})} N_k \leq c'[\log(\Gamma/\Delta_{min})d + \sum_{k=1}^{c\log(\Gamma/\Delta_{min})} \tau_k(\frac{2^k}{\Gamma})^2] \tag{37}$$

where we used $q(\epsilon) = O(d)$ by the guarantees on the rounding procedure and $\epsilon = 1/10$. Now, we focus on upper bounding the second term in the above expression. For $k = 1$, then

$$\tau_1(\frac{2^1}{\Gamma})^2 \leq \frac{c}{\Gamma} \leq c' \tag{38}$$

where we used the relation $\Gamma = \tau_1 \vee 1$. Thus, to obtain the upper bound on the sample complexity, it suffices to upper bound

$$\tau_k = \inf_{\lambda \in \mathbf{\Delta}} \mathbb{E}_{\eta \sim N(0,I)}[\max_{z \in \mathcal{Z}} \frac{(\tilde{z}_k - z)^\top A(\lambda)^{-1/2}\eta}{2^{-k}\Gamma + \widehat{\theta}_k^\top(\tilde{z}_k - z)}]^2$$

for $k > 1$. Fix $\lambda \in \mathbf{\Delta}$. We have that

$$\mathbb{E}_{\eta \sim N(0,I)}[\max_{z \in \mathcal{Z}} \frac{(\tilde{z}_k - z)^\top A(\lambda)^{-1/2}\eta}{2^{-k}\Gamma + \widehat{\theta}_k^\top(\tilde{z}_k - z)}]^2 \leq c\mathbb{E}_{\eta \sim N(0,I)}[\max_{z \in \mathcal{Z}} \frac{(\tilde{z}_k - z)^\top A(\lambda)^{-1/2}\eta}{2^{-k}\Gamma + \Delta_z}]^2$$

$$\leq c'[\mathbb{E}_{\eta \sim N(0,I)}[\max_{z \in \mathcal{Z}} \frac{(z_* - z)^\top A(\lambda)^{-1/2}\eta}{2^{-k}\Gamma + \Delta_z}]^2$$

$$+ \mathbb{E}_{\eta \sim N(0,I)}[\max_{z \in \mathcal{Z}} \frac{(z_* - \tilde{z}_k)^\top A(\lambda)^{-1/2}\eta}{2^{-k}\Gamma + \Delta_z}]^2]$$

We bound the first term as follows. Fix $z_0 \in \mathcal{Z} \setminus \{z_*\}$.

$$\mathbb{E}_{\eta \sim N(0,I)}[\max_{z \in \mathcal{Z}} \frac{(z_* - z)^\top A(\lambda)^{-1/2}\eta}{2^{-k}\Gamma + \Delta_z}]^2$$

$$= \mathbb{E}_{\eta \sim N(0,I)}[\max_{z \in \mathcal{Z} \setminus \{z_*\}} \max(\frac{(z_* - z)^\top A(\lambda)^{-1/2}\eta}{2^{-k}\Gamma + \Delta_z}, 0)]^2$$

$$\leq \mathbb{E}_{\eta \sim N(0,I)}[\max_{z \in \mathcal{Z} \setminus \{z_*\}} |\frac{(z_* - z)^\top A(\lambda)^{-1/2}\eta}{2^{-k}\Gamma + \Delta_z}|]^2$$

$$\leq 8\mathbb{E}_{\eta \sim N(0,I)}[\max_{z \in \mathcal{Z} \setminus \{z_*\}} \frac{(z_* - z)^\top A(\lambda)^{-1/2}\eta}{2^{-k}\Gamma + \Delta_z}]^2 + 8\frac{\|z_* - z_0\|^2_{A(\lambda)^{-1}}}{(2^{-k}\Gamma + \Delta_{z_0})^2} \tag{39}$$

$$\leq 8[\mathbb{E}_{\eta \sim N(0,I)}[\max_{z \in \mathcal{Z} \setminus \{z_*\}} \frac{(z_* - z)^\top A(\lambda)^{-1/2}\eta}{\Delta_z}]^2$$

$$+ \max_{z \neq z_*} \frac{\|z_* - z\|^2_{A(\lambda)^{-1}}}{\Delta_z^2}] \tag{40}$$

where line (39) follows by exercise 7.6.9 in [32].

It remains to bound the second term. Note that

$$\mathbb{E}_{\eta \sim N(0,I)}[\max_{z \in \mathcal{Z}} \frac{(z_* - \tilde{z}_k)^\top A(\lambda)^{-1/2}\eta}{2^{-k}\Gamma + \Delta_z}]^2 \leq \mathbb{E}_{\eta \sim N(0,I)}[\max(\frac{(z_* - \tilde{z}_k)^\top A(\lambda)^{-1/2}\eta}{2^{-k}\Gamma}, 0)]^2$$

$$\leq c\frac{\|z_* - \tilde{z}_k\|^2_{A(\lambda)^{-1}}}{(2^{-k}\Gamma)^2}$$

$$\leq c\frac{\|z_* - \tilde{z}_k\|^2_{A(\lambda)^{-1}}}{\Delta_{\tilde{z}_k}^2} \tag{41}$$

$$\leq c\max_{z \in \mathcal{Z} \setminus \{z_*\}} \frac{\|z_* - z\|^2_{A(\lambda)^{-1}}}{\Delta_z^2} \tag{42}$$

where line (41) follows since $\tilde{z}_k \in S_{k+2}$ by Lemma 1.

Thus, combining (37), (38), (40), and (42) yield the upper bound

$$\sum_{k=1}^{c\log(\Gamma/\Delta_{min})} N_k \le c\log(\Gamma/\Delta_{min})[d + \gamma^* + \rho^*].$$

**Step 5: Computation.** Next, we show that we can solve the optimization problems $\gamma(\mathcal{Z})$ and (27) approximately and bound the number of oracle calls. In the interest of brevity, define

$$g_k(\lambda) := \mathbb{E}_{\eta\sim N(0,I)}[\max_{z\in\mathcal{Z}}\frac{(\tilde{z}_k - z)^\top A(\lambda)^{-1/2}\eta}{2^{-k}\Gamma + \widehat{\theta}_k^\top(\tilde{z}_k - z)}]^2$$

Let $\mathcal{D}_{k,1}$ denote the event that $\text{GetAlloc}(\tilde{z}_k, \widehat{\theta}_k, 2^{-k}\Gamma, \frac{6\delta}{4\pi^2(k+1)^2})$ returns $\lambda_k \in \mathbf{\Delta}$ such that

$$g_k(\lambda_k) \le c[\inf_{\lambda\in\mathbf{\Delta}} g_k(\lambda) + 1] \tag{43}$$

Let $\mathcal{D}_{k,1}$ denote the event that $\text{GetAlloc}(\tilde{z}_k, \widehat{\theta}_k, 2^{-k}\Gamma, \frac{6\delta}{4\pi^2(k+1)^2})$ uses at most the following number of oracle calls

$$c[d + \log(\phi\cdot k^2) + \log(\log(d)^2\frac{d^3}{(\Gamma 2^{-k})^2}\frac{1}{\delta^2})]\log(d)^2\frac{d^3}{(\Gamma 2^{-k})^2}\frac{k^4}{\delta^2} \tag{44}$$

where $\phi \le \max_z\Delta_z + \Gamma$. Furthermore, define $\mathcal{D}_1 = \cap_k\mathcal{D}_{k,1}$ and $\mathcal{D}_2 = \cap_k\mathcal{D}_{k,1}$.

GetAlloc is applied with confidence level $\frac{6\delta}{4\pi^2 k^2}$, and thus by Theorem 8 and a standard union bound argument, with probability at least $\mathbb{P}(\mathcal{D}_1) \ge 1 - \frac{\delta}{4}$ and $\mathbb{P}(\mathcal{D}_2) \ge 1 - \frac{1}{2^d}\cdot\frac{1}{4}$.

Next, let $\mathcal{C}_k$ denote that event that $\text{EvalAlloc}(\tilde{z}_k, \widehat{\theta}_k, 2^{-k}\Gamma, \frac{6\delta}{4\pi^2(k+1)^2})$ that the algorithm outputs a $\tau_k$ such that

$$g_k(\lambda_k) \le \tau_k \le c[g_k(\lambda_k) + 1] \tag{45}$$

and the number of oracle calls is upper bounded by

$$O(\frac{d^2}{(\Gamma 2^{-k})^2}\log(k/\delta)\log(\frac{dk}{\Gamma 2^{-k}\delta})).$$

Define $\mathcal{C} = \cap_k\mathcal{C}_k$. Since EvalAlloc is applied with confidence level $\delta = \frac{6\delta}{4\pi^2 k^2}$ and by Lemma 3 and a standard union bound argument, $\mathbb{P}(\mathcal{C}) \ge 1 - \frac{\delta}{2}$.

Suppose that $\mathcal{D}_1\cap\mathcal{C}\cap\mathcal{E}$ occurs. Inspection of the proof reveals that nothing is lost by the approximation in (43) and (45). Thus, by a union bound, it follows that with probability at least $1 - 4\delta$, the algorithm terminates and returns $z_*$ after the stated number of samples in the theorem.

Now, suppose $\mathcal{D}_1 \cap \mathcal{D}_2 \cap \mathcal{C} \cap \mathcal{E}$ holds. Since there are $c\log(\Gamma/\Delta_{\min})$ rounds, the bound on the number of oracle calls follows by the dominant term appearing in line (44). Thus, by the union bound and assuming $\delta \ge \frac{1}{2^d}$, the event $\mathcal{D}_1 \cap \mathcal{D}_2 \cap \mathcal{C} \cap \mathcal{E}$ occurs with probability at least $1 - 4\delta$. This completes the proof.

$\square$

The following Lemma is an essential ingredient in the proof of the upper bound for the computationally efficient algorithm for combinatorial bandits.

**Lemma 1.** *Let $k \ge 1$. Consider the $k$th round of Algorithm 2. Suppose that*

- *if $z \in S_k^c$,*

$$|(z_* - z)^\top(\widehat{\theta}_k - \theta)| \le \frac{\Delta_z}{8} \tag{46}$$

- *if $z \in S_k$,*

$$|(z_* - z)^\top(\widehat{\theta}_k - \theta)| \le \frac{2^{-k}\Gamma}{8}. \tag{47}$$

*Then, the following hold:*

1.

$$\tilde{z}_k \in S_{k+2}, \tag{48}$$

2. *if* $z \in S_k^c$

$$|(\tilde{z}_k - z)^\top \widehat{\theta}_k - (z_* - z)^\top \theta| \le \frac{1}{2}\Delta_z. \tag{49}$$

3. *if* $z \in S_k$,

$$|(\tilde{z}_k - z)^\top \widehat{\theta}_k - (z_* - z)^\top \theta| \le \frac{1}{2}2^{-k}\Gamma. \tag{50}$$

4. *There exist universal constants* $c, c' > 0$ *such that*

$$c\mathbb{E}[\sup_{z \in \mathcal{Z}} \frac{(\tilde{z}_k - z)^\top A(\lambda)^{-1/2}\eta}{\Delta_z + 2^{-k}\Gamma}]^2 \le \mathbb{E}[\sup_{z \in \mathcal{Z}} \frac{(\tilde{z}_k - z)^\top A(\lambda)^{-1/2}\eta}{(\tilde{z}_k - z)^\top \widehat{\theta}_k + 2^{-k}\Gamma}]^2$$

$$\le c'\mathbb{E}[\sup_{z \in \mathcal{Z}} \frac{(\tilde{z}_k - z)^\top A(\lambda)^{-1/2}\eta}{\Delta_z + 2^{-k}\Gamma}]^2$$

*Proof.* **Step 1: 1 holds at round** $k$**.** Note that if $z \in S_{k+2}^c \cap S_k$, then

$$\widehat{\theta}_k^\top (z_* - z) \ge \Delta_z - \frac{2^{-k}\Gamma}{8} > 0$$

by (47) and since $z \in S_{k+2}^c \cap S_k$ implies that $\Delta_z \ge \frac{2^{-k}\Gamma}{4}$. Thus, $z \ne \tilde{z}_k$. On the other hand, if $z \in S_k^c$,

$$\widehat{\theta}_k^\top (z_* - z) \ge \Delta_z - \frac{\Delta_z}{8} > 0$$

by (46), so that $z \ne \tilde{z}_k$. Together, these cases together imply that $\tilde{z}_k \in S_{k+2}$.

**Step 2: 2 and 3 hold at round** $k$**.** First, suppose $z \in S_k^c$. We have that

$$
\begin{aligned}
|(\tilde{z}_k - z)^\top \widehat{\theta}_k - (z_* - z)^\top \theta| &\le |(\tilde{z}_k - z)^\top (\widehat{\theta} - \theta)| + |\theta^\top (\tilde{z}_k - z) - \theta^\top (z_* - z)| \\
&\le |(\tilde{z}_k - z_*)^\top (\widehat{\theta} - \theta)| + |(z_* - z)^\top (\widehat{\theta} - \theta)| + |\theta^\top (\tilde{z}_k - z_*)| \\
&\le \frac{1}{8}(2^{-k}\Gamma + \Delta_z) + \frac{1}{4}2^{-k}\Gamma \tag{51} \\
&\le \frac{1}{2}\Delta_z
\end{aligned}
$$

where line (51) follows by (46) and by (48) which we have shown holds at round $k$. By a similar argument, if $z \in S_k$,

$$|(\tilde{z}_k - z)^\top \widehat{\theta}_k - (z_* - z)^\top \theta| \le \frac{1}{2}2^{-k}\Gamma.$$

**Step 3: 4 holds at round** $k$**.** We have shown that (48) and (49) hold at round $k$. Fix $z \in \mathcal{Z}$. If $z \in S_k^c$, by (49) we have that $\Delta_z \ge \frac{2}{3}\widehat{\theta}^\top (\tilde{z}_k - z)$ and thus

$$\frac{1}{\Delta_z + 2^{-k}\Gamma} \le \frac{3}{2}\frac{1}{(\tilde{z}_k - z)^\top \widehat{\theta}_k + 2^{-k}\Gamma}.$$

On the other hand, if $z \in S_k$, by (50), we have that $\Delta_z \ge \widehat{\theta}_k(\tilde{z}_k - z) - \frac{2^{-k}\Gamma}{2}$. Thus,

$$\frac{1}{\Delta_z + 2^{-k}\Gamma} \le 2\frac{1}{(\tilde{z}_k - z)^\top \widehat{\theta}_k + 2^{-k}\Gamma}.$$

Therefore, since in addition $\tilde{z}_k \in \mathcal{Z}$, we may apply Lemma 14 to obtain

$$\mathbb{E}[\sup_{z \in \mathcal{Z}} \frac{(\tilde{z}_k - z)^\top A(\lambda)^{-1/2}\eta}{\Delta_z + 2^{-k}\Gamma}] \le 2\mathbb{E}[\sup_{z \in \mathcal{Z}} \frac{(\tilde{z}_k - z)^\top A(\lambda)^{-1/2}\eta}{(\tilde{z}_k - z)^\top \widehat{\theta}_k + 2^{-k}\Gamma}]$$

yielding one of the inequalities. By a similar argument, we obtain the other inequality, proving the claim. $\qquad \square$

# D Computational Results for Computationally Efficient Algorithm for Combinatorial Bandits

In this section, we present the computational subroutines for the computationally efficient algorithm for combinatorial bandits. The main optimization problem in Algorithm 2 is given in line (27). Fix $z_0 \in \mathcal{Z}$, $b > 0$, and $\theta_0 \in \mathbb{R}^d$ for the remainder of the section; we will omit dependence on these quantities because they are fixed. Since the Gaussian width is nonnegative, it suffices to solve:

$$\inf_{\lambda \in \boldsymbol{\Delta}} g(\lambda) := \mathbb{E}\max_{z \in \mathcal{Z}} \frac{(z_0 - z)^\top A(\lambda)^{-1/2}\eta}{b + \theta_0^\top(z_0 - z)}.$$

Define the following functions

$$g(\lambda; \eta) := \max_{z \in \mathcal{Z}} \frac{(z_0 - z)^\top A(\lambda)^{-1/2}\eta}{b + \theta_0^\top(z_0 - z)}$$

$$g(\lambda; \eta; z) := \frac{(z_0 - z)^\top A(\lambda)^{-1/2}\eta}{b + \theta_0^\top(z_0 - z)}$$

$$g(\lambda; \eta; r) := \max_{z \in \mathcal{Z}} z^\top(A(\lambda)^{-1/2}\eta + r\theta_0) - r(b + \theta_0^\top z_0) - z_0^\top A(\lambda)^{-1/2}\eta$$

$$g(\lambda; \eta; r; z) := z^\top(A(\lambda)^{-1/2}\eta + r\theta_0) - r(b + \theta_0^\top z_0) - z_0^\top A(\lambda)^{-1/2}\eta$$

## D.1 Main Subroutine

---

**Input:** $z_0 \in \mathcal{Z}$, $\theta_0 \in \mathbb{R}^d$, Offset $b > 0$, $\delta > 0$ ;
$\lambda \longleftarrow \text{GetAlloc}(z_0, \theta_0, b, \delta)$;
$\tau \longleftarrow \text{EvalAlloc}(z_0, \theta_0, b, \lambda, \delta)$;
**Return** $(\lambda, \tau)$

---

**Algorithm 6:** ComputeAlloc$(z_0, \theta_0, b, \delta)$

ComputeAlloc$(z_0, \theta_0, b, \delta)$ is the main subroutine; it solves and evaluates $\inf_\lambda g(\lambda)$. GetAlloc$(z_0, \theta_0, b, \delta)$ and EvalAlloc$(z_0, \theta_0, b, \lambda, \delta)$ only use calls to the linear maximization oracle. GetAlloc$(z_0, \theta_0, b, \delta)$ finds a solution within a constant additive factor of the optimal solution to the optimization problem $\inf_{\lambda \in \boldsymbol{\Delta}} g(\lambda)$ with probability at least $1 - \delta$. EvalAlloc$(z_0, \theta_0, b, \lambda, \delta)$ determines the value of $g(\lambda)$ within a constant additive factor with probability at least $1 - \delta$.

GetAlloc (Algorithm 7) performs stochastic mirror descent over the subset of the simplex that is a mixture with the uniform distribution

$$\tilde{\boldsymbol{\Delta}} := \{\lambda \in \mathbb{R}^d : \lambda = \frac{1}{2}(\kappa + \kappa') \text{ where } \kappa \in \boldsymbol{\Delta} \text{ and } \kappa' = (1/d, \dots, 1/d)^\top\}.$$

Define the Bregman divergence associated with a function $f$:

$$D_f(x, y) = f(x) - f(y) - \nabla f(y)^\top(x - y).$$

GetAlloc calls estimateGradient (Algorithm 8) to obtain an unbiased estimate of the gradient. estimateGradient needs to solve a maximization problem, for which it calls computeMax (Algorithm 9), a subroutine that essentially performs binary search.

EvalAlloc (Algorithm 10) estimates the number of samples to take in a round, only using calls to the linear maximization oracle. Because it estimates the mean of estimator that is not necessarily sub-Gaussian, but has controlled variance, this subroutine uses the median-of-means estimator.

**Input:** $z_0 \in \mathcal{Z}$, Offset $b \in \mathbb{R}$, $\theta_0 \in \mathbb{R}^d$, confidence level $\delta > 0$ ;
Define $\Phi(\lambda) = \sum_{i=1}^{d} \lambda_i \log(\lambda_i)$;
$T \longleftarrow c \log(d)^2 \frac{d^3}{b^2} \frac{1}{\delta^2}$ where $c > 0$ is a universal constant obtained in the proof of Theorem 8;
$\kappa = \frac{c'}{\frac{d^3}{b^2}} \sqrt{\frac{2}{T}}$ where $c' > 0$ is a universal constant obtained in the proof in of Theorem 8;
$\lambda^{(1)} \longleftarrow \operatorname{argmin}_{\lambda \in \tilde{\Delta}} \Phi(\lambda)$;
**for** $s = 1, 2, \ldots, T$ **do**
  Let $r_s \longleftarrow$ estimateGradient$(z_0, \theta_0, b, \lambda)$;
  $\lambda_{s+1} = \operatorname{argmin}_{\lambda \in \tilde{\Delta}} \kappa r_s^\top \lambda + D_\Phi(\lambda, \lambda_s)$
**Return** $\frac{1}{T} \sum_{s=1}^{T} \lambda^{(s)}$

**Algorithm 7:** GetAlloc$(z_0, \theta_0, b, \delta)$: Stochastic Mirror Descent for Transductive Bandits with linear maximization oracle

---

**Input:** $\lambda \in \Delta$, $z_0 \in \mathcal{Z}$, Offset $b \in \mathbb{R}$, $\theta_0 \in \mathbb{R}^d$;
Draw $\eta \sim N(0, I)$;
MAX-VAL $\longleftarrow$ computeMax$(z_0, \theta_0, b, \lambda, \eta, 0)$;
Choose

$$\bar{z} \in \arg\max_{z \in Z} g(\lambda; \eta; \text{MAX-VAL}; z)$$

Return $\nabla_\lambda g(\lambda; \eta; \bar{z})$

**Algorithm 8:** estimateGradient$(z_0, \theta_0, b, \lambda)$: Compute unbiased stochastic subgradient

---

**Input:** $\lambda \in \Delta$, $z_0 \in \mathcal{Z}$, Offset $b \in \mathbb{R}$, $\theta_0 \in \mathbb{R}^d$, $\eta \in \mathbb{R}^d$, TOL $\geq 0$;
Define

$$\text{LOW} = 0, \qquad \text{HIGH} = 2$$

**while** $g(\lambda; \eta : \text{HIGH}) \geq 0$ **do**
  HIGH $\longleftarrow 2 \cdot$ HIGH;
**while** $g(\lambda; \eta; \text{LOW}) \neq 0$ *or* $\frac{1}{2}(\text{HIGH} + \text{LOW}) > $ TOL **do**
  **if** $g(\lambda; \eta; \frac{1}{2}(\text{HIGH} + \text{LOW})) < 0$ **then**
    LOW $\longleftarrow \frac{1}{2}(\text{HIGH} + \text{LOW})$
  **else**
    HIGH $\longleftarrow \frac{1}{2}(\text{HIGH} + \text{LOW})$
  LOW $\longleftarrow g(\lambda; \eta; z')$ for some $z' \in \arg\max g(\lambda; \eta; \text{LOW}; z)$
Return LOW

**Algorithm 9:** computeMax$(z_0, \theta_0, b, \lambda, \eta, \text{TOL})$: Compute $g(\lambda; \eta)$

---

**Input:** $\lambda \in \Delta$, $z_0 \in \mathcal{Z}$, $\theta_0 \in \mathbb{R}^d$, Offset $b \in \mathbb{R}$ ;
$T \longleftarrow 864 \frac{d^2}{b^2} \log(1/\delta)$;
Draw $\eta_1, \ldots, \eta_T \sim N(0, I)$ ;
$y_s \longleftarrow$ computeMax$(z_0, \theta_0, b, \lambda, \eta_s, \text{TOL} = 1/2)$ for $s = 1, \ldots, T$;
Let $\tau$ be the output of the median of means estimator applied to $y_1, \ldots, y_T$;
**Return** $[\tau + 1]^2$

**Algorithm 10:** EvalAlloc$(z_0, \theta_0, b, \lambda)$: Estimate $g(\lambda)$

## D.2 Proofs

Recall the definitions:

$$g(\lambda; \eta) := \max_{z \in \mathcal{Z}} \frac{(z_0 - z)^\top A(\lambda)^{-1/2} \eta}{b + \theta_0^\top (z_0 - z)}$$

$$g(\lambda; \eta; z) := \frac{(z_0 - z)^\top A(\lambda)^{-1/2} \eta}{b + \theta_0^\top (z_0 - z)}$$

$$g(\lambda; \eta; r) := \max_{z \in \mathcal{Z}} z^\top (A(\lambda)^{-1/2} \eta + r\theta_0) - r(b + \theta_0^\top z_0) - z_0^\top A(\lambda)^{-1/2} \eta$$

$$g(\lambda; \eta; r; z) := z^\top (A(\lambda)^{-1/2} \eta + r\theta_0) - r(b + \theta_0^\top z_0) - z_0^\top A(\lambda)^{-1/2} \eta$$

We note that we may assume without loss of generality that $\theta_0^\top (z_0 - z) \geq 0$ for all $z \in \mathcal{Z}$. We define the event $\mathcal{B}_\lambda = \{| \arg \max_{z \in \mathcal{Z}} g(\lambda; \eta; z)| = 1\}$, which will appear in the analysis several times.

The following Lemma provides the guarantee for estimateGradient. Define $\phi = \sqrt{\max_{z \in \mathcal{Z}} \theta_0^\top (z_0 - z) + b}$.

**Lemma 2.** *Consider the combinatorial bandit setting. Fix $z_0 \in \mathcal{Z}$, $b > 0$, $\theta_0 \in \mathbb{R}^d$, and $\lambda \in \tilde{\Delta}$. estimateGradient$(z_0, \theta_0, b, \lambda)$ returns an unbiased stochastic gradient of the function $g(\lambda)$ with probability 1. Let $\xi > 0$. With probability at least $1 - \frac{2\xi}{2^d}$, it terminates after $O(d + \log(\frac{d}{b}) + \log(\frac{\phi}{\xi}))$ oracle calls.*

*Proof.* **Step 1: Correctness.** Let $\eta \sim N(0, I)$. Note that $\mathbb{E}g(\lambda; \eta) = g(\lambda)$. Since $\eta \sim N(0, I)$, with probability 1 $\arg \max_{z \in \mathcal{Z}} g(\lambda; \eta; z)$ is unique and, therefore,

$$\nabla_\lambda \max_z g(\lambda; \eta; z) = \nabla_\lambda g(\lambda; \eta; \arg \max_{z \in \mathcal{Z}} g(\lambda; \eta; z)).$$

By Lemma 8, we have that

$$\nabla_\lambda \mathbb{E}g(\lambda; \eta) = \mathbb{E}\nabla_\lambda g(\lambda; \eta) \mathbb{1}\{\mathcal{B}_\lambda\}.$$

Thus, we have

$$\nabla_\lambda \mathbb{E}\max_z g(\lambda; \eta; z) = \mathbb{E}\nabla_\lambda \max_z g(\lambda; \eta; z) \mathbb{1}\{\mathcal{B}_\lambda\} = \mathbb{E}\nabla_\lambda g(\lambda; \eta; \arg \max_{z \in \mathcal{Z}} g(\lambda; \eta; z)) \mathbb{1}\{\mathcal{B}_\lambda\}.$$

As a consequence, to show that estimateGradient returns an unbiased gradient, it suffices to show that Algorithm 8 identifies $\arg \max_{z \in \mathcal{Z}} g(\lambda; \eta; z)$. Note that $g(\lambda; \eta)$ is equivalent to the following linear program problem

$$r_* = \min_r r$$
$$\text{s.t. } g(\lambda; \eta; r) = \max_{z \in \mathcal{Z}} z^\top (A^{-1/2}(\lambda)\eta + \bar{r}\theta_0) - r(b + \theta_0^\top z_0) - z_0^\top A(\lambda)^{-1/2} \eta \leq 0.$$

The estimageGradient algorithm terminates once it finds $\bar{r} > 0$ such that $\max_{z \in \mathcal{Z}} g(\lambda; \eta; \bar{r}; z) = 0$. Let $\bar{z} \in \arg \max_{z \in \mathcal{Z}} g(\lambda; \eta; \bar{r}; z)$. Then,

$$0 = \max_z g(\lambda; \eta; \bar{r}; z)$$
$$= g(\lambda; \eta; \bar{r}; \bar{z})$$
$$= \bar{z}^\top (A^{-1/2}(\lambda)\eta + \bar{r}\theta_0) - \bar{r}(b + \theta_0^\top z_0) - z_0^\top A(\lambda)^{-1/2} \eta$$
$$> z^\top (A^{-1/2}(\lambda)\eta + \bar{r}\theta_0) - \bar{r}(b + \theta_0^\top z_0) - z_0^\top A(\lambda)^{-1/2} \eta.$$

where the strict inequality holds with probability 1 since $\eta \sim N(0, I)$. Rearranging the above inequality, this implies that for every for all $z \in \mathcal{Z} \setminus \{\bar{z}\}$

$$\frac{(z_0 - z)^\top A(\lambda)^{-1/2} \eta}{b + \theta_0^\top (z_0 - z)} < \bar{r} = \frac{(z_0 - \bar{z})^\top A(\lambda)^{-1/2} \eta}{b + \theta_0^\top (z_0 - \bar{z})}$$

implying that $\bar{z} = \arg \max_z g(\lambda; \bar{r}; z)$, showing estimateGradient returns an unbiased gradient.

**Step 2: Running time.** Next, we bound the number of oracle calls. Define $\tilde{y} = \sup_{z \in \mathcal{Z}} \frac{(z_0 - z)^\top A(\lambda)^{-1/2} \eta}{b + \theta_0^\top (z_0 - z)}$. By Theorem 5.8 of [2], we have that

$$\mathbb{V}(\tilde{y}) \leq 4 \sup_{z \in \mathcal{Z}} \mathbb{V}(\frac{(z_0 - z)^\top A(\lambda)^{-1/2} \eta}{b + \theta_0^\top (z_0 - z)}) \leq 8 \frac{d^2}{b^2}.$$

where we used $\lambda \in \tilde{\Delta}$. Define the event

$$\mathcal{E} = \{\sup_{z \in \mathcal{Z}} \frac{(z_0 - z)^\top A(\lambda)^{-1/2}\eta}{b + \theta_0^\top(z_0 - z)} \le 8\frac{d^2}{b^2}\frac{1}{\delta}\}$$

Thus, by Chebyshev's inequality, we have that

$$\mathbb{P}(\mathcal{E}^c) \le \delta. \tag{52}$$

Thus, choosing $\delta = \frac{\xi}{2^d}$, we have with probability at least $\mathbb{P}(\mathcal{E}^c) \le \frac{\xi}{2^d}$. Then, by Lemma 5 the first while loop requires

$$O(\log(\frac{d}{b\delta})) = O(\log(\frac{d}{b}) + d + \log(\frac{1}{\xi}))$$

oracle calls.

Next, we consider the second while loop. Define the event

$$\mathcal{D} = \{|g(\lambda; \eta; z) - g(\lambda; \eta; z)| > \frac{\xi}{\phi 2^{2d}}, \forall z \ne z' \in \mathcal{Z}\}.$$

By Lemma 4, we have that with probability at least $\mathcal{D} \ge 1 - \frac{\xi}{2^d}$. Then, by Lemma 5, the second while loop requires at most $O(d + \log(\frac{d}{b}) + \log(\frac{\phi}{\xi}))$ oracle calls. A standard union bound argument for event $\mathcal{E} \cap \mathcal{D}$ yields the result.

$\square$

The following Theorem provides the guarantee for GetAlloc.

**Theorem 8.** *Consider the combinatorial bandit setting. Fix $z_0 \in \mathcal{Z}$, $b > 0$, and $\theta_0 \in \mathbb{R}^d$. With probability at least $1 - \delta$ GetAlloc($z_0, \theta_0, b, \delta$) returns $\bar{\lambda} \in \Delta$ such that*

$$g(\bar{\lambda})^2 \le c[\min_{\lambda \in \Delta} g(\lambda)^2 + 1].$$

*Let $\xi > 0$. Furthermore, with probability at least $1 - \frac{2\xi}{2^d}$, the number of oracle calls is bounded above by*

$$c[d + \log(\phi/\xi) + \log(\log(d)^2 \frac{d^3}{b^2}\frac{1}{\delta^2})] \log(d)^2 \frac{d^3}{b^2}\frac{1}{\delta^2}.$$

*Proof.* **Step 1: Guarantee on final allocation.** Note that for any $z \in \mathcal{Z}$,

$$|\nabla_\lambda g(\lambda; \eta; z)_i \mathbb{1}\{\mathcal{B}_\lambda\}| = \mathbb{1}\{\mathcal{B}_\lambda\}\mathbb{1}\{i \in z_0 \Delta z\}|\frac{\lambda_i^{-3/2}\eta_i}{b + \theta_0^\top(z_0 - z)}|$$

and thus

$$\mathbb{E}\max_{z \in \mathcal{Z}} \|\nabla_\lambda g(\lambda; \eta; z)\mathbb{1}\{\mathcal{B}_\lambda\}\|_\infty^2 \le c\frac{d^3}{b^2}\mathbb{E}\max_i \eta_i^2$$

$$\le \log(d)c\frac{d^3}{b^2}$$

where we used the fact that $\lambda \in \tilde{\Delta}$.

Note that the mirror map used is

$$\Phi(\lambda) = \sum_{i=1}^d \lambda_i \log(\lambda_i).$$

It is not hard to see that

$$\sup_{\lambda \in \tilde{\Delta}} \Phi(\lambda) - \min_{\lambda' \in \tilde{\Delta}} \Phi(\lambda') \le \log(d)$$

By Theorem 6.1 in [4],

$$\mathbb{E}g(\bar{\lambda}) - \min_{\lambda \in \tilde{\Delta}} g(\lambda) \leq c \log(d) \frac{d^{3/2}}{b} \sqrt{\frac{1}{T}}.$$

Then, by Markov's inequality,

$$\mathbb{P}(g(\bar{\lambda}) - \min_{\lambda \in \tilde{\Delta}} g(\lambda) \geq 1) \leq \mathbb{E}g(\bar{\lambda}) - \min_{\lambda \in \Delta} g(\lambda)$$

$$\leq c \log(d) \frac{d^{3/2}}{b} \sqrt{\frac{1}{T}}$$

$$\leq c \log(d) \frac{d^{3/2}}{b} \sqrt{\frac{1}{T}}$$

$$= \delta$$

by our choice of $T$. Noting that $\min_{\lambda \in \tilde{\Delta}} g(\lambda) \leq \sqrt{2} \min_{\lambda \in \Delta} g(\lambda)$ yields the result.

**Step 2: Bound the number of oracle calls.** Using Lemma 2 with $\xi' = \frac{\xi}{T}$ and union bounding over each of the $T$ iterations, with probability at least $1 - \frac{2\xi}{2^d}$ the number of oracle calls is at most

$$c[d + \log(\frac{d}{b\delta}) + \log(\phi/\xi \cdot T)]T = c[d + \log(\phi/\xi) + \log(\log(d)^2 \frac{d^3}{b^2} \frac{1}{\delta^2})] \log(d)^2 \frac{d^3}{b^2} \frac{1}{\delta^2}.$$

$\square$

The following Lemma provides the guarantee for Algorithm 10.

**Lemma 3.** *With probability at least $1 - \delta$, Algorithm 10 returns $\tau$ such that $g(\lambda)^2 \leq (\tau + 1)^2 \leq g(\lambda)^2 + 4$. Furthermore, with probability at least, $1 - \delta$, it uses $O(\frac{d^2}{b^2} \log(1/\delta) \log(\frac{d}{b\delta}))$ oracle calls.*

*Proof.* Let $\tilde{y}_s = \sup_{z \in \mathcal{Z}} \frac{(z_0 - z)^\top A(\lambda)^{-1/2} \eta_s}{b + \theta_0^\top (z_0 - z)}$. By Theorem 5.8 of [2], we have that

$$\mathbb{V}(\tilde{y}_s) \leq 4 \sup_{z \in \mathcal{Z}} \mathbb{V}(\frac{(z_0 - z)^\top A(\lambda)^{-1/2} \eta}{b + \theta_0^\top (z_0 - z)}) \leq 8 \frac{d^2}{b^2}$$

Applying the median of means estimator (see [18]) to $\tilde{y}_1, \ldots, \tilde{y}_T$ yields that with probability at least $1 - \delta$ $\tilde{\tau}$ satisfies

$$|\tilde{\tau} - \mathbb{E} \sup_{z \in \mathcal{Z}} \frac{(z_0 - z)^\top A(\lambda)^{-1/2} \eta_s}{b + \theta_0^\top (z_0 - z)}| \leq 1/2$$

by our choice of $T$ and standard results for median of means estimation. Since the procedure computeMax a tolerance of $1/2$, by Lemma 5, we have that $|y_s - \tilde{y}_s| \leq 1/2$ for all $s = 1, \ldots, T$. Thus, it follows that $|\tilde{\tau} - \tau| \leq 1/2$. Thus,

$$|\tau - \mathbb{E} \sup_{z \in \mathcal{Z}} \frac{(z_0 - z)^\top A(\lambda)^{-1/2} \eta_s}{b + \theta_0^\top (z_0 - z)}| \leq 1.$$

Manipulating the above inequality yields the result.

It remains to bound the number of oracle calls. Consider $\tilde{y}_s = \sup_{z \in \mathcal{Z}} \frac{(z_0 - z)^\top A(\lambda)^{-1/2} \eta_s}{b + \theta_0^\top (z_0 - z)}$. By the same argument made in inequality (52), we have that with probability at least $1 - \frac{\delta}{T}$, $\tilde{y}_s \leq O(\frac{d^2 T}{b^2 \delta})$. Union bounding over all $\tilde{y}_s$ $s \in [T]$, we have that with probability at least $1 - \delta$, $\sup_s \tilde{y}_s \leq O(\frac{d^4}{b^4 \delta} \log(1/\delta))$. Since the procedure computeMax uses a tolerance of $1/2$, by Lemma 5 we have that each call of computeMax uses at most $O(\log(\frac{d}{b\delta}))$ calls to the linear maximization oracle, yielding the result. $\square$

### D.3 Technical Lemmas

**Lemma 4.** *Consider the combinatorial bandit setting. Fix $\theta_0 \in \mathbb{R}^d$ and $b \geq 0$. Let $\xi > 0$. Then,*

$$\mathbb{P}(\exists z \neq z' \in \mathcal{Z} : |g(\lambda; \eta; z) - g(\lambda; \eta; z')| \leq \frac{\xi}{\phi 2^{2d}}) \leq \frac{\xi}{2^d}.$$

*Proof.* Let $m = |\mathcal{Z}|$. Fix $z \neq z' \in \mathcal{Z}$. Fix $z_0 \in \mathcal{Z}$, $k \in \mathbb{N}$, and $\theta_0 \in \mathbb{R}^d$. For the sake of brevity, define $h(\tilde{z}) := g(\lambda; \eta; \tilde{z})$. Note that $|h(z) - h(z')|$ is a truncated normal distribution. Now, we lower bound its variance.

$$
\begin{aligned}
\mathbb{V}(h(z) - h(z')) &= \left\| \frac{(z_0 - z)^\top A(\lambda)^{-1/2}}{2^{-k}B + \theta_0^\top (z_0 - z)} - \frac{(z_0 - z')^\top A(\lambda)^{-1/2}}{b + \theta_0^\top (z_0 - z')} \right\|_2 \\
&\geq \left\| \frac{(z_0 - z)^\top A(\lambda)^{-1/2}}{b + \theta_0^\top (z_0 - z)} - \frac{(z_0 - z')^\top A(\lambda)^{-1/2}}{b + \theta_0^\top (z_0 - z')} \right\|_\infty \\
&\geq \min_{v \in \mathcal{Z}} \left\| \frac{1}{b + \theta_0^\top (z_0 - v)} \right\|_\infty \\
&\geq \frac{1}{\phi^2}
\end{aligned}
$$

where we used the fact that for every $\|z - z'\|_\infty \geq 1$ for combinatorial bandits and and the definition of $\phi$.

Then, using the cdf of the half normal, we have that

$$
\begin{aligned}
\mathbb{P}(|h(z) - h(z')| \leq \frac{\xi}{\phi 2^{2d}}) &= \int_0^{\frac{\xi}{\phi 2^{2d}}} \frac{1}{\sqrt{\mathbb{V}(h(z) - h(z'))}} \sqrt{2/\pi} \exp\left(-\frac{y^2}{2\mathbb{V}(h(z) - h(z'))}\right) dy \\
&\leq \int_0^{\frac{\xi}{\phi 2^{2d}}} \phi \sqrt{2/\pi} \exp\left(-\frac{y^2}{2\mathbb{V}(h(z) - h(z'))}\right) dy \\
&\leq \sqrt{2/\pi} \frac{\xi}{2^{2d}}.
\end{aligned}
$$

Thus, using a union bound, we have that

$$
\begin{aligned}
\mathbb{P}(\exists z \neq z' \in \mathcal{Z} : |h(z) - h(z')| \leq \frac{\xi}{\phi 2^{2d}} &\leq \frac{|\mathcal{Z}|}{2^{2d}}) \\
&\leq \frac{\xi}{2^d}.
\end{aligned}
$$

$\square$

Lemmas 5, 6, and 7 show that Algorithm 9 essentially performs binary search.

**Lemma 5.** *The following two claims holds regarding Algorithm 9.*

1. *At the end of the first while loop of Algorithm 9, $g(\lambda; \eta) \in [\text{LOW}, \text{HIGH}]$ and it takes at most $O(\log(g(\lambda; \eta)))$ oracle calls.*

2. *In the second while loop of Algorithm 9, it always holds that $g(\lambda; \eta) \in [\text{LOW}, \text{HIGH}]$. Furthermore, define $\bar{z} = \arg\max_{z \in \mathcal{Z}} g(\lambda; \eta; z)$. Then, if $g(\lambda; \eta) - max_{z \neq \bar{z}} g(\lambda; \eta; z) > \varepsilon$, then it terminates after $O(\log(\frac{g(\lambda; \eta)}{\varepsilon}))$ oracle calls.*

*Proof.* We begin by proving the first claim. By Lemma 6, if $\text{HIGH} < g(\lambda; \eta)$, then $g(\lambda; \eta; \text{HIGH}) > 0$ and HIGH keeps increasing. At some point, we have $\text{HIGH} > g(\lambda; \eta)$, which by Lemma 6 implies that $g(\lambda; \eta; \text{HIGH}) < 0$ and the while loop terminates. Notice that since $z_0 \in \mathcal{Z}$, $g(\lambda; \eta) \geq 0 = \text{LOW}$. Furthermore, since HIGH doubles at each round the first while loop takes at most $O(\log(g(\lambda; \eta))))$ oracle calls. This completes the proof of the first claim.

Next, we prove the second claim regarding the second while loop. At the beginning of the second while loop, $g(\lambda; \eta) \in [\text{LOW}, \text{HIGH}]$. It is a straightforward consequence of Lemma 6 that at the end

of the if else statement in the second while loop it holds that $g(\lambda; \eta) \in [\text{LOW}, \text{HIGH}]$. In the last line of the while loop where

$$\text{LOW} \longleftarrow g(\lambda; \eta; z') \text{ for some } z' \in \arg\max g(\lambda; \eta; \text{LOW}; z)$$

it follows from Lemma 7 that $g(\lambda; \eta; \text{LOW}) \geq 0$. Then, by Lemma 6, it follows that $g(\lambda; \eta) \geq \text{LOW}$. Thus, the claim that $g(\lambda; \eta) \in [\text{LOW}, \text{HIGH}]$ during the second while loop holds.

Finally, we bound the number of oracle calls. Assume $g(\lambda; \eta) - \max_{z \neq \bar{z}} g(\lambda; \eta; z) > \varepsilon$ where $\bar{z} = \arg\max_{z \in \mathcal{Z}} g(\lambda; \eta; z)$. Since at the end of the first while loop $\text{HIGH} \leq 2g(\lambda; \eta)$ and the second while loop performs binary search, we have that after $O(\log(\frac{g(\lambda; \eta)}{\varepsilon}))$ oracle calls,

$$g(\lambda; \eta) \geq \text{LOW} > g(\lambda; \eta) - \varepsilon.$$

Let $y \in \arg\max_{z \in \mathcal{Z}} g(\lambda; \eta; \text{LOW}; z)$; we claim that $y = \arg\max_{z \in \mathcal{Z}} g(\lambda; \eta; z)$. By Lemma 7, we have that $g(\lambda; \eta; \text{LOW}; y) \geq 0$. Rearranging, we obtain

$$\frac{(z_0 - y)^\top A(\lambda)^{-1/2} \eta}{b + \theta_0^\top (z_0 - y)} \geq \text{LOW} > g(\lambda; \eta) - \varepsilon > \max_{z \neq \bar{z}} g(\lambda; \eta; z),$$

which implies that

$$y = \arg\max_{z \in \mathcal{Z}} g(\lambda; \eta; z) = \arg\max_{z \in \mathcal{Z}} \frac{(z_0 - z)^\top A(\lambda)^{-1/2} \eta}{b + \theta_0^\top (z_0 - z)}$$

proving the claim.

Thus, inspection of the algorithm shows that it suffices to show that $g(\lambda; \eta; g(\lambda; \eta)) = 0$, but this follows directly from Lemma 6.

$\square$

**Lemma 6.** *If $g(\lambda; \eta; r) < 0$, then $r > g(\lambda; \eta)$ and if $g(\lambda; \eta; r) > 0$, then $r < g(\lambda; \eta)$.*

*Proof.* Suppose $g(\lambda; \eta; r) < 0$. Then, by definition,

$$\max_{z \in \mathcal{Z}} z^\top (A^{-1/2}(\lambda)\eta + r\theta_0) - r(b + \theta_0^\top z_0) - z_0^\top A(\lambda)^{-1/2} \eta < 0.$$

Rearranging, we have that for all $z \in \mathcal{Z}$,

$$\frac{(z_0 - z)^\top A(\lambda)^{-1/2} \eta}{b + \theta_0^\top (z_0 - z)} < r,$$

thus proving the first claim. Next, suppose $g(\lambda; \eta; r) > 0$. Then, rearranging as above, there exists a $z \in \mathcal{Z}$ such that

$$\frac{(z_0 - z)^\top A(\lambda)^{-1/2} \eta}{b + \theta_0^\top (z_0 - z)} > r,$$

proving the second claim.

$\square$

**Lemma 7.** *If $\max_{z \in \mathcal{Z}} g(\lambda; \eta; z; L_1) \geq 0$, then letting $L_2 = g(\lambda; \eta; z')$ for some $z' \in \arg\max_{z \in \mathcal{Z}} g(\lambda; \eta; z; L_1)$, we have that $L_2 \geq L_1$ and $g(\lambda; \eta; L_2) \geq 0$. Furthermore, $g(\lambda; \eta; \text{LOW}) \geq 0$ throughout the execution of Algorithm 9.*

*Proof.* We have that

$$g(\lambda; \eta; z'; L_1) = (z')^\top (A^{-1/2}(\lambda)\eta + L_1\theta_0) - L_1(b + \theta_0^\top z_0) - z_0^\top A(\lambda)^{-1/2} \eta \geq 0.$$

Rearranging, we have that

$$L_2 := \frac{(z_0 - z')^\top A(\lambda)^{-1/2} \eta}{b + \theta_0^\top (z_0 - z')} \geq L_1,$$

proving the first claim. Furthermore, rearranging the equality

$$\frac{(z_0 - z')^\top A(\lambda)^{-1/2} \eta}{b + \theta_0^\top (z_0 - z')} = L_2$$

yields $0 = g(\lambda; \eta; z'; L_2) \leq \max_{z \in \mathcal{Z}} g(\lambda; \eta; z; L_2)$, yielding the second inequality.

Finally, $g(\lambda; \eta; \text{LOW}) \geq 0$ follows inductively. In the base case, $\text{LOW} = 0$ and we observe that for $0 = g(\lambda; \eta; z_0; 0) \leq \max_{z \in \mathcal{Z}} g(\lambda; \eta; z; 0)$. The inductive step follows by the update and the above claims.

$\square$

Lemma 8 shows $g(\lambda)$ is differentiable.

**Lemma 8.** *Let $\lambda \in \tilde{\Delta}$. Then, $\mathbb{E}g(\lambda; \eta)$ is differentiable at $\lambda$ and $\nabla_\lambda \mathbb{E}g(\lambda; \eta) = \mathbb{E}\nabla_\lambda g(\lambda; \eta)\mathbb{1}\{\mathcal{B}_\lambda\}$ where*

$$\mathcal{B}_\lambda = \{|\arg\max_{z \in \mathcal{Z}} g(\lambda; \eta; z)| = 1\}.$$

*Proof.* Note that

$$\mathbb{E}\nabla_\lambda g(\lambda; \eta)\mathbb{1}\{\mathcal{B}_\lambda\} = \nabla_\lambda \mathbb{E}g(\lambda; \eta) \iff (\mathbb{E}\nabla_\lambda g(\lambda; \eta)\mathbb{1}\{\mathcal{B}_\lambda\})_i = (\nabla_\lambda \mathbb{E}g(\lambda; \eta))_i \quad \forall i$$

and thus, it suffices to prove the statement for a single fixed $i$. Note that since $\lambda \in \tilde{\Delta}$, $A(\lambda)^{-1/2}$ is full rank and hence each $\frac{A(\lambda)^{-1/2}(z_0 - z)}{b + \theta_0^\top (z_0 - z)}$ is distinct . Therefore, since $\eta \sim N(0, I)$, $\mathcal{B}_\lambda$ occurs with probability 1.

Note that for each fixed $z$, since $\lambda \in \tilde{\Delta}$,

$$|\frac{\partial g(\lambda; \eta; z)}{\partial \lambda_i}| \leq |\frac{1}{2}\frac{\lambda_i^{-3/2}\eta_i}{b + \theta_0^\top (z_0 - z)}| \leq c|\frac{d^{3/2}\eta_i}{b}| =: L_\eta$$

$g(\lambda; \eta; z)$ is $L_\eta$-Lipschitz in $\lambda_i$. Since $g(\lambda; \eta) = \max_{z \in \mathcal{Z}} g(\lambda; \eta; z)$, $g(\lambda; \eta)$ is $L_\eta$-Lipschitz in $\lambda_i$. Thus, we have that for any $h > 0$,

$$\frac{g(\lambda + he_i; \eta) - g(\lambda; \eta)}{h} \leq L_\eta$$

and so by the Dominated Convergence Theorem

$$\lim_{h \to 0} \mathbb{E}[\frac{g(\lambda + he_i; \eta) - g(\lambda; \eta)}{h}] = \mathbb{E}[\lim_{h \to 0} \frac{g(\lambda + he_i; \eta) - g(\lambda; \eta)}{h}]$$

$$= \mathbb{E}[\lim_{h \to 0} \frac{g(\lambda + he_i; \eta) - g(\lambda; \eta)}{h}\mathbb{1}\{\mathcal{B}_\lambda\}]$$

$$= \mathbb{E}[\frac{\partial g(\lambda; \eta)}{\partial \lambda_i}\mathbb{1}\{\mathcal{B}_\lambda\}]$$

which shows that the partial derivatives of $\mathbb{E}[g(\lambda; \eta)]$ exist and

$$\frac{\partial \mathbb{E}[g(\lambda; \eta)]}{\partial \lambda_i} = \mathbb{E}[\frac{\partial g(\lambda; \eta)}{\partial \lambda_i}\mathbb{1}\{\mathcal{B}_\lambda\}].$$

To show that $\mathbb{E}[g(\lambda; \eta)]$ is differentiable, it suffices to show that the partial derivatives $\frac{\partial \mathbb{E}[g(\lambda; \eta)]}{\partial \lambda_i}$ are continuous. Let $\lambda^{(n)} \in \tilde{\Delta}$ be a sequence such that $\lim_{n \to \infty} \lambda^{(n)} = \lambda$. Then, a straightforward application of the Dominated Convergence Theorem shows that

$$\lim_{n \to \infty} \mathbb{E}[\frac{\partial g(\lambda^{(n)}; \eta)}{\partial \lambda_i^{(n)}}\mathbb{1}\{\mathcal{B}_{\lambda^{(n)}}\}] = \mathbb{E}[\lim_{n \to \infty} \frac{\partial g(\lambda^{(n)}; \eta)}{\partial \lambda_i^{(n)}}\mathbb{1}\{\mathcal{B}_{\lambda^{(n)}}\}]$$

$$= \mathbb{E}[\lim_{n \to \infty} \frac{\partial g(\lambda^{(n)}; \eta)}{\partial \lambda_i^{(n)}}\mathbb{1}\{\mathcal{B}_{\lambda^{(n)}}\}\mathbb{1}\{\mathcal{B}_\lambda\}]$$

$$= \mathbb{E}[\frac{\partial g(\lambda; \eta)}{\partial \lambda_i}\mathbb{1}\{\mathcal{B}_\lambda\}]$$

where we used that $\lim_{n \to \infty} \mathbb{1}\{\mathcal{B}_{\lambda^{(n)}}\}\mathbb{1}\{\mathcal{B}_\lambda\} = \mathbb{1}\{\mathcal{B}_\lambda\}$. This is true since clearly if $\eta$ is such that $\mathbb{1}\{\mathcal{B}_\lambda\} = 0$, then the claim follows. On other hand, if $\eta$ is such that $\mathbb{1}\{\mathcal{B}_\lambda\} = 1$, then using the Lipschitzness of $g(\lambda; \eta; z)$ (as previously argued), we have that $\lim_{n \to \infty} \mathbb{1}\{\mathcal{B}_{\lambda^{(n)}}\} = \mathbb{1}\{\mathcal{B}_\lambda\}$. Thus, we conclude that the partial derivatives are continuous, which completes the proof.

$\square$

# E   Fixed Budget Upper Bound Proofs

Lemma 9 is the main step in the proof of the upper bound for the fixed budget algorithm.

**Lemma 9.** *Suppose $T \geq cR\max([\rho^* + \gamma^*], d)$. If $z_* \in \mathcal{Z}_k$, then $z_*$ is eliminated in round $k$ with probability at most*

$$2\exp(\frac{-T}{c'[\rho^* + \gamma^*]}).$$

*Proof.* Let $N = \lfloor T/R \rfloor$. Let $\mathcal{X} = \{x_1, \ldots, x_m\}$. Let $\lambda_k$ denote the design chosen by the algorithm in round $k$. Let $x_{I_1}, \ldots, x_{I_N}$ denote the measurement vectors selected in round $k$ and define $\bar{\lambda} \in \Delta$ by $\bar{\lambda}_i = \frac{1}{N} \sum_{s=1}^{N} \mathbb{1}\{I_s = i\}$. Let $\xi > 0$ (a constant to be chosen later). Define

$$\Delta = \mathrm{argmin} \Delta'$$

$$\text{s.t.} \quad \sup_{z,z' \in \mathcal{Z}_k} \frac{\|z - z'\|_{A(\bar{\lambda})^{-1}}^2}{(\Delta')^2} \leq \xi[\rho^* + \gamma^*].$$

Define the event

$$\mathcal{E} = \{ \sup_{z,z' \in \mathcal{Z}_k} \frac{|(z - z')^\top (\widehat{\theta}_k - \theta)|}{\Delta} \leq \sqrt{\frac{\mathbb{E}[\sup_{z,z' \in \mathcal{Z}_k} \frac{(z-z')^\top A(\bar{\lambda})^{-1/2}\eta}{\Delta}]^2}{\lfloor T/R \rfloor}} + \frac{1}{2} \}.$$

By Theorem 5.8 in [2] with probability at least

$$\mathbb{P}(\mathcal{E}^c) \leq 2\exp(\frac{-\lfloor T/R \rfloor}{8 \frac{\sup_{z,z' \in \mathcal{Z}_k} \|z - z'\|_{A(\bar{\lambda})^{-1}}^2}{\Delta^2}}) \leq 2\exp(\frac{-\lfloor T/R \rfloor}{8\xi[\rho^* + \gamma^*]})$$

where we used the definition of $\Delta$. Suppose $\mathcal{E}$ occurs for the remainder of the proof.

Define

$$\mathcal{Z}_{k,wrong} = \{z \in \mathcal{Z}_k : \widehat{\theta}_k^\top (z_* - z) < 0\}.$$

Towards a contradiction, suppose $z_*$ is eliminated at round $k$. Then, by definition of the algorithm,

$$\gamma(\mathcal{Z}_{k,wrong} \cup \{z_*\}) \geq \frac{\gamma(\mathcal{Z}_k)}{2} = \frac{1}{2}\mathbb{E} \sup_{z,z' \in \mathcal{Z}_k} (z - z')^\top A(\lambda_k)^{-1/2}\eta.$$

Define $z_0 = \arg\max_{z \in \mathcal{Z}_{k,wrong}} \Delta_z$. Then,

$$\frac{1}{2(1+\epsilon)}\mathbb{E} \sup_{z,z' \in \mathcal{Z}_k} (z - z')^\top A(\bar{\lambda})^{-1/2}\eta \leq \frac{1}{2}\mathbb{E} \sup_{z,z' \in \mathcal{Z}_k} (z - z')^\top A(\lambda_k)^{-1/2}\eta \tag{53}$$

$$\leq \min_\lambda \mathbb{E} \sup_{z,z' \in \mathcal{Z}_{k,wrong} \cup \{z_*\}} (z - z')^\top A(\lambda)^{-1/2}\eta$$

$$\leq c\min_\lambda \mathbb{E} \sup_{z \in \mathcal{Z}_{k,wrong} \cup \{z_*\}} (z_* - z)^\top A(\lambda)^{-1/2}\eta$$

$$\leq c'\min_\lambda \mathbb{E} \sup_{z \in \mathcal{Z}_{k,wrong}} (z_* - z)^\top A(\lambda)^{-1/2}\eta$$

$$+ \|z_* - z_0\|_{A(\lambda)^{-1}} \tag{54}$$

where line (53) follows by the guarantees of the rounding procedure and Lemma 11 and line (54) follows by Lemma 16. Thus,

$$\mathbb{E}[\frac{\sup_{z,z' \in \mathcal{Z}_k} (z - z')^\top A(\bar{\lambda})^{-1/2}\eta}{\Delta_{z_0}}]^2 \leq c\min_\lambda \mathbb{E}[\sup_{z \in \mathcal{Z}_{k,wrong}} \frac{(z_* - z)^\top A(\lambda)^{-1/2}\eta}{\Delta_{z_0}}]^2$$

$$+ \frac{\|z_* - z_0\|_{A(\lambda)^{-1}}^2}{\Delta_{z_0}^2}$$

$$\leq c'[\gamma^* + \rho^*] \tag{55}$$

where line (55) follows by Lemma 13. Furthermore, we have that

$$\mathbb{E}[\frac{\sup_{z,z'\in\mathcal{Z}_k}(z-z')^\top A(\bar{\lambda})^{-1/2}\eta}{\Delta_{z_0}}]^2 \geq c \sup_{z,z'\in\mathcal{Z}_k} \frac{\|z-z'\|_{A(\bar{\lambda})^{-1}}^2}{\Delta_{z_0}^2} \qquad (56)$$

by Lemma 12. Combining inequalities (55) and (56), we have that there exists a univesral constant $\xi > 0$ such that $\Delta_{z_0} \geq \Delta$ (choose this $\xi$).

Then,

$$|(z_* - z_0)^\top(\widehat{\theta}_k - \theta)| \leq \sup_{z,z'\in\mathcal{Z}_k} |(z-z')^\top(\widehat{\theta}_k - \theta)|$$

$$\leq \Delta_{z_0}\sqrt{\frac{\mathbb{E}[\sup_{z,z'\in\mathcal{Z}_k}\frac{(z-z')^\top A(\bar{\lambda})^{-1/2}\eta}{\Delta_{z_0}}]^2}{\lfloor T/R \rfloor}} + \frac{\Delta}{2} \qquad (57)$$

$$\leq \Delta_{z_0}c'\sqrt{\frac{\gamma^* + \rho^*}{\lfloor T/R \rfloor}} + \frac{\Delta}{2} \qquad (58)$$

$$< \frac{\Delta_{z_0}}{2} + \frac{\Delta}{2} \qquad (59)$$

$$\leq \Delta_{z_0}.$$

where line (57) follows by the event $\mathcal{E}$, line (58) follows by (55), and line (59) follows since $T \geq cR[\rho^* + \gamma^*]$ for an appropriately large universal constant $c > 0$. Rearranging the above inequality implies that

$$(z_* - z_0)^\top\widehat{\theta}_k > 0$$

and thus $z_0 \notin \mathcal{Z}_{k,wrong}$, a contradiction. Therefore, on $\mathcal{E}$, $z_*$ is not eliminated.

$\square$

*Proof of Theorem 7.* Define the event

$$E_k = \{z_* \text{ is not eliminated in round } k\},$$
$$E = \cap_{k=0}^R E_k.$$

Then, by the law of total probability, Lemma 9, and the definition of $R = \lceil\log(\gamma(\mathcal{Z}))\rceil$,

$$\mathbb{P}(E^c) \leq \mathbb{P}(E_1^c) + \sum_{k=2}^R \mathbb{P}(E_k^c | \cap_{l=1}^{k-1} E_l)$$

$$\leq \lceil\log(\gamma(\mathcal{Z}))\rceil \exp(\frac{-T}{32R[\rho^* + \gamma^*]}).$$

Assume the event $E$ holds. Recall the assumption that $\gamma(\{z, z_*\}) \geq 1$ for all $z \in \mathcal{Z} \setminus \{z_*\}$. Since by the definition of the algorithm and $R$,

$$\gamma(\mathcal{Z}_R) \leq \frac{\gamma(\mathcal{Z})}{2^R} \leq 1$$

the algorithm must terminate in one of the $\lceil\log(\gamma(\mathcal{Z}))\rceil$ rounds and return $z_*$, completing the proof.

$\square$

# F  $\gamma^*$ Results

In this Section, we prove various results related to $\gamma^*$.

*Proof of Proposition 2.* Define $\theta = e_1$ and $z_* = e_1$. Let

$$\mathcal{Z} = \{v \in \mathbb{R}^d : \|v\|_2 = 1, v_1 = 0\} \cup \{z_*\}.$$

Let $\mathcal{X} = \{e_1, \ldots, e_d\}$. Then, for any $\lambda \in \boldsymbol{\Delta}$,

$$\mathbb{E}_{\eta \sim N(0,I)}[\max_{z \in \mathcal{Z} \setminus \{z_*\}} \frac{(z_* - z)^\top A(\lambda)^{-1/2} \eta}{\theta^\top (z_* - z)}]^2 = \mathbb{E}_{\eta \sim N(0,I)}[\max_{z \in \mathcal{Z} \setminus \{z_*\}} (z_* - z)^\top A(\lambda)^{-1/2} \eta]^2$$

$$= \mathbb{E}_{\eta \sim N(0,I)}[\max_{z \in \mathcal{Z} \setminus \{z_*\}} z^\top A(\lambda)^{-1/2} \eta]^2$$

$$\geq (d-1) \mathbb{E}_{\eta \sim N(0,I)}[\max_{z \in \mathcal{Z} \setminus \{z_*\}} z^\top \eta]^2$$

$$\geq (d-1)(d+c)$$

where $c$ is a universal constant where the second to last inequality follows by symmetry and the last inequality follows by example 7.5.7 in [32]. On the other hand,

$$\rho^* = \inf_\lambda \max_{z \in \mathcal{Z} \setminus \{z_*\}} \frac{\|z_* - z\|_{A(\lambda)^{-1}}^2}{\theta^\top (z_* - z)^2}$$

$$= \inf_\lambda \max_{z \in \mathcal{Z} \setminus \{z_*\}} \|z_* - z\|_{A(\lambda)^{-1}}^2$$

$$\leq 4d$$

where we took $\lambda = (1/d, \ldots, 1/d)^\top$. Thus, there exists an instance where $\rho^* \leq dc$ and $\gamma^* \geq c'd^2$, proving the result.

Top-K is an example of a problem instance where $\gamma^* \leq c \log(d) \rho^*$ (see Proposition 6).

$\square$

**Proposition 6.** *Consider an instance of Top-K. Assume wlog $\theta_1 \geq \theta_2 \geq \ldots \geq \theta_d$.*

$$\gamma^* \leq c \log(d) [\sum_{i \leq k} (\theta_i - \theta_{k+1})^{-2} + \sum_{i > k} (\theta_k - \theta_i)^{-2}].$$

*Proof of Proposition 6.* Define

$$\Delta_i = \begin{cases} \theta_i - \theta_{k+1} & \text{if } i \leq k \\ \theta_k - \theta_i & \text{if } i > k \end{cases}$$

Set $\lambda_i = \frac{\Delta_i^{-2}}{\sum_{j \in [d]} \Delta_j^{-2}}$. Note that $\mathcal{Z} = \{z \subset [d] : |z| = k\}$. Then,

$$\gamma^* \leq \mathbb{E}_{\eta \sim N(0,I)}[\max_{z \subset \mathcal{Z} \setminus [k]} \frac{\sum_{i \in [k] \Delta z} \frac{1}{\sqrt{\lambda_i}} \eta_i}{\sum_{i \in [k] \setminus z} \theta_i - \sum_{j \in z \setminus [k]} \theta_j}]^2$$

$$= \sum_{j \in [n]} \Delta_j^{-2} \mathbb{E}_{\eta \sim N(0,I)}[\max_{z \subset \mathcal{Z} \setminus [k]} \frac{\sum_{i \in [k] \setminus z} (\theta_i - \theta_k) \eta_i + \sum_{j \in z \setminus [k]} (\theta_{k+1} - \theta_j) \eta_j}{\sum_{i \in [k] \setminus z} \theta_i - \sum_{j \in z \setminus [k]} \theta_j}]^2$$

$$= \sum_{j \in [n]} \Delta_j^{-2} \mathbb{E}_{\eta \sim N(0,I)}[\max_{z \subset \mathcal{Z} \setminus [k]} v_z^\top \eta + w_z^\top \eta]^2$$

where we defined the vectors

$$(v_z)_i = \frac{\theta_i - \theta_k}{\sum_{i \in [k] \setminus z} \theta_i - \sum_{j \in z \setminus [k]} \theta_j} \mathbf{1}\{i \in [k] \setminus z\}$$

$$(w_z)_i = \frac{\theta_{k+1} - \theta_i}{\sum_{i \in [k] \setminus z} \theta_i - \sum_{j \in z \setminus [k]} \theta_j} \mathbf{1}\{i \in [z] \setminus [k]\}$$

Note that

$$\|v_z\|_1 = \frac{\sum_{i \in [k] \setminus z} (\theta_i - \theta_k)}{\sum_{i \in [k] \setminus z} \theta_i - \sum_{j \in z \setminus [k]} \theta_j} \leq \frac{\sum_{i \in [k] \setminus z} \theta_i - \sum_{j \in z \setminus [k]} \theta_j}{\sum_{i \in [k] \setminus z} \theta_i - \sum_{j \in z \setminus [k]} \theta_j} = 1$$

where we used the fact that $|[k] \setminus z| = |z \setminus [k]|$ and the assumption $\theta_1 \geq \theta_2 \geq \ldots \geq \theta_n$. Similarly,

$$\|w_z\|_1 \leq 1.$$

Thus,

$$
\begin{aligned}
\gamma^* &\leq \sum_{j \in [d]} \Delta_j^{-2} \mathbb{E}_{\eta \sim N(0,I)} [\max_{z \subset \mathcal{Z} \setminus [k]} v_z^\top \eta + w_z^\top \eta]^2 \\
&\leq \sum_{j \in [d]} \Delta_j^{-2} \mathbb{E}_{\eta \sim N(0,I)} [\max_{v:\|v\|_1 \leq 1} v^\top \eta + \max_{w:\|w\|_1 \leq 1} w^\top \eta]^2 \\
&\leq c \log(d) \sum_{j \in [d]} \Delta_j^{-2}
\end{aligned}
$$

where in the final inequality we used Example 7.5.9 of [32].

$\square$

*Proof of Proposition 1.*

$$
\begin{aligned}
\gamma^* &= \inf_\lambda \mathbb{E}_{\eta \sim N(0,I)} [\max_{z \in \mathcal{Z}} \frac{[A(\lambda)^{-1/2}(z_* - z)]^\top \eta}{\theta^\top (z_* - z)}]^2 \\
&\leq \inf_\lambda c \log(|\mathcal{Z}|) \operatorname{diam}(\{\frac{A(\lambda)^{-1/2}(z_* - z)}{\theta^\top (z_* - z)} : z \in \mathcal{Z} \setminus \{z_*\}\})^2 \quad (60) \\
&\leq c' \log(|\mathcal{Z}|) \inf_\lambda \max_{z \in \mathcal{Z} \setminus z_*} \left\| \frac{A(\lambda)^{-1/2}(z_* - z)}{\theta^\top (z_* - z)} \right\|_2^2 \\
&= c' \log(|\mathcal{Z}|) \inf_\lambda \max_{z \in \mathcal{Z} \setminus z_*} \frac{\|z_* - z\|_{A(\lambda)^{-1}}^2}{\theta^\top (z_* - z)^2} \\
&= c' \log(|\mathcal{Z}|) \rho^*
\end{aligned}
$$

where we used exercise 7.5.10 of [32] in line (60). On the other hand, Proposition 7.5.2 of [32] implies that

$$\gamma^* \leq d\rho^*$$

Now, we prove the lower bound. There exists $\xi > 0$, $z_1 \in \mathcal{Z}$, and $\lambda_1 \in \boldsymbol{\Delta}$ such that

$$\xi + \inf_{z \neq z_*} \inf_{\lambda \in \boldsymbol{\Delta}} \frac{\|z_* - z\|_{A(\lambda)^{-1}}^2}{\Delta_z^2} \geq \frac{\|z_* - z_1\|_{A(\lambda_1)^{-1}}}{\Delta_{z_1}^2}.$$

Let $\lambda_2 \in \mathbf{\Delta}$ attain $\gamma^*$. Let $\bar{\lambda} = \frac{1}{2}(\lambda_1 + \lambda_2)$ Then,

$$\min_{\lambda \in \mathbf{\Delta}} \max_{z \neq z_*} \frac{\|z_* - z\|^2_{A(\lambda)^{-1}}}{\Delta_z^2}$$

$$\leq \max_{z \neq z_*} \frac{\|z_* - z\|^2_{A(\bar{\lambda})^{-1}}}{\Delta_z^2}$$

$$\leq 4(\max_{z \neq z_*} \left\| \frac{z_* - z}{\Delta_z} - \frac{z_* - z_1}{\Delta_{z_1}} \right\|^2_{A(\bar{\lambda})^{-1}} + \frac{\|z_* - z_1\|^2_{A(\bar{\lambda})^{-1}}}{\Delta_{z_1}^2})$$

$$\leq 4(\frac{\pi}{2} \mathbb{E}_{\eta \sim N(0,I)} \max_{z,z' \in \mathcal{Z} \setminus \{z_*\}} (\frac{z_* - z}{\Delta_z} - \frac{z_* - z_1}{\Delta_{z_1}})^\top A(\bar{\lambda})^{-1/2} \eta]^2 \qquad (61)$$

$$+ \frac{\|z_* - z_1\|^2_{A(\bar{\lambda})^{-1}}}{\Delta_{z_1}^2})$$

$$= 4(2\pi \mathbb{E}_{\eta \sim N(0,I)} \max_{z \in \mathcal{Z} \setminus \{z_*\}} (\frac{z_* - z}{\Delta_z})^\top A(\bar{\lambda})^{-1/2} \eta]^2 + \frac{\|z_* - z_1\|^2_{A(\bar{\lambda})^{-1}}}{\Delta_{z_1}^2})$$

$$\leq 8(2\pi \mathbb{E}_{\eta \sim N(0,I)} \max_{z \in \mathcal{Z} \setminus \{z_*\}} (\frac{z_* - z}{\Delta_z})^\top A(\lambda_2)^{-1/2} \eta]^2 + \frac{\|z_* - z_1\|^2_{A(\lambda_1)^{-1}}}{\Delta_{z_1}^2})$$

$$\qquad (62)$$

$$\leq 8(2\pi \inf_{\lambda \in \mathbf{\Delta}} \mathbb{E}_{\eta \sim N(0,I)} \max_{z \in \mathcal{Z} \setminus \{z_*\}} (\frac{z_* - z}{\Delta_z})^\top A(\lambda_2)^{-1/2} \eta]^2$$

$$+ \inf_{z \neq z_*} \inf_{\lambda \in \mathbf{\Delta}} \frac{\|z_* - z\|^2_{A(\lambda)^{-1}}}{\Delta_z^2} + \xi).$$

where line (61) follows by Lemma 12 and line (62) follows by the Sudakov-Fernique inequality (Theorem 7.2.11 of [32]) since $A(\bar{\lambda})^{-1} \preceq 2A(\lambda_2)^{-1}$. Since $\xi > 0$ is arbitrary, sending $\xi \longrightarrow 0$ yields the lower bound. $\qquad \square$

*Proof of Proposition 4.* Recall the definition $B(z,r) = \{z' \in \mathcal{Z} : \|z - z'\|_2 = r\}$. Let $\lambda_i = \frac{\varphi_i}{\varphi^*}$. Further, define

$$A_i = \{j \in [d] : \log(d|B(z_*, j)|) \in [2^{i-1}, 2^i]\}.$$

Let $v > 0$ a constant to be chosen later. Then,

$$\sqrt{\gamma^*} \leq \mathbb{E}_{\eta \sim N(0,I)}[\max_{z \in \mathcal{Z} \setminus z_*} \frac{\sum_{i \in z_* \Delta z} \frac{1}{\sqrt{\lambda_i}} \eta_i}{\Delta_z}]$$

$$= \frac{1}{v} \mathbb{E}_{\eta \sim N(0,I)}[\max_{z \in \mathcal{Z} \setminus z_*} v \frac{\sum_{i \in z_* \Delta z} \frac{1}{\sqrt{\lambda_i}} \eta_i}{\Delta_z}]$$

$$= \frac{\sqrt{\varphi^*}}{v} \mathbb{E}_{\eta \sim N(0,I)}[\log(\max_{z \in \mathcal{Z} \setminus z_*} \exp(v \frac{\sum_{i \in z_* \Delta z} \frac{1}{\sqrt{\varphi_i}} \eta_i}{\Delta_z}))]$$

$$\leq \frac{\sqrt{\varphi^*}}{v} \log(\mathbb{E}_{\eta \sim N(0,I)}[\max_{z \in \mathcal{Z} \setminus z_*} \exp(v \frac{\sum_{i \in z_* \Delta z} \frac{1}{\sqrt{\varphi_i}} \eta_i}{\Delta_z})]) \qquad (63)$$

$$= \frac{\sqrt{\varphi^*}}{v} \log(\mathbb{E}_{\eta}[\max_{i \in [4\log(d)]} \max_{z \neq z_*, |z \Delta z_*| \in A_i} \exp(v \frac{\sum_{i \in z_* \Delta z} \frac{1}{\sqrt{\varphi_i}} \eta_i}{\Delta_z})]) \qquad (64)$$

$$\leq \frac{\sqrt{\varphi^*}}{v} \log(\sum_{i \in [4\log(d)]} \mathbb{E}_{\eta}[\max_{z \neq z_*, |z \Delta z_*| \in A_i} \exp(v \frac{\sum_{i \in z_* \Delta z} \frac{1}{\sqrt{\varphi_i}} \eta_i}{\Delta_z})]) \qquad (65)$$

$$\leq \frac{\sqrt{\varphi^*}}{v} \log(4\log(d) \max_{i \in [4\log(d)]} \mathbb{E}_{\eta}[\max_{z \neq z_*, |z \Delta z_*| \in A_i} \exp(v \frac{\sum_{i \in z_* \Delta z} \frac{1}{\sqrt{\varphi_i}} \eta_i}{\Delta_z})]) \qquad (66)$$

where line (63) follows by Jensen's inequality, where line (64) follows by the definition of $A_i$, and line (65) follows since the max is upper bounded by the sum.

Notice that line (66) contains the moment generating function of a Gaussian random variable. We upper bound its variance as follows. Suppose $|z_*\Delta z| \in A_i$. Then,

$$\mathbb{V}(\frac{\sum_{i\in z_*\Delta z}\frac{1}{\sqrt{\varphi_i}}\eta_i}{\Delta_z})$$

$$= \mathbb{V}(\frac{\sum_{i\in z_*\Delta z}\min_{z':i\in z_*\Delta z'}\frac{\Delta_{z'}}{\sqrt{|z_*\Delta z'|\log(d|B(z_*,|z_*\Delta z'|)|)}}\eta_i}{\Delta_z}) \tag{67}$$

$$= \sum_{i\in z_*\Delta z}\frac{\min_{z':i\in z_*\Delta z'}\frac{\Delta_{z'}^2}{|z_*\Delta z'|\log(d|B(z_*,|z_*\Delta z'|)|)}}{\Delta_z^2} \tag{68}$$

$$= \frac{1}{|z_*\Delta z|\log(d|B(z_*,|z_*\Delta z|)|)}\sum_{i\in z_*\Delta z}\frac{\min_{z':i\in z_*\Delta z'}\frac{\Delta_{z'}^2}{|z_*\Delta z'|\log(d|B(z_*,|z_*\Delta z'|)|)}}{\frac{\Delta_z^2}{|z_*\Delta z|\log(d|B(z_*,|z_*\Delta z|)|)}}$$

$$\leq \frac{1}{\log(d|B(z_*,|z_*\Delta z|)|)}$$

$$\leq \frac{1}{2^{i-1}} \tag{69}$$

where line (67) follows by the definition of $\varphi_i$, line (68) follows since $\eta \sim N(0,I)$, and line (69) follows since $|z_*\Delta z| \in A_i$. Now, continuing and using this upper bound on the variance, we have

$$\frac{\sqrt{\varphi^*}}{v}\log(4\log(d)\max_{i\in[4\log(d)]}\mathbb{E}_\eta[\max_{z\neq z_*,|z\Delta z_*|\in A_i}\exp(v\frac{\sum_{i\in z_*\Delta z}\frac{1}{\sqrt{\varphi_i}}\eta_i}{\Delta_z})])$$

$$\leq \frac{\sqrt{\varphi^*}}{v}\log(4\log(d)\max_{i\in[4\log(d)]}|\cup_{j\in A_i}B(z_*,j)|\exp(v^2\frac{c}{2^{i+1}})) \tag{70}$$

$$= \max_{i\in[4\log(d)]}\sqrt{\varphi^*}[\log(4\log(d))/v + \log(|\cup_{j\in A_i}B(z_*,j)|)/v + v\frac{c}{2^{i+1}}]$$

$$= \max_{i\in[4\log(d)]}\sqrt{\varphi^*\log(4\log(d))\log(|\cup_{j\in A_i}B(z_*,j)|)\frac{c}{2^{i+1}}} \tag{71}$$

$$\leq \max_{i\in[4\log(d)]}\sqrt{\varphi^*\log(4\log(d))\max_{j\in A_i}\log(|A_i|B(z_*,j)|)\frac{c}{2^{i+1}}}$$

$$\leq \max_{i\in[4\log(d)]}\sqrt{\varphi^*\log(4\log(d))\max_{j\in A_i}\log(d|B(z_*,j)|)\frac{c}{2^{i+1}}} \tag{72}$$

$$\leq c\sqrt{\varphi^*\log(\log(d))} \tag{73}$$

where (70) follows by Lemma 17 and $\{z \in \mathcal{Z} : z \neq z_*, |z\Delta z_*| \in A_i\} \subset \cup_{j\in A_i}B(z_*,j)$, line (71) follows by maximizing the constant $v$, (72) follows since $|A_i| \leq d$, and line (73) follows by definition of $A_i$.

$\square$

*Proof of Proposition 3.* Define the allocation

$$\lambda_i \propto \tilde{\Delta}_i^{-2}$$

where

$$\tilde{\Delta}_i = \begin{cases} \theta^\top z_* - \max_{z\in\mathcal{Z}:i\in z}\theta^\top z & i \notin z_* \\ \theta^\top z_* - \max_{z\in\mathcal{Z}:i\notin z}\theta^\top z & i \in z_* \end{cases}$$

Then,

$$\gamma^* \leq \mathbb{E}_{\eta \sim N(0,I)}[\max_{z \in \mathcal{Z} \setminus z_*} \frac{\sum_{i \in z_* \Delta z} \frac{1}{\sqrt{\lambda_i}} \eta_i}{\sum_{i \in z_* \setminus z} \theta_i - \sum_{j \in z \setminus z_*} \theta_j}]^2$$

$$= \sum_{i=1}^{d} \tilde{\Delta}_i^{-2} \mathbb{E}_{\eta \sim N(0,I)}[\max_{z \in \mathcal{Z} \setminus z_*} \frac{\sum_{i \in z_* \Delta z} \tilde{\Delta}_i \eta_i}{\sum_{i \in z_* \setminus z} \theta_i - \sum_{j \in z \setminus z_*} \theta_j}]^2$$

$$= \sum_{i=1}^{d} \tilde{\Delta}_i^{-2} \mathbb{E}_{\eta \sim N(0,I)}[\max_{z \in \mathcal{Z} \setminus z_*} \frac{\sum_{i \in z_* \setminus z} \tilde{\Delta}_i \eta_i + \sum_{i \in z \setminus z_*} \tilde{\Delta}_i \eta_i}{\sum_{i \in z_* \setminus z} \theta_i - \sum_{j \in z \setminus z_*} \theta_j}]^2$$

$$= \sum_{i=1}^{d} \tilde{\Delta}_i^{-2} \mathbb{E}_{\eta \sim N(0,I)}[\max_{z \in \mathcal{Z} \setminus z_*} \frac{v_z^\top \eta + w_z^\top \eta}{\sum_{i \in z_* \setminus z} \theta_i - \sum_{j \in z \setminus z_*} \theta_j}]^2$$

where we defined the vectors

$$(v_z)_i = \frac{\theta^\top z_* - \max_{z \in \mathcal{Z}: i \notin z} \theta^\top z}{\sum_{j \in z_* \setminus z} \theta_j - \sum_{j \in z \setminus z_*} \theta_j} \mathbf{1}\{i \in z_* \setminus z\}$$

$$(w_z)_i = \frac{\theta^\top z_* - \max_{z \in \mathcal{Z}: i \in z} \theta^\top z}{\sum_{j \in z_* \setminus z} \theta_j - \sum_{j \in z \setminus z_*} \theta_j} \mathbf{1}\{i \in z \setminus z_*\}$$

It remains to bound the expected suprema. Suppose wlog $z_* = \{1, \ldots, r\}$. By Lemma 10, there exists a bijection $\sigma : z_* \longrightarrow z$ such that for every $i \in z_*$, $z^{(i)} := (z_* \setminus \{i\}) \cup \{\sigma(i)\} \in \mathcal{Z}$. Note that

$$\theta^\top z_* - \max_{z \in \mathcal{Z}: i \notin z} \theta^\top z \leq \theta^\top (z_* - z^{(i)}) = \theta_i - \theta_{\sigma(i)}.$$

Therefore,

$$\|v_z\|_1 = \sum_{i \in z_* \setminus z} \frac{|\theta^\top z_* - \max_{z \in \mathcal{Z}: i \notin z} \theta^\top z|}{\sum_{j \in z_* \setminus z} \theta_j - \sum_{j \in z \setminus z_*} \theta_j}$$

$$\leq \sum_{i \in z_* \setminus z} \frac{\theta_i - \theta_{\sigma(i)}}{\sum_{j \in z_* \setminus z} \theta_j - \sum_{j \in z \setminus z_*} \theta_j}$$

$$\leq 1.$$

A similar argument show that $\|w_z\|_1 \leq 1$. Thus,

$$\mathbb{E}_{\eta \sim N(0,I)}[\max_{z \subset \mathcal{Z} \setminus z_*} v_z^\top \eta + w_z^\top \eta]^2 \leq \mathbb{E}_{\eta \sim N(0,I)}[\max_{v: \|v\|_1 \leq 1} v^\top \eta + \max_{w: \|w\|_1 \leq 1} w^\top \eta]^2$$

$$\leq c \log(d)$$

where in the final inequality we used Example 7.5.9 of [32].

$\square$

The following Lemma appears as Corollary 3 in [3].

**Lemma 10.** *Given two bases $B_1$ and $B_2$ of a matroid $\mathcal{M} = (E, I)$, there exists a bijection $\sigma : B_1 \longrightarrow B_2$ such that $(B_2 \setminus \sigma(e)) \cup e \in I$ for all $e \in I$.*

## G  Additional Lower Bounds

In this section, we show that in several common situations $\Omega(d)$ samples are required. The following Theorem applies to combinatorial bandits and implies Theorem 5.

**Theorem 9.** *Let $\delta \in (0, 1/4)$. Consider the combinatorial bandit setting. Fix $\theta \in \Theta$ such that there is a unique best arm. Suppose $\Theta$ satisfies the following property: for all $i \in [d]$*

$$(\theta + e_i \cdot \min_{i \in z_* \setminus z} \Delta_z \in \Theta) \text{ or } (\theta - e_i \cdot \min_{i \in z \setminus z_*} \Delta_z \in \Theta) \text{ is true.}$$

*If an algorithm $\mathcal{A}$ is $\delta$-pac wrt $(\mathcal{X}, \mathcal{Z}, \Theta)$, then*

$$\mathbb{E}_\theta[\sum_{i=1}^d T_i] \geq \frac{d}{2}.$$

*where $T_i$ denotes the number of times that $\mathcal{A}$ pulls $e_i$.*

**Remark 1.** *Note that if $\Theta = \mathbb{R}^d$, then $\Theta$ satisfies the condition in the above Theorem.*

*Proof of Theorem 9.* Without loss of generality, suppose $1 = \arg\max_i \theta^\top z_i$. Towards a contradiction, suppose there is some arm $i$ such that $\mathbb{E}_\theta[T_i] \leq \frac{1}{2}$. Let $z_j$ such that $i \in z_j \Delta z_1$ and suppose that $i \in z_j \setminus z_1$ (the other case is similar). Define

$$\tilde{\theta}_k = \begin{cases} \theta_k & \text{if } k \neq i \\ \theta_i + 2\theta^\top(z_1 - z_j) & \text{if } k = i. \end{cases}$$

Note that $(z_1 - z_j)^\top \tilde{\theta} < 0$. Observe that

$$\frac{1}{2} \geq \mathbb{E}_\theta[T_i] \geq \mathbb{P}_\theta(T_i > 0).$$

Define the event $A = \{T_i = 0\} \cap \{I = 1\}$, where $I$ denotes the index of the set output by $\mathcal{A}$ as its answer for the best set. Note that

$$\mathbb{P}_\theta(A^c) \leq \mathbb{P}_\theta(T_i > 0) + \mathbb{P}_\theta(I \neq 1) \leq \frac{1}{2} + \delta \leq \frac{3}{4}$$

so that $\mathbb{P}_\theta(A) \geq \frac{1}{4}$.

Define

$$\widehat{\mathrm{kl}}_{i,T_i} = \sum_{s=1}^{T_i} \log(f_\theta(Z_s)/f_{\tilde{\theta}}(Z_s))$$

where $Z_s$ is the observation on the $s$th pull of $e_i$, $f_\theta$ denotes the density of the distribution associated with $e_i \in \mathcal{X}$ under $\theta$, and $f_{\tilde{\theta}}$ denotes the density of the distribution associated with $e_i \in \mathcal{X}$ under $\tilde{\theta}$. Then, by the change of measure identity (Lemma 18) from [25],

$$\begin{aligned} \mathbb{P}_{\tilde{\theta}}(I = 1) &\geq \mathbb{P}_{\tilde{\theta}}(A) \\ &= \mathbb{E}_\theta[\mathbb{1}\{A\} \exp(-T_i \widehat{\mathrm{kl}}_{i,T_i})] \\ &= \mathbb{P}_\theta(A) \\ &\geq \frac{1}{4}. \end{aligned}$$

where we used the fact that the only difference between problem $\theta$ and problem $\tilde{\theta}$ is the $i$th arm and on the event $A$, $T_i = 0$. Thus, on problem instance $(\mathcal{Z}, \tilde{\theta})$, $\mathcal{A}$ gives the incorrect answer with probability $1/4 > \delta$, which is a contradiction.

$\square$

The following Theorem gives a lower bound for best arm identification in linear bandits.

**Theorem 10.** *Let $\delta \in (0,1)$. Let $\mathcal{X} \subset \mathbb{R}^d$, such that $\|x_i\|_2 \leq 1$ for all $i \in [|\mathcal{X}|]$, $\mathcal{Z} = \mathcal{X}$, and $\Theta = \mathbb{R}^d$. Fix $\theta \in \Theta$ such that there is a unique best arm and let $x_1 = \arg\max_i \theta^\top x_i$. If an algorithm $\mathcal{A}$ is $\delta$-pac wrt $(\mathcal{X}, \mathcal{Z}, \Theta)$ and $d \geq 3$, then*

$$\mathbb{E}_\theta[\sum_{x \in \mathcal{X}} T_x] \geq c \log(\frac{1}{2.4}\delta) \min_{i \neq 1} \frac{d}{\theta^\top(x_1 - x_i)^2}$$

*where $T_x$ denotes the number of times that $\mathcal{A}$ pulls $x \in \mathcal{X}$.*

*Proof of Theorem 10.* By Theorem 1 of [12], we have that

$$\mathbb{E}_\theta[\sum_{x\in\mathcal{X}} T_x] \geq \log(\frac{1}{2.4}\delta)c\rho^*$$

so it suffices to lower bound $\rho^*$.

Since

$$\rho^* = \min_\lambda \max_{i\neq 1}\frac{\|x_1 - x_i\|^2_{A(\lambda)^{-1}}}{\theta^\top(x_1 - x_i)^2} \geq [\min_{j\neq 1}\frac{cd}{\theta^\top(x_1 - x_j)^2}]\min_\lambda \max_{i\neq 1}\|x_1 - x_i\|^2_{A(\lambda)^{-1}},$$

it suffices to show that

$$\min_\lambda \max_{i\neq 1}\|x_1 - x_i\|^2_{A(\lambda)^{-1}} \geq cd.$$

Let

$$\lambda^* = \operatorname{argmin}_\lambda \max_{i\neq 1}\|x_1 - x_i\|_{A(\lambda)^{-1}}.$$

Then,

$$\begin{aligned}
\sqrt{d} &= \min_\lambda \max_{i\in[|\mathcal{X}|]}\|x_i\|_{A(\lambda)^{-1}} \\
&\leq \min_\lambda \max_{i\neq 1}\|x_1 - x_i\|_{A(\lambda)^{-1}} + \|x_1\|_{A(\lambda)^{-1}} \\
&\leq \max_{i\neq 1}\|x_1 - x_i\|_{[\frac{1}{2}A(\lambda^*)+\frac{1}{2}x_1 x_1^\top]^{-1}} + \|x_1\|_{[\frac{1}{2}A(\lambda^*)+\frac{1}{2}x_1 x_1^\top]^{-1}} \\
&\leq \sqrt{2}\max_{i\neq 1}\|x_1 - x_i\|_{A(\lambda^*)^{-1}} + \sqrt{2}\|x_1\|_{(x_1 x_1^\top)^+} \\
&= \sqrt{2}\max_{i\neq 1}\|x_1 - x_i\|_{A(\lambda^*)^{-1}} + \sqrt{2}.
\end{aligned} \tag{74}$$

The first line follows by Keifer-Wolfowitz (Theorem 21.1 in [26]). The second to last inequality follows because

$$\frac{1}{2}A(\lambda^*) + \frac{1}{2}x_1 x_1^\top \succeq \frac{1}{2}A(\lambda^*)$$

which implies

$$(\frac{1}{2}A(\lambda^*))^{-1} \succeq (\frac{1}{2}A(\lambda^*) + \frac{1}{2}x_1 x_1^\top)^{-1}.$$

Also, since $x_1 \in \operatorname{span}(x_1)$, the same fact implies that

$$\|x_1\|_{(\frac{1}{2}x_1 x_1^\top)^+} \geq \|x_1\|_{[\frac{1}{2}A(\lambda^*)+\frac{1}{2}x_1 x_1^\top]^{-1}}.$$

Rearranging the inequality (74), we obtain

$$2(\sqrt{d} - \sqrt{2})^2 \leq \min_\lambda \max_{i\neq 1}\|x_1 - x_i\|^2_{A(\lambda)^{-1}}$$

and thus the result follows.

$\square$

# H   Rounding

In this Section, we justify the application of the rounding procedure from [1]. Define

$$S_N = \{v \in \mathbb{N}^{|X|} : \sum_{i=1}^{|\mathcal{X}|} v_i \leq N\}$$

$$C_N = \{v \in [0, N]^{|X|} : \sum_{i=1}^{|\mathcal{X}|} v_i \leq N\}$$

The following Theorem appears in [1].

**Theorem 11.** *Let $F : \mathbb{S}_d^+ \longrightarrow \mathbb{R}$ such that*

- *For any $A, B \in \mathbb{S}_d^+$, if $A \preceq B$, then $F(A) \geq F(B)$,*

- *for any $A \in \mathbb{S}_d^+$ and $t \in (0,1)$, $F(tA) = t^{-1}F(A)$.*

*Let $\epsilon \in (0, 1/6]$. Then, if $|\mathcal{X}| \geq N \geq 5\frac{d}{\epsilon^2}$, for any $\pi \in C_N$, there exists an algorithm that in $\tilde{O}(|\mathcal{X}|d^2)$ time rounds $\pi$ to $\kappa \in S_N$ such that*

$$F(A(\kappa)) \leq (1 + 6\epsilon)F(A(\pi)).$$

The following result shows that the optimization problem

**Lemma 11.** *Fix $V \subset \mathbb{R}^d$. Define the functions $F, G : \mathbb{S}_d^+ \longrightarrow \mathbb{R}$*

$$F(A) = \mathbb{E}_{\eta \sim N(0,I)}[max_{v \in V} v^\top A^{-1/2}\eta]^2$$
$$G(A) = max_{v \in V} v^\top A v.$$

*$F$ and $G$ satisfy the conditions of Theorem 11.*

*Proof.* It is trivial to see that $G$ satisfies the conditions of Theorem 11. Thus, we focus on the function $F$. Let $A, B \in \mathbb{S}_d^+$ such that $A \preceq B$. Then, $A^{-1} \succeq B^{-1}$. Fix $v, w \in V$. Then,

$$\mathbb{E}[(v - w)^\top A^{-1/2}\eta]^2 = \|v - w\|_{A^{-1}} \leq \|v - w\|_{B^{-1}} = \mathbb{E}[(v - w)^\top B^{-1/2}\eta]^2$$

Then, by Sudakov-Fernique inequality (Theorem 7.2.11 in [32]), it follows that $F(A) \geq F(B)$.

The second condition is trivial. $\qquad\square$

# I    Technical Lemmas related to $\gamma^*$

In this Section, we state and prove several useful technical lemmas.

**Lemma 12.** *Let $S \subset \mathcal{Z}$. Then,*

$$\mathbb{E}_{\eta \sim N(0,I)}[max_{z,z' \in S}[A(\lambda)^{-1/2}(z - z')]^\top \eta]^2 \geq \frac{2}{\pi}max_{z,z' \in S}\|z - z'\|_{A(\lambda)^{-1}}^2.$$

*Proof.* Fix $z_1, z_2 \in S$. Then,

$$\mathbb{E}_{\eta \sim N(0,I)}[\max_{z,z' \in S}[A(\lambda)^{-1/2}(z - z')]^\top \eta] \geq \mathbb{E}_{\eta \sim N(0,I)}\left[|A(\lambda)^{-1/2}(z_1 - z_2)]^\top \eta|\right]$$

$$= \|z_1 - z_2\|\sqrt{\frac{2}{\pi}}$$

$\qquad\square$

**Lemma 13.** *Let $\alpha > 0$ be a constant. Then,*

$$\inf_\lambda \mathbb{E}_{\eta \sim N(0,I)}[max_{z \in \mathcal{Z}\setminus\{z_*\}} \frac{(z_* - z)^\top A(\lambda)^{-1/2}\eta}{\theta^\top(z_* - z)}]^2 + max_{z \in \mathcal{Z}\setminus\{z_*\}} \frac{\|z_* - z\|_{A(\lambda)^{-1}}^2}{\theta^\top(z_* - z)^2}\alpha$$

$$\leq c[\gamma^* + \rho^*\alpha]$$

*Proof.* Let $\lambda_1$ denote the solution to $\gamma^*$ and $\lambda_2$ the solution to $\rho^*$. Define $\lambda = \frac{1}{2}(\lambda_1 + \lambda_2)$. It suffices to show that

$$\mathbb{E}_{\eta \sim N(0,I)}[\max_{z \in \mathcal{Z}\setminus\{z_*\}} \frac{(z_* - z)^\top A(\lambda)^{-1/2}\eta}{\theta^\top(z_* - z)}]^2$$

$$\leq c\mathbb{E}_{\eta \sim N(0,I)}[\max_{z \in \mathcal{Z}\setminus\{z_*\}} \frac{(z_* - z)^\top A(\lambda_1)^{-1/2}\eta}{\theta^\top(z_* - z)}]^2$$

and

$$\max_{z \in \mathcal{Z} \setminus \{z_*\}} \frac{\|z_* - z\|^2_{A(\lambda)^{-1}}}{\theta^\top (z_* - z)^2} \leq c \max_{z \in \mathcal{Z} \setminus \{z_*\}} \frac{\|z_* - z\|^2_{A(\lambda_2)^{-1}}}{\theta^\top (z_* - z)^2}.$$

Note that

$$1/2 \sum_{x \in \mathcal{X}} (\lambda_{1,x} + \lambda_{2,x}) x x^\top \succeq 1/2 \sum_{x \in \mathcal{X}} \lambda_{i,x} x x^\top$$

for $i = 1, 2$. Therefore,

$$2A(\lambda_i)^{-1} \geq A(\lambda)^{-1} \tag{75}$$

for $i = 1, 2$.

(75) immediately implies

$$\max_{z \in \mathcal{Z} \setminus \{z_*\}} \frac{\|z_* - z\|^2_{A(\lambda)^{-1}}}{\theta^\top (z_* - z)^2} \leq c \max_{z \in \mathcal{Z} \setminus \{z_*\}} \frac{\|z_* - z\|^2_{A(\lambda_2)^{-1}}}{\theta^\top (z_* - z)^2}.$$

(75) implies via Sudakov-Fernique inequality (Theorem 7.2.11 in [32]) that

$$\mathbb{E}_{\eta \sim N(0,I)} [\max_{z \in \mathcal{Z} \setminus \{z_*\}} \frac{(z_* - z)^\top A(\lambda)^{-1/2} \eta}{\theta^\top (z_* - z)}]^2$$

$$\leq c \mathbb{E}_{\eta \sim N(0,I)} [\max_{z \in \mathcal{Z} \setminus \{z_*\}} \frac{(z_* - z)^\top A(\lambda_1)^{-1/2} \eta}{\theta^\top (z_* - z)}]^2.$$

$\square$

**Lemma 14.** *Let $V = \{v_1, \ldots, v_l\} \subset \mathbb{R}^d$ and suppose $0 \in V$. Let $a_i \geq 1$ for all $i$. Then,*

$$\mathbb{E}_{\eta \sim N(0,I)} \sup_{v_i \in V} v_i^\top \eta \leq \mathbb{E}_{\eta \sim N(0,I)} \sup_{v_i \in V} a_i v_i^\top \eta$$

*Proof.* Fix $\eta \in \mathbb{R}^d$. Then, clearly,

$$\sup_{v_i \in V} v_i^\top \eta \leq \sup_{v_i \in V} a_i v_i^\top \eta.$$

Taking the expectation wrt $\eta \sim N(0, I)$ yields the result. $\square$

**Lemma 15.** *Fix $V \subset \mathbb{R}^d$. Then,*

$$\mathbb{E}_{\eta \sim N(0,I)} \sup_{v \in V} v^\top \eta \geq 0.$$

*Proof.* Fix $v_0 \in V$. Then,

$$\mathbb{E}_{\eta \sim N(0,I)} \sup_{v \in V} v^\top \eta \geq \mathbb{E}_{\eta \sim N(0,I)} v_0^\top \eta = 0.$$

$\square$

**Lemma 16.** *Let $V \subset \mathbb{R}^d$ and suppose $0 \in V$. Fix $v_0 \in V$. Then,*

$$\mathbb{E} \sup_{v \in V} v^\top \eta \leq 2(\|v_0\|_2 + \mathbb{E} \sup_{v \in V \setminus \{0\}} v^\top g)$$

*Proof.*

$$\mathbb{E} \sup_{v \in V} v^\top \eta \leq \mathbb{E} \sup_{v \in V \setminus \{0\}} |v^\top \eta| \leq 2(\|v_0\|_2 + \mathbb{E} \sup_{v \in V \setminus \{0\}} v^\top g)$$

where the last inequality follows by exercise 7.6.9 of [32].

$\square$

**Lemma 17.** *Consider a sub-Gaussian random process $X_t$ indexed by $t \in \mathcal{T}$ such that for any $\nu$ we have $\mathbb{E}[\exp(\nu X_t)] \leq \exp(\nu^2 \sigma_t^2/2)$. Then $\mathbb{E}\left[\sup_{t\in\mathcal{T}} X_t\right] \leq \sqrt{2\sup_{t\in\mathcal{T}} \sigma_t^2 \log(|\mathcal{T}|)}$.*

*Proof.*

$$
\begin{aligned}
\mathbb{E}\left[\sup_{t\in\mathcal{T}} X_t\right] &= \frac{1}{\nu}\mathbb{E}\left[\sup_{t\in\mathcal{T}} \nu X_t\right] \\
&= \frac{1}{\nu}\mathbb{E}\left[\log\left(\sup_{t\in\mathcal{T}} \exp(\nu X_t)\right)\right] \\
&\leq \frac{1}{\nu}\log\left(\mathbb{E}\left[\sup_{t\in\mathcal{T}} \exp(\nu X_t)\right]\right) \\
&\leq \frac{1}{\nu}\log\left(|\mathcal{T}|\sup_{t\in\mathcal{T}} \mathbb{E}[\exp(\nu X_t)]\right) \\
&\leq \frac{1}{\nu}\log\left(|\mathcal{T}|\sup_{t\in\mathcal{T}} \exp(\nu^2\sigma_t^2/2)\right) \\
&= \frac{1}{\nu}\log(|\mathcal{T}|) + \nu\sup_{t\in\mathcal{T}} \sigma_t^2/2 \\
&\leq \sqrt{2\sup_{t\in\mathcal{T}} \sigma_t^2 \log(|\mathcal{T}|)}
\end{aligned}
$$

$\square$

## J  Some Useful Results regarding Computational Efficiency

The following result shows that after a suitable monotonic transformation, the objective function in the optimization problems for finding a good allocation in Algorithms 1 and 2 is convex when $\mathcal{X} = \{e_1, \ldots, e_d\}$, which holds in the combinatorial bandit problem. We note that Lemma 15 shows that the gaussian width is nonnegative and thus it suffices consider the squareroot of the objective function.

**Proposition 7.** *Fix $V \subset \mathbb{R}^d$.*

$$
f(\lambda) = \mathbb{E}_{\eta\sim N(0,I)}[max_{v\in V} v^\top diag(\frac{1}{\lambda_i^{1/2}})\eta]
$$

*is convex.*

*Proof.* Fix $\lambda, \kappa \in \mathbb{S}^{n-1}$ and $\alpha \in [0,1]$. By matrix convexity,

$$
\text{diag}(\frac{1}{\alpha\lambda_i + (1-\alpha)\kappa_i})^{1/2} \preceq \alpha\text{diag}(\frac{1}{\lambda_i})^{1/2} + (1-\alpha)\text{diag}(\frac{1}{\kappa_i})^{1/2}.
$$

Furthermore, since the above matrices are diagonal,

$$
\text{diag}(\frac{1}{\alpha\lambda_i + (1-\alpha)\kappa_i}) \preceq (\alpha\text{diag}(\frac{1}{\lambda_i})^{1/2} + (1-\alpha)\text{diag}(\frac{1}{\kappa_i})^{1/2})^2.
$$

Then, by Sudakov-Fernique inequality (Theorem 7.2.11 [32]),

$$
\begin{aligned}
f(\alpha\lambda + (1-\alpha)\kappa) &= \mathbb{E}_{\eta\sim N(0,\text{diag}(\frac{1}{\alpha\lambda_i+(1-\alpha)\kappa_i}))}\sup_{v\in V} v^\top\eta \\
&\leq \mathbb{E}_{\eta\sim N(0,(\alpha\text{diag}(\frac{1}{\lambda_i})^{1/2}+(1-\alpha)\text{diag}(\frac{1}{\kappa_i})^{1/2})^2)}\sup_{v\in V} z^\top\eta \\
&= \mathbb{E}_{\eta\sim N(0,I)}\sup_{v\in V} v^\top(\alpha\text{diag}(\frac{1}{\lambda_i})^{1/2} + (1-\alpha)\text{diag}(\frac{1}{\kappa_i})^{1/2})\eta \\
&\leq \alpha\mathbb{E}_{\eta\sim N(0,I)}\sup_{v\in V} v^\top\text{diag}(\frac{1}{\lambda_i})^{1/2}\eta \\
&\quad + (1-\alpha)\mathbb{E}_{\eta\sim N(0,I)}\sup_{v\in V} v^\top\text{diag}(\frac{1}{\kappa_i})^{1/2}\eta \\
&= \alpha f(\lambda) + (1-\alpha)f(\kappa)
\end{aligned}
$$

□

# K   Comparison Results

In this Section, we prove various results related to the sample complexities proposed in other works. Recall the notation for the sphere $B(z,r) = \{z' \in \mathcal{Z} : \|z - z'\|_2 = r\}$.

*Proof of Proposition 5.* Define $\theta_1 = \ldots = \theta_k = 1/2$, $\theta_{k+1} = \ldots = \theta_{2k-1} = \frac{1}{2} - \frac{1}{k^{1/2}}$ and $\theta_{2k} = \ldots = \theta_d = 0$ and $d = k^2$. Define

$$\Delta_i = \begin{cases} \theta_i - \theta_{k+1} & : i \le k \\ \theta_k - \theta_i & : i > k \end{cases}$$

$$\bar{\lambda}_i = \frac{\Delta_i^{-2}}{\sum_{i=1}^d \Delta_i^{-2}}$$

Note that

$$\rho^* \le \sum_i \Delta_i^{-2} \max_{z \ne z_*} \frac{\sum_{i \in z_* \Delta z} \Delta_i^2}{\Delta_z^2}$$

$$\le c \sum_i \Delta_i^{-2}$$

$$\le c[k^2 + d]$$

$$\le c'd.$$

Consider arm $d$. We will show that $\varphi_d \ge ck\log(d)$. Fix $\tilde{z} = \{k+1, k+2, \ldots, 2k-1, d\}$ and $z_* = [k]$. It suffices to show that

$$\frac{\|z_* - \tilde{z}\|_1 \log(|B(z_*, |z_* \Delta \tilde{z}|)|)}{\theta^\top (z_* - \tilde{z})^2} \ge c\log(d)k,$$

from which the claim will follow. Note that

$$\frac{\|z_* - \tilde{z}\|_1}{\theta^\top (z_* - \tilde{z})^2} = \frac{2k}{(\frac{k-1}{\sqrt{k}} + \frac{1}{2})^2} \ge c.$$

Furthermore,

$$\log(|B(z_*, |z_* \Delta \tilde{z}|)|) \ge \log(\binom{d-2k}{k})$$

$$\ge \log(\frac{(d-2k)^k}{k!})$$

$$\ge k\log(\frac{d-2k}{k})$$

$$\ge k\log(k-2)$$

$$\ge \frac{1}{4}k\log(d)$$

where in the last inequality we used $d = k^2$. Thus, the claim follows and $\varphi_d \ge ck$. A similar argument applies to arms $\{2k, \ldots, d-1\}$ yielding the result. □

The following proposition shows that $\rho^*$ is lower bounded by the typical measure of hardness for top-k [25]. It implies that the sample complexity of [8, 12] is off by a factor of $k$.

**Proposition 8.** *Consider the top-k problem where* $\theta_1 \ge \ldots \theta_k > \theta_{k+1} \ge \ldots \ge \theta_n$.

$$\rho^* \ge \sum_{i \le k} \frac{1}{(\theta_i - \theta_{k+1})^2} + \sum_{i > k} \frac{1}{(\theta_k - \theta_i)^2}$$

*Proof.*

$$\rho^* = \inf_\lambda \max_{z \in \mathcal{Z} \setminus \{z_*\}} \frac{\|z_* - z\|^2_{A(\lambda)^{-1}}}{\theta^\top (z_* - z)^2}$$

$$= \inf_\lambda \max_{A \neq [k]} \frac{\sum_{i \in A \Delta [k]} \frac{1}{\lambda_i}}{(\sum_{i \in [k]} \theta_i - \sum_{i \in A} \theta_i)^2}$$

$$\geq \min_\lambda \max \left( \max_{i \in [k]} \frac{\frac{1}{\lambda_i} + \frac{1}{\lambda_{k+1}}}{(\theta_i - \theta_{k+1})^2}, \max_{i \in [d] \setminus [k]} \frac{\frac{1}{\lambda_i} + \frac{1}{\lambda_k}}{(\theta_k - \theta_i)^2} \right)$$

$$\geq \min_\lambda \max \left( \max_{i \in [k]} \frac{\frac{1}{\lambda_i}}{(\theta_i - \theta_{k+1})^2}, \max_{i \in [d] \setminus [k]} \frac{\frac{1}{\lambda_i}}{(\theta_k - \theta_i)^2} \right)$$

To minimize the RHS, we set it to a constant $c$. Then,

$$\lambda_i = \begin{cases} \frac{1}{c(\theta_i - \theta_{k+1})^2} & i \leq k \\ \frac{1}{c(\theta_i - \theta_k)^2} & \text{otherwise} \end{cases}$$

Then, the result follows from the below and solving for $c$.

$$1 = \sum_{i=1} \lambda_i = \frac{1}{c} \left[ \sum_{i \leq k} \frac{1}{(\theta_i - \theta_{k+1})^2} + \sum_{i > k} \frac{1}{(\theta_k - \theta_i)^2} \right].$$

$\square$

The following gives an instance where $|\mathcal{Z}|$ is linear in the dimension $d$, but $\varphi^*$ is loose by a $\sqrt{d}$ factor.

**Proposition 9.** *Consider the combinatorial bandit setting. There exists a problem where $|\mathcal{Z}|$ is linear in the dimension $d$ and $\varphi^* \geq c\rho^* \sqrt{d}$.*

*Proof.* Fix $k < d$. Define $z_1 = [k], z_2 = \{k+1\}, z_3 = \{k+3\}, \ldots, z_{d-k} = \{d\}$ and let $\mathcal{Z} = \{z_1, \ldots, z_{d-k}\}$. Note $|\mathcal{Z}| \leq d$ and thus satisfies the hypothesis. Fix $\epsilon > 0$ and let

$$\theta_i = \begin{cases} \epsilon & i \leq k \\ 0 & \text{otherwise} \end{cases}.$$

Then, $z_* = z_1$. The upper bound guarantee of [5, 20] is at least

$$\sum_{i=1}^k \max_{z \neq z_*} \frac{|z_* \Delta z|}{\theta^\top (z_* - z)^2} + \sum_{i=k+1}^d \frac{|z_* \Delta z_i|}{\theta^\top (z_* - z_i)^2} = d \frac{k+1}{(k\epsilon)^2}$$

$$\geq \frac{d}{k\epsilon^2}. \tag{76}$$

On the other hand, we have that

$$\rho^* = \max_{z \neq z_*} \frac{\|z_* - z\|^2_{A(\lambda)^{-1}}}{\theta^\top (z_* - z)^2} \leq 2 \left[ \frac{k^2 + d}{(k\epsilon)^2} \right.$$

$$\leq 2 \left[ \frac{1}{\epsilon^2} + \frac{d}{(k\epsilon)^2} \right] \tag{77}$$

where we took

$$\lambda_i = \begin{cases} \frac{1}{2k} + \frac{1}{2d} & i \leq k \\ \frac{1}{2d} & \text{otherwise} \end{cases}.$$

Putting $k = \sqrt{d}$ into (76) and (77) yields the result. $\square$

In the matching problem, if $\theta = \mathbb{1}\{i \in z\}\Delta$ for some $z \in Z$ and $\Delta > 0$, we say that it is an instance of HOMOGENOUS MATCHING. The following result appears in [5]. It shows that the sample complexity of [5, 20] is correct for the homogeneous matching problem.

**Proposition 10.** *Consider the homogenous* MATCHING *problem. Then,* $\rho^* = \Theta(d/\Delta^2)$. *Further, letting*

$$\varphi_i = max_{z \in \mathcal{Z} \setminus \{z_*\}: i \in z_* \Delta z} \frac{|z_* \Delta z| \log(|B(z_*, |z_* \Delta z|)|)}{\Delta_z^2}$$

*we have that* $\sum_{i=1}^n \varphi_i = O(d/\Delta^2)$.

**Remark 2.** *It follows from Proposition 4 that for the homogenous* MATCHING *problem,* $\gamma^* \leq O(\log(\log(d))d/\Delta^2)$

The following result appears in [8]. It shows that there is a gap of order $d$ between the sample complexities in [10] and [13] and the lower bound.

**Proposition 11.** *Let $d$ be even. Consider the combinatorial bandit setting where $\mathcal{X} = \{e_1, \ldots, e_d\}$ and $\mathcal{Z} = \{[d/2], \{d/2 + 1, \ldots, d\}$ and $\theta_i = \epsilon \mathbb{1}\{i \leq d/2\}$. Then, the guarantee of the CLUCB in [10] and the algorithm in [13] is $\Omega(d\epsilon^{-2} \ln(1/d))$. On the other hand, $\rho^* = \epsilon^{-2}$.*

The following result shows that the sample complexity cannot depend on $\log(\mathcal{Z})$ because $|\mathcal{Z}|$ can be arbitrarily large while $\gamma^* \leq 1$.

**Proposition 12.** *For any $N \in \mathbb{N}$, there exists an instance of the transductive linear bandit problem where $|\mathcal{Z}| \geq N$ and $\gamma^* \leq 1$.*

*Proof of Proposition 12.* Let $\mathcal{X} = \{e_1, \ldots, e_d\}$. Let $\theta = ae_1$ for a constant $a > 0$ to be chosen later. Let $z_1 = e_1$. Let $z_2, \ldots, z_N$ such that for every $i$ $e_1^\top z_i = 0$ and $\|z_i\|_2 = 1$. Then, $\Delta_i := \theta^\top(z_1 - z) = a$ for all $i$ and some $\Delta > 0$. Then, by Proposition 7.5.2 of [32], we have that

$$\inf_{\lambda \in \mathbf{\Delta}} \mathbb{E}[\sup_{i>1} \frac{(z_* - z_i)^\top A(\lambda)^{-1/2} \eta}{\Delta_i}] = \frac{1}{a} \inf_{\lambda \in \mathbf{\Delta}} \mathbb{E}[\sup_{i>1}(z_* - z_i)^\top A(\lambda)^{-1/2} \eta]$$

$$\leq \frac{\sqrt{d}}{a} max_{i>1} \left\| A(\lambda)^{-1/2}(z_* - z_i) \right\|_2$$

$$\leq \frac{d}{a} max_{i>1} \|z_* - z_i\|_2$$

$$= \frac{2d}{a}$$

$$\leq 1$$

for $a = 2d$. Thus, the claim follows.

$\square$

# L   Extension to SubGaussian noise

We briefly sketch the extension to SubGaussian noise. First, we define some notation: If $Y$ is a random variable, define $\|Y\|_{\psi_2} := \inf\{s > 0 : E\frac{Y^2}{s^2} \leq 1\}$, i.e., the 2-Orlicz norm. If $Y$ is a random vector, then $\|Y\|_{\psi_2} = \sup_{v:\|v\|_2=1} \left\| v^\top Y \right\|_{\psi_2}$ (see [32] for a reference).

Let $n \geq d$ and fix a set of measurements $x_{I_1}, \ldots, x_{I_n}$ and let $y_1, \ldots, y_n$ be the associated observations where we assume $y_i = x_i^\top \theta + \eta_i$ for $\eta_i$ is independent mean-0 subGauss(1) noise. Define the matrix

$$X = \begin{pmatrix} x_{I_1}^\top \\ \vdots \\ x_{I_T}^\top \end{pmatrix}$$

Define $\widehat{\theta} = (X^\top X)^{-1}X^\top Y$. Note that $\widehat{\theta} - \theta = (X^\top X)^{-1}X^\top \eta$. Note that $\|\eta\|_{\psi_2} \leq 1$. For any $v \in \mathbb{R}^d$,

$$
\left\|v^\top (X^\top X)^{-1}X^\top \eta\right\|_{\psi_2} = \left\|X(X^\top X)^{-1}v\right\|_2 \left\|\frac{1}{\|X(X^\top X)^{-1}v\|_2}v^\top(X^\top X)^{-1}X^\top\eta\right\|_{\psi_2}
$$
$$
\leq \left\|X(X^\top X)^{-1}v\right\|_2
$$
$$
= \|v\|_{(X^\top X)^{-1}}
$$

This shows that $\left\|v^\top(X^\top X)^{-1}X^\top \eta\right\|_{\psi_2} \leq \left\|v^\top(X^\top X)^{-1}X^\top \tilde{\eta}\right\|_{\psi_2}$ where $\tilde{\eta} \sim N(0, I)$. Thus, applying Theorem 8.5.5 and Talagrand's majorizing measure theomem (Theorem 8.6.1) from [32] yields for all $z \in \mathcal{Z} \setminus \{z_*\}$

$$
(z_* - z)^\top \widehat{\theta} \geq (z_* - z)^\top \theta - c\Big(\mathbb{E}_{\eta \sim N(0, I_d)}\Big[\sup_{z \in \mathcal{Z}\setminus\{z_*\}} (z_* - z)^\top A^{-1/2}\eta\Big]
$$
$$
- \sqrt{2 \sup_{z \in \mathcal{Z}\setminus\{z_*\}} \|z_* - z\|_{A^{-1}}^2 \log(\tfrac{1}{\delta})}\Big),
$$

where $c > 0$ is a universal constant, which is the essential concentration inequality used for the arguments in this paper.

## M   Experiment Details

**Combinatorial Bandit Experiments:** We used Python 3 and parallized the simulations on an Intel(R) Xeon(R) CPU E5-2690. For each experiment, we generate noise from a standard normal distribution. We used the stochastic mirror descent algorithm described in Section K, but let $\lambda \in \boldsymbol{\Delta}$ (instead of $\tilde{\boldsymbol{\Delta}}$). We ran the algorithm for 1000 iterations with a batch size of 10 on all experiments. Once we obtained a $\lambda \in \boldsymbol{\Delta}$, we used 2,000 samples to form an empirical mean to estimate the Gaussian width. We considered the setting where it is known that $\max_z \Delta_z \leq 2d$, which holds for example when $\theta \in [-1, 1]$, and thus solved

$$
\inf_{\lambda \in \boldsymbol{\Delta}} \mathbb{E}_{\eta \sim N(0,I)}[\max_{z \in \mathcal{Z}} \frac{(\tilde{z}_k - z)^\top A(\lambda)^{-1/2}\eta}{2^{-k} \cdot 2d + \widehat{\theta}_k^\top(\tilde{z}_k - z)}]^2
$$

instead of (27). We rounded our designs $\tau_k \lambda$ simply by taking the ceiling (which only incurs a loss of an additive factor of $d$ because $|\mathcal{X}| \leq d$.

To implement CLUCB, we use a state-of-the-art anytime confidence bound (inequality (2) from [17]), which is much better than the one used in [10]. For the uniform allocation algorithm, we use the termination condition that one obtains from applying the TIS inequality (Theorem 5.8 in [2]) to the process $\widehat{\theta}^\top(z - z')$.

We used 20 trials for the matching experiment, 30 trials for the shortest path experiment, and 60 trials for the biclique experiment. We generated 95% confidence intervals using the bootstrap.

**Transductive Linear Bandits:** We made two main changes to the algorithm as written, both focused on computing the objective $\inf_{\lambda \in \Delta} \tau(\lambda; \mathcal{Z}_k)$ more effectively. Firstly, we considered two different subproblems: $\min_{\lambda \in \Delta} \mathbb{E}_{\eta \sim N(0,I)}[\max_{z', z \in \mathcal{Z}_k}(z - z')^\top A(\lambda)^{-1/2}\eta]^2$ and $\min_\lambda \max_{z,z'} \|z' - z\|_{A(\lambda)^{-1}}^2$. In the setting where there are extremely large number of arms, it is not practical to take a max over all pairs of them - so in both subproblems we only took the max over $\hat{z}_k - \mathcal{Z}_k$ where $\hat{z}_k = \operatorname{argmax}_{z \in \mathcal{Z}_k} \hat{\theta}_k^\top z_k$. To justify this, we point out that by Theorem 7.5.2 of [32] $\mathbb{E}_{\eta \sim N(0,I)}[\max_{z', z \in \mathcal{Z}_k}(z - z')^\top A(\lambda)^{-1/2}\eta] = 2\mathbb{E}_{\eta \sim N(0,I)}[\max_{z \in \mathcal{Z}_k}(\hat{z}_k - z)^\top A(\lambda)^{-1/2}\eta]$, and $\min_\lambda \max_{z,z'} \|z' - z\|_{A(\lambda)^{-1}}^2 \leq 4 \min_\lambda \max_{z \in \mathcal{Z}_k} \|\hat{z}_k - z\|_{A(\lambda)^{-1}}^2$. Motivated by this, we computed the distribution $\lambda' = \operatorname{argmin}_\lambda \mathbb{E}_{\eta \sim N(0,I)}[\max_{z \in \mathcal{Z}_k}(\hat{z}_k - z)^\top A(\lambda)^{-1/2}\eta]$ and $\lambda'' = \min_\lambda \max_z \|\hat{z}_k - z\|_{A(\lambda)^{-1}}^2$ and set $\lambda_k = (\lambda' + \lambda'')/2$. Note that using this distribution only makes the algorithm perform worst than if the optimal - it does not affect correctness in anyway.

**Fixed Budget:** As in the previous, we computed an allocation not using $\gamma(Z_k)$ but rather a minimum over the differences $\hat{z}_k - \mathcal{Z}_k$.