[Reviews · NeurIPS 2020]

Review 1

Summary and Contributions: The authors propose a new sample-complexity parameter for best-arm identification in linear / transductive / combinatorial bandits for the moderate confidence regime. The parameter is motivated by a concentration inequality for the suprema of a Gaussian process, and is shown to be unavoidable for a non-interactive oracle policy. The authors proceed with proposing new algorithms that optimize the new sample-complexity bound directly, both for the fixed-confidence and the fixed-budget setting. An oracle efficient variant is proposed as well. All algorithms are shown to (nearly) match the oracle (non-interactive) lower bound. Empirically, the new approach is shown to outperform previous methods.

Strengths: This paper goes significantly beyond prior work on best arm identification by optimizing the design not only for the asymptotic regime, but for a lower order (moderate sample) complexity parameter as well. The approach is well motivated and theoretically grounded. It is shown that the new sample complexity can be larger than the asymptotic sample complexity by a dimensional factor. The empirical evaluation suggests that the proposed approach significantly improves on previous work.

Weaknesses: The lower bound assumes a non-interactive policy, so leaves open the possibility that an adaptive policy can outperform the bound. The oracle efficient variant of the problem achieves the lower bound up to a factor log(1/delta), and it is left open if the gap is unavoidable. As expected, the maximal inequality used to motivate the new sample complexity parameter, appears in the proof. The concentration bound requires iid/fixed design data, hence the algorithm proceed in stages and discards data from previous rounds.

Correctness: After a high-level check, the proofs appear correct to me. The experiments are well-suited to illustrate/compare the approach.

Clarity: Overall the paper is well written and provides enough intuition/motivation to make the key ideas easy to follow. One point that is unclear to me, is that Algorithm 2 seems to optimize only \gamma^*, and not \gamma^*log(1/delta)+ \rho^* as in Algorithm 1. Do I understand this correctly, and is this the source of the additional log(1/delta) factor in the bound? Equation (3),(4) and (5) would benefit from more explanation/motivation.

Relation to Prior Work: Prior work is adequately discussed and compared to.

Reproducibility: Yes

Additional Feedback: line 520: prsent -> present Update: I have read the author's response and my score remains the same.


Review 2

Summary and Contributions: This paper studies the pure-exploration linear bandit problem using Gaussian process and hence improving the performance of algorithms devised from a union bound. The paper provides a new lower bound and matching upper bound that improves the previous bounds. The paper also develops an efficient combinatorial bandit algorithm.

Strengths: The application of a Gaussian process to improve the dependence of a union bound is a neat idea. I believe the result is significant and could contribute nicely to the conference publications.

Weaknesses: However, I do find that the presentation of the results needs some improvement. The lower bound, which seems to be a major contribution of the paper, is very confusing. I believe it requires some revising to make it clearer.

Correctness: Yes. I believe so.

Clarity: Not very clear.

Relation to Prior Work: Yes.

Reproducibility: Yes

Additional Feedback: * My major confusion comes from section 4. Theorem 3 and the texts before it. How does the algorithm use theta? If theta is given to the algorithm, then no samples need to be used. What is theta^*? I do not actually understand the statement of Theorem 3. * Another minor issue is that you claim when delta->0, the d * rho* does not violates the upper bound, does it mean log(1/delta) is significantly larger than d when delta -> 0. Some other comments: * What if you consider an approximate z, i.e., |\hat{z} - z^*|<=eps? In this case, I think applying a union bound over an epsilon-net of Z gives you a d*log(1/eps)*rho* bound. * The work requires the linear optimization oracle, which might not be efficient when Z is big. But when the oracle is efficient, does it mean that its structure can also guarantee a small union bound? Is there any relationship between the sample complexity of the algorithm with the efficiency of the oracle? * For rho* and gamma*, can you compute it or compute an upper bound of from an algorithm instance? * When you say Theorem 4 has an actual factor of d, did you mean the last term? And why you claim your upper bound "matches the lower bound"?


Review 3

Summary and Contributions: The paper tackles pure exploration problems with a large number of arms such as combinatorial bandits. The main idea is to replace the union bound over the arms with a quantity that could be much smaller in favorable settings. The authors use the Tsirelson-Ibragimov-Sudakov inequality to bring into play the Gaussian width of the set of suboptimal arms which is then minimized by finding appropriate allocations of experiments to arms. The paper presents algorithms both for the fixed confidence case and the fixed budget case assuming access to linear optimization oracle over the arm set. In terms of sample complexity, the algorithms are nearly optimal for matroid arm sets. Finally, the experiments demonstrate substantially improved performance over CLUCB and RAGE.

Strengths: The paper does a good job on connecting the theory of extrema of gaussian processes with the practically relevant problem of sequential experimentation and identification of the best alternative among a large class of alternatives. The claims in the paper are sound and the empirical evaluation conclusively supports the claims in the paper. Furthermore, the paper makes significant and novel advances in its domain and is relevant to the NeurIPS community.

Weaknesses: The paper exploits the observation model (y = theta * x + gaussian) to its limits. The paper does not discuss a) what happens when the noise assumption is wrong b) what happens when the noise assumption is almost correct. c) if other noise assumptions can be dealt with in a similar way. The paper also assumes an exact maximization oracle and does not discuss if approximate oracles could be used.

Correctness: Claims, method and methodology are correct.

Clarity: Yes

Relation to Prior Work: The paper does a good job positioning itself with respect to existing literature and compare and contrast the various notions of sample complexity in the other papers and how they relate with the quantities proposed here for different setups.

Reproducibility: Yes

Additional Feedback: Tsirelson-Ibramov-Sudakov should be Tsirelson-Ibragimov-Sudakov ############ [After Rebuttal]: I have read the authors' response and my score remains the same. ############


Review 4

Summary and Contributions: This paper provides sample complexity bounds for the pure-exploration linear bandits problem. In this problem, there is an unknown d-dimensional vector θ. There is a set X of d-dimensional measurement vectors and a set Z of d-dimensional candidate vectors. At each round, an agent selects a vector x ∈ X and observes <x, θ> + η, where η is mean-0 Gaussian noise. In the end, the goal is to find a vector in argmax_{z ∈ Z} <z, θ>. The authors describe two ways to evaluate an algorithm for this problem. The first is the “fixed-confidence” setting: given a confidence parameter δ, how many samples x ∈ X are sufficient to ensure that with probability 1 - δ, the algorithm’s output ẑ is in argmax_{z ∈ Z} <z, θ>? The second is the “fixed-budget” setting: after T rounds, for some budget T, what’s the difference between <ẑ, θ> and max_{z ∈ Z} <z, θ>? Prior research (I think in the fixed-confidence setting) by Fiez et al. [’19] proposed an algorithm whose sample complexity grows with log(|Z|) due to a union bound over all |Z| arms. The overall goal of this paper is to replace log(|Z|) with a term that depends on the Gaussian-width of the arm set. The authors’ bounds depend crucially on two terms ρ* (line 124) and ɣ* (line 125) which seem to essentially be measurements of the width of the arm sets. The first result is a lower bound on the sample complexity in the fixed-confidence setting of a particular “non-interactive” algorithm, which is endowed with knowledge of the vector θ. It chooses the arms x_1, x_2, … for all rounds up front, sets θ’ to be MLE, and outputs argmax_{z ∈ Z} <z, θ’>. (Given knowledge of θ, I’m not sure why it doesn’t just output argmax_{z ∈ Z} <z, θ>, but I guess this is just a thought experiment?) They prove that this algorithm requires Ω(ρ* + ɣ*log(1/δ)) samples. They also provide an algorithm with sample complexity Õ(ρ* + ɣ*log(1/δ) + d), which nearly matches the lower bound except for the extra factor of d. When Z is combinatorial (X is the standard basis and Z ⊆ {0,1}), they provide a computationally efficient algorithm given access to a linear optimization oracle (Equation (1)). They also provide guarantees for the fixed-budget setting. Finally, they provide experiments demonstrating that in the settings they consider their algorithm requires far fewer samples than several baselines.

Strengths: I am not particularly familiar with this area, but it seems natural and interesting that sample complexity bounds need not depend on the number of arms |Z|. The experiments also seem to make a compelling case that this algorithm’s sample complexity in the experimental setups considered is much smaller than the baselines by [Chen et al., ’14; Fiez et al., ’12].

Weaknesses: It’s definitely nice that the lower bound of Ω(ρ* + ɣ*log(1/δ)) for the non-interactive MLE algorithm is similar to the proposed algorithm’s upper bound. Of course, this would be a lot stronger if the lower bound held for any algorithm, not just this specific non-interactive MLE algorithm. The authors write that this is “presumably the best possible non-interactive algorithm,” but I would appreciate a sentence or two justifying this claim. Since this algorithm is given access to the vector θ, it seems like the “best possible non-interactive algorithm" would simply output argmax_{z ∈ Z} <z, θ>, which of course requires no samples, so maybe the authors mean "best possible" non-trivial non-interactive algorithm? Also, to be upfront, I would mention in the abstract that the lower bound holds for a specific algorithm, not any algorithm. On a related note, “near-optimal” seems like a bit of an overstatement given the fact that the lower bound is algorithm-specific and the fact that the upper bound has an extra Õ(d) additive factor. I’m not sure if this is because I’m not particularly familiar with this area, but I found the paper somewhat difficult to follow. Here are a few ways it might be clarified: - The notation P_θ and E_θ are used in several places (Definition 1, Theorem 2, …) but I’m a bit confused by what this is meant to denote. It seems like it’s saying that the expectation is over the draw of θ, but as far as I’m aware, θ is fixed… - The definitions of ρ* and ɣ* are said to be based off of the bound from Line 110. But the inner products that show up in the denominators of ρ* and ɣ* are not there in Line 110… Intuitively, why do they show up here? - It’s always really helpful for the reader to use words as well as mathematical notation to explain concepts. For example, the pseudo-code for Algorithm 1 is quite technical and the explanation in words before Theorem 4 is rather terse. It would be helpful to add a more reader-friendly description of Algorithm 1. The same goes for the other algorithms. - What is the intuition behind the “Linear bandits” experimental setup? Why are X and Z defined in those particular ways? ----------------------After rebuttal---------------------- Thanks for the clarifications. It would be great to see a lower bound that holds for any algorithm, but this seems like a good first step.

Correctness: I believe the claims are correct.

Clarity: I think clarity could be improved, as I described in the “weaknesses” section.

Relation to Prior Work: Yes, although I am not too familiar with the area, I did find the related work section (Section 7) helpful.

Reproducibility: Yes

Additional Feedback: Line 73: Are you saying that I = Z? Line 101: Is θ* a typo? Lines 124-125: The notation is alternating between z* and z_* and θ^Tz and <θ, z>. I’m not factoring this into my rating, but saying that this paper might help cut the time it takes to cure cancer in half is a very bold statement… I’d tone down the Broader Impact section.

[Author Response · NeurIPS 2020]

We thank the reviewers for their valuable and extensive feedback. We briefly recap our contributions. In our work we
provide improved lower bounds for the linear-bandit pure exploration problem, an algorithm that acheives this as an
upper bound, computationally efficient algorithms in the combinatorial bandit setting, a novel fixed-budget algorithm
for linear bandits, and empirical validation of our methods. We emphasize that our paper focuses on using methods
from empirical process theory, such as the TIS inequality and Gaussian width, to develop a novel geometric notion
of sample complexity, $\gamma^*$. Importantly $\gamma^*$ captures the *moderate confidence* sample complexity in the regime where
where $\delta$ is a constant such as $\delta = 0.05$, the typical case in practice. By optimizing $\gamma^*$, we design a *single algorithm*
whose sample complexity is the first to essentially be within a $\log(d)$ factor of every other moderate confidence sample
complexity result in the pure exploration MAB literature.

**Regarding comments on the lower bound by Reviewers 2 and 4:** Reviewers 2 and 4 point out there is an additive $d$
factor in our upper bounds. Theorem 8 in the supplementary material shows that there is a broad class of problems
where $d$ samples are necessary. For example, if $\theta$ is unconstrained in the sense that the learner has no apriori knowledge
about $\theta$ prior to the beginning of the game, then in combinatorial bandits at least $d$ samples are necessary. We will
expand this discussion in the final manuscript.

We thank reviewers 2 and 4 for their comments on the the lower bound, and will clarify the result in the final manuscript.
The purpose of the Theorem is to shed light on the following question: if an agent chooses a distribution over $\mathcal{X}$, samples
from it, and then terminates using the MLE (a very natural estimator), what is the best possible sample complexity? We
agree with reviewers 2 and 4 that of course given the true $\theta$, the natural approach is to compute $z_*$, but this estimator tells
us nothing about the sample complexity of the problem. By restricting the estimator to output the empirical maximizer
of the MLE, we shed light on what an algorithm without knowledge of $\theta$ could realistically achieve.

We also note that the lower bound can be strengthened quite easily to state: *if the non-interactive MLE takes fewer*
*than $c(\gamma^* + \log(1/\delta)\rho^*)$ samples on a given problem $(\mathcal{X}, \mathcal{Z}, \theta)$, then it makes a mistake with probability at least $\delta$.*
The argument proceeds similarly to the proof in the paper by showing that if fewer than $c(\gamma^* + \log(1/\delta)\rho^*)$ samples
are taken, then anti-concentration bounds of a normal random variable (e.g., Proposition 2.1.2 of [30]) and the TIS
inequality imply that with probability at least $\delta$, the algorithm makes a mistake.

**Reviewers 1 and 3:** Reviewer 1 is correct that the extra $\log(1/\delta)$ comes from only optimizing $\gamma^*$. We did this for
computational reasons and will include motivation and intuition in the final manuscript. We agree with reviewer 3
regarding the importance of extensions to an approximate oracle and of the noise assumption, and will discuss these in
the final manuscript. Our algorithms can be extended to the Sub-Gaussian noise case (see the supplementary material).

**Reviewer 2:** The suggested $\epsilon$-net would give an extremely loose sample complexity: $d\log(1/\epsilon)\rho^*$ whereas our
algorithm attains $c\log(d/\delta)\rho^*$ in the problem Top-K. The point of our paper is to find a data-efficient union bound in a
generic and computationally efficient manner. In addition, constructing an $\epsilon$-net directly is computationally burdensome.

"For $\rho^*$ and $\gamma^*$, can you compute it...": $\rho^*$ is always convex and $\gamma^*$ is convex in the combinatorial bandit setting; we
conjecture that $\gamma^*$ is convex in general. A linear maximization oracle can be used to compute $\gamma^*$.

"when the oracle is efficient, does it mean that its structure can also guarantee a small union bound?": There is no
clear-cut relationship. Proposition 2 gives an example where the union bound is large and the oracle is efficient.

"When you say Theorem 4 has an actual factor of d, did you mean the last term?": We are referring to the additive term
$+d$ in our sample complexity, which is essentially $\rho^* \log(1/\delta) + \gamma^* + d$.

**Reviewer 4:** "What is the intuition behind the "Linear bandits" experimental setup?..." The pure exploration linear
bandit problem is well established, see [12,27]. As we describe, it is also sufficiently general to encapsulate many
problems in MAB, such as combinatorial bandits. Our sample complexity results are essentially within a $\log(d)$ factor
of every result in pure exploration MAB.

"It's always really helpful for the reader to use words as well as mathematical notation to explain concepts..." Thank
you for the feedback. We will add some clarifying sentences such as "at round $k$, Algorithm 1 finds a design $\lambda_k$ that
makes the application of the TIS inequality as tight as possible when applied to the remaining items in $\mathcal{Z}$. Then, it
takes enough samples to guarantee that all items z with gap larger than $2^{-k}\Gamma$ are eliminated." We will provide a similar
description of our other algorithms.

"The notation $P_\theta$ and $E_\theta$ are used in several places..." $\theta$ is is not random; $\mathbb{P}_\theta$ denotes the probability law induced by $\theta$
and the sampling rule of an algorithm (see [23] for similar usage). We will clarify its meaning in the final manuscript.

"The definitions of $\rho^*$ and $\gamma^*$ are said to be based off..." If $c\rho^* \log(|\mathcal{Z}|/\delta)$ samples are taken, then manipulating the
inequality (2) shows that each gap is estimated well enough to identify $z_*$ (see [27] Section 3 for a helpful discussion).

[Meta-Review · NeurIPS 2020]

The reviewers generally like this work, some quite a lot. I also like the idea and recommend acceptance. There are some issues about clarity in specific sections, as pointed out by several reviewers. These should be addressed carefully in the final version.